# Expressive Power of Invariant and Equivariant Graph Neural Networks

**Waïss Azizian**
ENS, PSL University, Paris, France
`waiss.azizian@ens.fr`

**Marc Lelarge**
INRIA & ENS, PSL University, Paris, France
`marc.lelarge@ens.fr`

## Abstract

Various classes of Graph Neural Networks (GNN) have been proposed and shown to be successful in a wide range of applications with graph structured data. In this paper, we propose a theoretical framework able to compare the expressive power of these GNN architectures. The current universality theorems only apply to intractable classes of GNNs. Here, we prove the first approximation guarantees for practical GNNs, paving the way for a better understanding of their generalization. Our theoretical results are proved for invariant GNNs computing a graph embedding (permutation of the nodes of the input graph does not affect the output) and equivariant GNNs computing an embedding of the nodes (permutation of the input permutes the output). We show that Folklore Graph Neural Networks (FGNN), which are tensor based GNNs augmented with matrix multiplication are the most expressive architectures proposed so far for a given tensor order. We illustrate our results on the Quadratic Assignment Problem (a NP-Hard combinatorial problem) by showing that FGNNs are able to learn how to solve the problem, leading to much better average performances than existing algorithms (based on spectral, SDP or other GNNs architectures). On a practical side, we also implement masked tensors to handle batches of graphs of varying sizes.

## 1 Introduction

Graph Neural Networks (GNN) are designed to deal with graph structured data. Since a graph is not changed by permutation of its nodes, GNNs should be either invariant if they return a result that must not depend on the representation of the input (typically when building a graph embedding) or equivariant if the output must be permuted when the input is permuted (typically when building an embedding of the nodes). More fundamentally, incorporating symmetries in machine learning is a fundamental problem as it allows to reduce the number of degree of freedom to be learned.

**Deep learning on graphs.** This paper focuses on learning deep representation of graphs with network architectures, namely GNN, designed to be invariant to permutation or equivariant by permutation. From a practical perspective, various message passing GNNs have been proposed, see Dwivedi et al. (2020) for a recent survey and benchmarking on learning tasks. In this paper, we study 3 architectures: Message passing GNN (MGNN) which is probably the most popular architecture used in practice, order-$k$ Linear GNN ($k$-LGNN) proposed in Maron et al. (2018) and order-$k$ Folklore GNN ($k$-FGNN) first introduced by Maron et al. (2019a). MGNN layers are local thus highly parallelizable on GPUs which make them scalable for large sparse graphs. $k$-LGNN and $k$-FGNN are dealing with representations of graphs as tensors of order $k$ which make them of little practical use for $k \geq 3$.

In order to compare these architectures, the separating power of these networks has been compared to a hierarchy of graph invariants developed for the graph isomorphism problem. Namely, for $k \geq 2$, $k$-WL$(G)$ are invariants based on the Weisfeiler-Lehman tests (described in Section 4.1). For each $k \geq 2$, $(k+1)$-WL has strictly more separating power than $k$-WL (in the sense that there is a pair of non-isomorphic graphs distinguishable by $(k+1)$-WL and not by $k$-WL). GIN (which are invariant MGNN) introduced in Xu et al. (2018) are shown to be as powerful as 2-WL. In Maron et al. (2019a), Geerts (2020b) and Geerts (2020a), $k$-LGNN are shown to be as powerful as $k$-WL and 2-FGNN is shown to be as powerful as 3-WL. In this paper, we extend this last result about $k$-FGNN to general values of $k$. So in term of separating power, when restricted to tensors of order $k$, $k$-FGNN is the

most powerful architecture among the ones considered in this work. This means that for a given pair of graphs $G$ and $G'$, if $(k+1)$-WL$(G) \neq (k+1)$-WL$(G')$, then there exists a $k$-FGNN, say $\mathbf{GNN}_{G,G'}$ such that $\mathbf{GNN}_{G,G'}(G) \neq \mathbf{GNN}_{G,G'}(G')$.

**Approximation results for GNNs.** Results on the separating power of GNNs only deal with pairwise comparison of graphs: we need a priori a different GNN for each pair of graphs in order to distinguish them. Such results are of little help in a practical learning scenario. Our main contribution in this paper overcomes this issue and we show that a single GNN can give a meaningful representation for all graphs. More precisely, we characterize the set of functions that can be approximated by MGNNs, $k$-LGNNs and $k$-FGNNs respectively. Standard Stone-Weierstrass theorem shows that if an algebra $\mathcal{A}$ of real continuous functions separates points, then $\mathcal{A}$ is dense in the set of continuous function on a compact set. Here we extend such a theorem to general functions with symmetries and apply it to invariant and equivaraint functions to get our main result for GNNs. As a consequence, we show that $k$-FGNNs have the best approximation power among architectures dealing with tensors of order $k$.

**Universality results for GNNs.** Universal approximation theorems (similar to Cybenko (1989) for multi-layers perceptron) have been proved for linear GNNs in Maron et al. (2019b); Keriven & Peyré (2019); Chen et al. (2019). They show that some classes of GNNs can approximate any function defined on graphs. To be able to approximate any invariant function, they require the use of very complex networks, namely $k$-LGNN where $k$ tends to infinity with $n$ the number of nodes. Since we prove that any invariant function less powerful than $(k+1)$-WL can be approximated by a $k$-FGNN, letting $k$ tends to infinity directly implies universality. Universality results for $k$-FGNN is another contribution of our work.

**Equivariant GNNs.** Our second set of results extends previous analysis from invariant functions to equivariant functions. There are much less results about equivariant GNNs: Keriven & Peyré (2019) proves the universality of linear equivariant GNNs, and Maehara & Hoang (2019) shows the universality of a new class of networks they introduced. Here, we consider a natural equivariant extension of $k$-WL and prove that equivariant $(k+1)$-LGNNs and $k$-FGNN can approximate any equivariant function less powerful than this equivariant $(k+1)$-WL for $k \geq 1$. At this stage, we should note that all universality results for GNNs by Maron et al. (2019b); Keriven & Peyré (2019); Chen et al. (2019) are easily recovered from our main results. Also our analysis is valid for graphs of varying sizes.

**Empirical results for the Quadratic Assigment Problem (QAP).** To validate our theoretical contributions, we empirically show that 2-FGNN outperforms classical MGNN. Indeed, Maron et al. (2019a) already demonstrate state of the art results for the invariant version of 2-FGNNs (for graph classification or graph regression). Here we consider the graph alignment problem and show that the equivariant 2-FGNN is able to learn a node embedding which beats by a large margin other algorithms (based on spectral method, SDP or GNNs).

**Outline and contribution.** After reviewing more previous works and notations in the next section, we define the various classes of GNNs studied in this paper in Section 3 : message passing GNN, linear GNN and folklore GNN. Section 4 contains our main theoretical results for GNNs. First in Section 4.2 we describe the separating power of each GNN architecture with respect to the Weisfeiler-Lehman test. In Section 4.3, we give approximation guarantees for MGNNs, LGNNs and FGNNs at fixed order of tensor. They cover both the invariant and equivariant cases and are our main theoretical contributions. For these, we develop in Section D a fine-grained Stone-Weierstrass approximation theorem for vector-valued functions with symmetries. Our theorem handles both invariant and equivariant cases and is inspired by recent works in approximation theory. In Section 6, we illustrate our theoretical results on a practical application: the graph alignment problem, a well-known NP-hard problem. We highlight a previously overlooked implementation question: the handling of batches of graphs of varying sizes. A PyTorch implementation of the code necessary to reproduce the results is available at https://github.com/mlelarge/graph_neural_net

## 2 RELATED WORK

The pioneering works that applied neural networks to graphs are Gori et al. (2005) and Scarselli et al. (2009) that learn node representation with recurrent neural networks. More recent message passing architectures make use of non-linear functions of the adjacency matrix (Kipf & Welling, 2016),

for example polynomials (Defferrard et al., 2016). For regular-grid graphs, they match classical convolutional networks which by design can only approximate translation-invariant functions and hence have limited expressive power. In this paper, we focus instead on more expressive architectures.

Following the recent surge in interest in graph neural networks, some works have tried to extend the pioneering work of Cybenko (1989); Hornik et al. (1989) for various GNN architectures. Among the first ones is Scarselli et al. (2009), which studied invariant message-passing GNNs. They showed that such networks can approximate, in a weak sense, all functions whose discriminatory power is weaker than 1-WL. Yarotsky (2018) described universal architectures which are invariant or equivariant to some group action. These models rely on polynomial intermediate layers of arbitrary degrees, which would be prohibitive in practice. Maron et al. (2019b) leveraged classical results about the polynomials invariant to a group action to show that $k$-LGNN are universal as $k$ tends to infinity with the number of nodes. Keriven & Peyré (2019) derived a similar result, in the more complicated equivariant case by introducing a new Stone-Weierstrass theorem. Similarly to Maron et al. (2019b), they require the order of tensors to go to infinity. Another route towards universality is the one of Chen et al. (2019). In the invariant setting, they show for a class of GNN that universality is equivalent to being able to discriminate between (non-isomorphic) graphs. However, the only way to achieve such discriminatory power is to use tensors of arbitrary high order, see also Ravanbakhsh (2020). Our work encompass and precise these results using high-order tensors as it yields approximation guarantees even at fixed order of tensor.

CPNGNN in Sato et al. (2019) and DimeNet in Klicpera et al. (2020) are message passing GNN incorporating more information than those studied here. Partial results about their separating power follows from Garg et al. (2020) which provides impossibility results to decide graph properties including girth, circumference, diameter, radius, conjoint cycle, total number of cycles, and $k$-cliques. Chen et al. (2020) studies the ability of GNNs to count graph substructures. Though our theorems are much more general, note that their results are improved by the present work. Note also, that if the nodes are given distinct features, MGNNs become much more expressive Loukas (2019) but looses their invariant or equivariant properties. Averaging i.e. relational pooling (RP) has been proposed to recover these properties Murphy et al. (2019a). However, the ideal RP, leading to a universal approximation, cannot be used for large graphs due to its complexity of $O(|V|!)$. Regarding the other classes of RPGNN i.e. the $k$-ary pooling (Murphy et al., 2019b), we will show how our general theorems in the invariant case can be applied to characterize their approximation power (see Section 5).

Note that for neural networks on sets, the situation is a bit simpler. Efficient architectures such as DeepSets (Zaheer et al., 2017) or PointNet (Qi et al., 2017) have been shown to be invariant universal. Similar results exist in the equivariant case (Segol & Lipman, 2020; Maron et al., 2020), whose proofs rely on polynomial arguments. Though this is not our main motivation, our approximation theorems could also be applied in this context see Sections D.3 and D.4.

## 2.1 Notations: graphs as tensors

We denote by $\mathbb{F}, \mathbb{F}_0, \mathbb{F}_{1/2}, \mathbb{F}_1, \ldots$ arbitrary finite-dimensional spaces of the form $\mathbb{R}^p$ (for various values of $p$) typically representing the space of features. Product of vectors in $\mathbb{R}^p$ always refer to component-wise product. There are two ways to see graphs with features. First, graphs can be seen as tensors of order $k$: $G \in \mathbb{F}^{n^k}$. The classical representation of a graph by its (weighted) adjacency matrix for $k = 2$ is a tensor of order 2 in $\mathbb{R}^{n^2}$. This case allows for features on edges by replacing $\mathbb{R}^{n^2}$ with $\mathbb{F}^{n^2}$ where $\mathbb{F}$ is some $\mathbb{R}^p$. Second, graphs can also be represented by their discrete structure with an additional feature vector. More exactly, denote by $\mathcal{G}_n$ the set of discrete graphs $G = (V, E)$ with $n$ nodes $V = [n]$ and edges $E \subseteq V^2$ (with no weights on edges). Such a $G \in \mathcal{G}_n$ with a vector $h^0 \in \mathbb{F}^n$ represents a graphs with features on the vertices.

## 2.2 Definitions: invariant and equivariant operators

Let $[n] = \{1, \ldots, n\}$. The set of permutations on $[n]$ is denoted by $\mathcal{S}_n$. For $G \in \mathbb{F}^{n^k}$ and $\sigma \in \mathcal{S}_n$, we define: $(\sigma \star G)_{\sigma(i_1), \ldots, \sigma(i_k)} = G_{i_1, \ldots, i_k}$. Note that the $\star$ operation is valid between a permutation in $\mathcal{S}_n$ and a graph $G$ as soon as the number of nodes of $G$ is $n$, i.e. it is valid for any order $k$ tensor

representation of the graph. Two graphs $G_1, G_2$ are said isomorphic if they have the same number of nodes and there exists a permutation $\sigma$ such that $G_1 = \sigma \star G_2$.

**Definition 1.** A function $f : \mathbb{F}_0^{n^k} \to \mathbb{F}_1$ is said to be invariant if $f(\sigma \star G) = f(G)$ for every permutation $\sigma \in \mathcal{S}_n$ and every $G \in \mathbb{F}_0^{n^k}$. A function $f : \mathbb{F}_0^{n^k} \to \mathbb{F}_1^{n^\ell}$ is said to be equivariant if $f(\sigma \star G) = \sigma \star f(G)$ for every permutation $\sigma \in \mathcal{S}_n$ and every $G \in \mathbb{F}_0^{n^k}$.

Note that composing an equivariant function with an invariant function gives an invariant function. For $k \geq 1$, we define the invariant summation layer $S^k : \mathbb{F}^{n^k} \to \mathbb{F}$ by $S^k(G) = \sum_{\mathbf{i} \in [n]^k} G_\mathbf{i}$ for $G \in \mathbb{F}^{n^k}$. We also define the equivariant reduction layer $S_1^k : \mathbb{F}^{n^k} \to \mathbb{F}^n$ as follows: $S_1^k(G)_i = \sum_{1 \leq i_2 \ldots i_k \leq n} G_{i,i_2,\ldots i_k}$. For message passing GNN, we will use the equivariant layer $\mathrm{Id} + \lambda S^1 : \mathbb{F}^n \to \mathbb{F}^n$ defined by, $(\mathrm{Id} + \lambda S^1)(G)_i = G_i + \lambda S^1(G)$, where $\lambda \in \mathbb{R}$ is a learnable parameter.

In the sequel, we will need a mapping $I^k$ lifting the input graph to a higher order tensor. We denote by $I^k : \mathbb{F}_0^{n^2} \to \mathbb{F}_1^{n^k}$ the initialization function mapping for a given graph each $k$-tuple to its isomorphism type. We refer to the appendix Section C.3 for a precise description of this linear equivariant function. Note at this stage that $I^2$ is given by, for $G \in \mathbb{F}^{n^2}$, $I(G)_{i,j} = (G_{i,j}, \delta_{i,j})$ where $\delta_{i,j}$ is 0 if $i \neq j$ and 1 otherwise. Indeed for a pair of nodes $i, j$ in a graph (without features), there are only three isomorphism types: $i = j$; $i \neq j$ and $(i, j)$ is an edge; $i \neq j$ but $(i, j)$ is not an edge.

## 3 GNN DEFINITIONS

In this section, we define the various GNN architectures studied in this paper. In all architectures, there is a main building block or layer mapping $\mathbb{F}_t^{n^k}$ to $\mathbb{F}_{t+1}^{n^k}$ where $\mathbb{F}_t^{n^k}$ can be seen as the space for the representation of the graph at layer $t$. We will define three different types of layers for message passing GNN, linear GNN and folklore GNN. The case $k = 2$ is probably the most interesting case from a practical point view and corresponds to a case where a layer takes as input a graph (with features on nodes and edges) and produces as output a graph (with new features on nodes and edges). For each type of GNNs, there will be an invariant and an equivaraint version. All architectures will share the last function: $m_I : \mathbb{F}_{T+1} \to \mathbb{F}$ for the invariant case and $m_E : \mathbb{F}_{T+1}^n \to \mathbb{F}^n$ for the equivariant case which are continuous functions. It is typically modeled by a Multi Layer Perceptron, which is applied on each component for the equivariant case. In words, each network takes as input a graph $G \in \mathbb{F}_0^{n^2}$, produces in the invariant case a graph embedding in $\mathbb{F}_{T+1}$ and in the equivaraint case a node embedding in $\mathbb{F}_{T+1}^n$, then these embeddings are passed through the function $m_I$ or $m_E$ respectively to get a feature in $\mathbb{F}$ or $\mathbb{F}^n$ for the learning task.

### 3.1 MESSAGE PASSING GNN

Message passing GNN (MGNN) are defined for classical graphs $G$ with features on the nodes. More exactly they take as input a discrete graph $G = (V, E) \in \mathcal{G}_n$ and features on the nodes $h^0 \in \mathbb{F}^n$. MGNN are then defined inductively as follows: let $h_i^\ell \in \mathbb{F}_\ell$ denote the feature at layer $\ell$ associated with node $i$, the updated features $h_i^{\ell+1}$ are obtained as: $h_i^{\ell+1} = f\left(h_i^\ell, \{\{h_j^\ell\}\}_{j \sim i}\right)$, where $j \sim i$ means that nodes $j$ and $i$ are neighbors in the graph $G$, i.e. $(i, j) \in E$, and the function $f$ is a learnable function taking as input the feature vector of the center vertex $h_i^\ell$ and the multiset of features of the neighboring vertices $\{\{h_j^\ell\}\}_{j \sim i}$. Indeed, it follows from Lem. 33 in Appendix, that any such function $f$ can be approximated by a layer of the form,

$$h_i^{\ell+1} = f_0\left(h_i^\ell, \sum_{j \sim i} f_1\left(h_i^\ell, h_j^\ell\right)\right), \tag{1}$$

where $f_0 : \mathbb{F}_\ell \times \mathbb{F}_{\ell+1/2} \to \mathbb{F}_{\ell+1}$ and $f_1 : \mathbb{F}_\ell \times \mathbb{F}_\ell \to \mathbb{F}_{\ell+1/2}$, so that $\mathbb{F}_\ell$ is the field for the features at the $\ell$-th layer. We call such a function a message passing layer and denote it by $F : \mathbb{F}_\ell^n \to \mathbb{F}_{\ell+1}^n$ (note that $F$ depends implicitly from the graph). Then an equivariant message passing GNN is simply obtained by the composition of message passing layers: $F_T \circ \ldots F_2 \circ F_1$, where each $F_i$ is a message passing layer. Clearly since each $F_i$ is equivariant, this message passing GNN is also equivariant and produces features on each node in the space $\mathbb{F}_T$. In order to obtain an invariant GNN, we apply

an invariant function from $\mathbb{F}_T^n \to \mathbb{F}_{T+1}$ on the output of an equivariant message passing GNN. In practice, a symmetric function is applied on the vectors of features indexed by the nodes, typically the sum of the features $\sum_i (F_T \circ \ldots F_2 \circ F_1(G))_i$ is taken as an invariant feature for the graph $G$. With our notation, $S^1 \circ F_T \circ \ldots F_2 \circ F_1$ (where $S^1$ was defined in Section 2.2) defines an invariant message passing GNN.

Hence, we define the sets of message passing GNNs as follows:

$$\begin{aligned} \text{MGNN}_I &= \{m_I \circ S^1 \circ F_T \circ \ldots F_2 \circ F_1, \forall T\} \\ \text{MGNN}_E &= \{m_E \circ (\text{Id} + \lambda S^1) \circ F_T \circ \ldots F_2 \circ F_1, \forall T\} \end{aligned}$$

where $F_t : \mathbb{F}_t^n \to \mathbb{F}_{t+1}^n$ are message passing layers.

## 3.2 LINEAR GNN

We define the linear graph layer of order $k$ as $F : \mathbb{F}_\ell^{n^k} \to \mathbb{F}_{\ell+1}^{n^k}$, where for all $G \in \mathbb{F}_\ell^{n^k}$, $F(G) = f(L[G])$ where $L : \mathbb{F}_\ell^{n^k} \to \mathbb{F}_\ell^{n^k}$ is a linear equivariant function, and $f : \mathbb{F}_\ell \to \mathbb{F}_{\ell+1}$ is a learnable function applied on each of the $n^k$ features and $\mathbb{F}_\ell$ is the field for the features at the $\ell$-th layer.

We then define the sets of linear GNNs as follows:

$$\begin{aligned} k\text{-LGNN}_I &= \{m_I \circ S^k \circ F_T \circ \ldots F_2 \circ F_1 \circ I^k, \forall T\} \\ k\text{-LGNN}_E &= \{m_E \circ S_1^k \circ F_T \circ \ldots F_2 \circ F_1 \circ I^k, \forall T\} \end{aligned}$$

where $I^k : \mathbb{F}_0^{n^2} \to \mathbb{F}_1^{n^k}$ is defined in §2.2 and for $t \geq 1$, $F_t : \mathbb{F}_t^{n^k} \to \mathbb{F}_{t+1}^{n^k}$ are linear equivariant layers.

## 3.3 FOLKLORE GNN

The main building block of Folklore GNN (FGNN) is what we call the folklore graph layer (FGL) of order $k$ defined as follows: for $k \geq 1$, $F : \mathbb{F}_\ell^{n^k} \to \mathbb{F}_{\ell+1}^{n^k}$ where for all $G \in \mathbb{F}_\ell^{n^k}$ and all $\mathbf{i} \in [n]^k$,

$$F(G)_\mathbf{i} = f_0 \left( G_\mathbf{i}, \sum_{j=1}^n \prod_{w=1}^k f_w \left( G_{i_1,\ldots,i_{w-1},j,i_{w+1},\ldots,i_k} \right) \right), \tag{2}$$

where $f_0 : \mathbb{F}_\ell \times \mathbb{F}_{\ell+1/2} \to \mathbb{F}_{\ell+1}$ and $f_k : \mathbb{F}_\ell \to \mathbb{F}_{\ell+1/2}$ are learnable functions. As shown in Lem. 33 in Appendix, FGL is an equivariant function which is indeed very expressive.

For classical graphs $G \in \mathbb{F}_0^{n^2}$, we can now define 2-FGNN by composing folklore graph layers $F_t : \mathbb{F}_t^{n^2} \to \mathbb{F}_{t+1}^{n^2}$, so that $F_T \circ \ldots F_1 \circ F_0$ is an equivariant GNN producing a graph in $\mathbb{F}_{T+1}^{n^2}$. To obtain an invariant feature of the graph, we use the summation layer $S^2$ defined in Section 2.2 so that $S^2 \circ F_T \circ \ldots F_1 \circ F_0$ is now an invariant 2-FGNN. In order to define general $k$-FGNN, we first need to lift the classical graph to a tensor in $\mathbb{F}^{n^k}$, then we apply folklore graph layers of order $k$ and finally we need to project the tensor in $\mathbb{F}^{n^k}$ to a tensor in $\mathbb{F}^n$ for the equivariant version and to a tensor in $\mathbb{F}$ for the invariant version. The first step is done with the linear equivariant function $I^k : \mathbb{F}_0^{n^2} \to \mathbb{F}_1^{n^k}$ defined in Section 2.2. The last step is done with the reduction layer $S_1^k$ for the equivariant case and the summation layer $S^k$ for the invariant case, both defined in Section 2.2.

We define the sets of folklore GNNs as follows:

$$\begin{aligned} k\text{-FGNN}_I &= \{m_I \circ S^k \circ F_T \circ \ldots F_2 \circ F_1 \circ I^k, \forall T\} \\ k\text{-FGNN}_E &= \{m_E \circ S_1^k \circ F_T \circ \ldots F_2 \circ F_1 \circ I^k, \forall T\} \end{aligned}$$

where $F_t : \mathbb{F}_t^{n^k} \to \mathbb{F}_{t+1}^{n^k}$ are FGLs.

## 4 THEORETICAL RESULTS FOR GNNS

### 4.1 WEISFEILER-LEHMAN INVARIANT AND EQUIVARIANT VERSIONS

We introduce a family of functions on graphs parametrized by integers $k \geq 2$ developed for the graph isomorphism problem and working with tuples of $k$ vertices. Each $k$-tuple $\mathbf{i} \in V^k = [n]^k$ is given

a color $c^0(\mathbf{i})$ corresponding to its isomorphism type (see Section B.2). The $k$-WL test relies on the following notion of neighborhood, defined by, for any $w \in [k]$, and $\mathbf{i} = (i_1, \ldots, i_k) \in V^k$, $N_w(\mathbf{i}) = \{(i_1, \ldots, i_{w-1}, j, i_{w+1}, \ldots, i_k) : j \in V\}$. Then, the colors of the $k$-tuples are refined as follows, $c^{t+1}(\mathbf{i}) = \text{Lex}\left(c^t(\mathbf{i}), (C_1^t(\mathbf{i}), \ldots, C_k^t(\mathbf{i}))\right)$ where, for $w \in [k]$, $C_w^t(\mathbf{i}) = \left\{\!\!\left\{ c^t(\tilde{\mathbf{i}}) : \tilde{\mathbf{i}} \in N_w(\mathbf{i}) \right\}\!\!\right\}$ and the function Lex means that all occuring colors are lexicographically ordered and replaced by an initial segment of the natural numbers.

For a graph $G$, let $k$-$\text{WL}_I^T(G)$ denote the multiset of colors of the $k$-WL algorithm at the $T^{th}$ iteration. After a finite number of steps (which depends on the number of vertices in the graph), the algorithm stops because a stable coloring is reached (no color class of $k$-tuples is further divided). We denote by $k$-$\text{WL}_I(G)$ the multiset of colors in the stable coloring. This is a graph invariant that is usually used to test if graphs are isomorphic. The power of this invariant increases with $k$ Cai et al. (1989).

We now define an equivariant version of $k$-WL test to express the discriminatory power of equivariant architectures For this, we construct a coloring of the vertices from the coloring of the $k$-tuples given by the standard $k$-WL algorithm. Formally, define $k$-$\text{WL}_E^T : \mathbb{F}_0^{n^2} \to \mathbb{F}^n$ by, for $i \in V$: $k$-$\text{WL}_E^T(G)_i = \left\{\!\!\left\{ c^T(\mathbf{i}) : \mathbf{i} \in V^k, i_1 = i \right\}\!\!\right\}$. Similarly, define $k$-$\text{WL}_E(G) = \left\{\!\!\left\{ c(\mathbf{i}) : \mathbf{i} \in V^k, i_1 = i \right\}\!\!\right\}$ where $c(\mathbf{i})$ is the stable coloring obtained by the algorithm.

## 4.2 SEPARATING POWER OF GNNS

We formulate our results using the equivalence relation introduced by Timofte (2005), which characterizes the separating power of a set of functions.

**Definition 2.** Let $\mathcal{F}$ be a set of functions $f$ defined on a set $X$, where each $f$ takes its values in some $Y_f$. The equivalence relation $\rho(\mathcal{F})$ defined by $\mathcal{F}$ on $X$ is: for any $x, x' \in X$,

$$(x, x') \in \rho(\mathcal{F}) \iff \forall f \in \mathcal{F}, \ f(x) = f(x').$$

Given two sets of functions $\mathcal{F}$ and $\mathcal{E}$, we say that $\mathcal{F}$ is more separating (resp. strictly more separating) than $\mathcal{E}$ if $\rho(\mathcal{F}) \subseteq \rho(\mathcal{E})$ (resp. $\rho(\mathcal{F}) \subsetneq \rho(\mathcal{E})$). Note that all the functions in $\mathcal{F}$ and $\mathcal{E}$ need to be defined on the same set but can take values in different sets. For example, we can easily see that for the $k$-WL algorithm defined above, the equivariant version is more separating than the invariant one.

Some properties of the WL hierarchy of tests can be rephrased with the notion of separating power. In particular, Cai et al. (1989) showed that $(k+1)$-$\text{WL}_I$ distinguishes strictly more than $k$-$\text{WL}_I$, which can be rewritten simply as (for a function $f$, we write $\rho(f)$ for $\rho(\{f\})$)

$$\rho((k+1)\text{-WL}_I) \subsetneq \rho(k\text{-WL}_I) . \tag{3}$$

This notion of separating power enables us to concisely summarize the current knowledge about the discriminatory power of classes of GNN.

**Proposition 3.** We have, for $k \geq 2$,

$$\rho(MGNN_I) = \rho(2\text{-}WL_I) \qquad\qquad \rho(MGNN_E) = \rho(2\text{-}WL_E) \tag{4}$$
$$\rho(k\text{-}LGNN_I) = \rho(k\text{-}WL_I) \qquad\qquad \rho(k\text{-}LGNN_E) \subseteq \rho(k\text{-}WL_E) \tag{5}$$
$$\rho(k\text{-}FGNN_I) = \rho((k+1)\text{-}WL_I) \qquad \rho(k\text{-}FGNN_E) = \rho((k+1)\text{-}WL_E) \tag{6}$$

Only results about the invariant cases were previously known: (4) comes from Xu et al. (2018), (5) from Maron et al. (2018) Geerts (2020a) and one inclusion of (6) comes from Maron et al. (2019a). The equality in (6) for general $k \geq 2$ is proved in Section C.

Note that for $k = 2$, all GNNs are dealing with tensors of order 2 i.e. with the adjacency matrix of the graph. However, the complexities of the various layers are quite different: for the message passing GNN, all computations are local (scaling with the maximum degree in the graph) and can be done in parallel; for the linear layer, there are only 15 linear functions from $\mathbb{R}^{n^2} \to \mathbb{R}^{n^2}$ for all values of $n$ (Maron et al., 2018); the folklore layer involves a (dense) matrix multiplication of shape $n \times n$. If 2-FGNN is the most complex architecture, we see that it has the best separating power among all architectures proposed so far dealing with tensors of order 2.

### 4.3 APPROXIMATION RESULTS FOR GNNS

For $X, Y$ finite-dimensional spaces, let us denote by $\mathcal{C}_I(X, Y), \mathcal{C}_E(X, Y)$, , the set of invariant, respectively equivariant, continuous functions from $X$ to $Y$. The closure of a class of function $\mathcal{F}$ for the uniform norm is denoted by $\overline{\mathcal{F}}$. Our result extend easily to graphs of varying sizes but this is deferred to Section F.2 for clarity.

The theorem below states in particular that the class $k$-FGNN can approximate any continuous function that is less separating than $(k + 1)$-WL in the invariant and in the equivariant cases.

**Theorem 4.** *Let $K_{discr} \subseteq \mathcal{G}_n \times \mathbb{F}_0^n$, $K \subseteq \mathbb{F}_0^{n^2}$ be compact sets. For the invariant case, we have:*

$$\overline{MGNN_I} = \{f \in \mathcal{C}_I(K_{discr}, \mathbb{F}) : \ \rho\,(2\text{-}WL_I) \subseteq \rho\,(f)\}$$

$$\overline{k\text{-}LGNN_I} = \{f \in \mathcal{C}_I(K, \mathbb{F}) : \ \rho\,(k\text{-}WL_I) \subseteq \rho\,(f)\}$$

$$\overline{k\text{-}FGNN_I} = \{f \in \mathcal{C}_I(K, \mathbb{F}) : \ \rho\,((k+1)\text{-}WL_I) \subseteq \rho\,(f)\}$$

*For the equivariant case, we have:*

$$\overline{MGNN_E} = \{f \in \mathcal{C}_E(K_{discr}, \mathbb{F}^n) : \ \rho\,(2\text{-}WL_E) \subseteq \rho\,(f)\}$$

$$\overline{k\text{-}LGNN_E} = \{f \in \mathcal{C}_E(K, \mathbb{F}^n) : \ \rho\,(k\text{-}LGNN_E) \subseteq \rho\,(f)\} \supset \{f \in \mathcal{C}_E(K, \mathbb{F}^n) : \ \rho\,(k\text{-}WL_E) \subseteq \rho\,(f)\}$$

$$\overline{k\text{-}FGNN_E} = \{f \in \mathcal{C}_E(K, \mathbb{F}^n) : \ \rho\,((k+1)\text{-}WL_E) \subseteq \rho\,(f)\}$$

In the invariant case for $k = 2$, we have $\overline{MGNN_I} = \overline{2\text{-}LGNN_I} \subsetneq \overline{2\text{-}FGNN_I}$ where the strictness of the last inclusion comes from (3). In other words, 2-FGNN$_I$ has a better power of approximation than the other architectures working with tensors of order 2. We already knew by Proposition 3 that 2-FGNN$_I$ is the best separating architecture among those studied in this paper, dealing with tensor of order 2 and our theorem implies that this is also the case for the approximation power.

To clarify the meaning of these statements, we explain why the inclusions "$\subseteq$" are actually straightforward. For concreteness, we focus on $\overline{k\text{-}FGNN_I} \subseteq \{f \in \mathcal{C}_I(K, \mathbb{F}) : \ \rho\,((k+1)\text{-}WL_I) \subseteq \rho\,(f)\}$. Take $h \in \overline{k\text{-}FGNN_I}$, this means that there is a sequence $\mathbf{GNN}_j \in k\text{-}FGNN_I$ such that, $\sup_{G \in K} \|h(G) - \mathbf{GNN}_j(G)\|$ goes to zero when $j$ goes to infinity.

Therefore, $h$ is continuous and constant on each $\rho\,(k\text{-}FGNN_I)$-class. Indeed, for any $(G, G') \in \rho\,(k\text{-}FGNN_I)$, $\mathbf{GNN}_j(G) = \mathbf{GNN}_j(G')$ so that $h(G) = \lim_i \mathbf{GNN}_j(G) = \lim_j \mathbf{GNN}_j(G') = h(G')$. Hence we have $\rho\,(k\text{-}FGNN_I) \subseteq \rho\,(h)$ and by Prop. 3, $\rho\,(k\text{-}FGNN_I) = \rho\,((k+1)\text{-}WL_I)$, allowing us to get the inclusion above.

On the contrary, the reverse inclusions "$\supset$" are much more intricate but they are also the most valuable. For instance, consider the inclusion $\overline{k\text{-}FGNN_I} \supset \{f \in \mathcal{C}_I(K, \mathbb{F}) : \ \rho\,((k+1)\text{-}WL_I) \subseteq \rho\,(f)\}$. If one wishes to learn a function $h \in \mathcal{C}_I(K, \mathbb{F})$ with $k\text{-}FGNN_I$ , this function must at least be approximable by the class of $k\text{-}FGNN_I$. Our theorem precisely guarantees that if $h$ is less separating that $k\text{-}WL_I$, it can be approximated by $k\text{-}FGNN_I$:

$$\forall \epsilon > 0, \ \exists \mathbf{GNN} \in k\text{-}FGNN_I, \ \sup_{G \in K} \|h(G) - \mathbf{GNN}(G)\| \leq \epsilon.$$

For this, we show a much more general version of the famous Stone-Weierstrass theorem (see Section D) which relates the separating power with the approximation power. Following the elegant idea of Maehara & Hoang (2019), we augment the input space to transform vector-valued equivariant functions into scalar invariant maps. Then, we apply a fine-grained approximation theorem from Timofte (2005). We also provide specialized versions of our abstract theorem in Section 5, which can be easily used to determine the approximation capabilities of any deep learning architecture.

Our theorem has also implications for universality results like Maron et al. (2019b); Keriven & Peyré (2019). A class of GNN is said to be universal if its closure on a compact set $K$ is the whole $\mathcal{C}_I(K, \mathbb{F})$ (or $\mathcal{C}_E(K, \mathbb{F}^n)$). In particular, Thm. 4 implies that $n$-LGNN and $n$-FGNN are universal as $n$-WL distinguishes non-isomorphic graphs of size $n$. This recovers a result of Ravanbakhsh (2020) for LGNN. Moreover, we can leverage the extensive literature on the WL tests to give more subtle results. For instance, Cai et al. (1989, §8.2) show that, for planar graphs, $O(\sqrt{n})$-WL can distinguish non-isomoprhic instances. Therefore, $O(\sqrt{n})$-LGNN or $O(\sqrt{n})$-FGNN achieve universality in the particular, yet common, case of planar graphs. On a more practical side, Fürer (2010, Thm. 4.5) shows that the spectrum of a graph is less separating than 3-WL so that functions of the spectrum can actually be well approximated by 2-FGNN.

## 5 EXPRESSIVENESS OF GNNS

We now state the general theorems which are our main tools in proving our approximation guarantees for GNNs. Theirs proofs are deferred to Section D.9 which contains our generalization of the Stone-Weierstrass theorem with symmetries. We need to first introduce more general definitions: If $G$ is a finite group acting on some topological space $X$, we say that $G$ acts continuously on $X$ if, for all $g \in G$, $x \mapsto g \cdot x$ is continuous. If $G$ is a finite group acting on some compact set $X$ and some topological space $Y$, we define the sets of equivariant and invariant continuous functions by,

$$\mathcal{C}_E(X, Y) = \{f \in \mathcal{C}(X, Y) : \forall x \in X, \ \forall g \in G, \ f(g \cdot x) = g \cdot f(x)\}$$
$$\mathcal{C}_I(X, Y) = \{f \in \mathcal{C}(X, Y) : \forall x \in X, \ \forall g \in G, \ f(g \cdot x) = f(x)\}$$

Note that these definitions extend Definition 1 to a general group.

**Theorem 5.** *Let $X$ be a compact space, $\mathbb{F} = \mathbb{R}^p$ be some finite-dimensional vector space, $G$ be a finite group acting (continuously) on $X$.*

*Let $\mathcal{F}_0 \subseteq \bigcup_{h=1}^{\infty} \mathcal{C}_I(X, \mathbb{R}^h)$ be a non-empty set of invariant functions, stable by concatenation, and consider,*

$$\mathcal{F} = \{m \circ f : f \in \mathcal{F}_0 \cap \mathcal{C}(X, \mathbb{R}^h), \ m : \mathbb{R}^h \to \mathbb{F} \ MLP, \ h \geq 1\} \subseteq \mathcal{C}(X, \mathbb{F}).$$

*Then the closure of $\mathcal{F}$ is,*

$$\overline{\mathcal{F}} = \{f \in \mathcal{C}_I(X, \mathbb{F}) : \rho(\mathcal{F}_0) \subseteq \rho(f)\}.$$

We can apply Theorem 5 to the class of $k$-ary relational pooling GNN introduced in Murphy et al. (2019a). As a result, we get that this class of invariant $k$-RP GNN can approximate any continuous function $f$ with $\rho(k - \text{RPGNN}) \subseteq \rho(f)$ but to the best of our knowledge, $\rho(k - \text{RPGNN})$ is not known and only $\rho(k - \text{RPGNN}) \subset \rho(2 - WL_I)$ is proved in Murphy et al. (2019a). We now state our general theorem for the equivariant case:

**Theorem 6.** *Let $X$ be a compact space, $\mathbb{F} = \mathbb{R}^p$ and $G = \mathcal{S}_n$ the permutation group, acting (continuously) on $X$ and acting on $\mathbb{F}^n$ by, for $\sigma \in \mathcal{S}_n$, $x \in \mathbb{F}^n$,*

$$\forall i \in \{1, \ldots, p\}, \ (\sigma \cdot x)_i = x_{\sigma^{-1}(i)},$$

*Let $\mathcal{F}_0 \subseteq \bigcup_{h=1}^{\infty} \mathcal{C}_E\left(X, (\mathbb{R}^h)^n\right)$ be a non-empty set of equivariant functions, stable by concatenation, and consider,*

$$\mathcal{F} = \{x \mapsto (m(f(x)_1), \ldots, m(f(x)_n)) : f \in \mathcal{F}_0 \cap \mathcal{C}\left(X, (\mathbb{R}^h)^n\right), \ m : \mathbb{R}^h \to \mathbb{F} \ MLP, \ h \geq 1\}$$

*Assume, that, if $f \in \mathcal{F}_0$, then,*

$$x \mapsto \left(\sum_{i=1}^{n} f(x)_i, \sum_{i=1}^{n} f(x)_i, \ldots, \sum_{i=1}^{n} f(x)_i\right) \in \mathcal{F}_0.$$

*Then the closure of $\mathcal{F}$ is,*

$$\overline{\mathcal{F}} = \{f \in \mathcal{C}_E(X, \mathbb{F}^n) : \rho(\mathcal{F}_0) \subseteq \rho(f)\}.$$

Applications of these theorems fo the case of Pointnet Qi et al. (2017) are provided in Section D.9

## 6 QUADRATIC ASSIGNMENT PROBLEM

To empirically evaluate our results, we study the Quadratic Assignment Problem (QAP), a classical problem in combinatorial optimization. For $A, B$ $n \times n$ symmetric matrices, it consists in solving

$$\text{maximize trace}(AXBX^{\top}), \quad \text{subject to } X \in \Pi,$$

where $\Pi$ is the set of $n \times n$ permutation matrices. Many optimization problems can be formulated as QAP. An example is the network alignment problem, which consists in finding the best matching

between two graphs, represented by their adjacency matrices $A$ and $B$. Though QAP is known to be NP-hard, recent works such as Nowak et al. (2018) have investigated whether it can be solved efficiently w.r.t. a fixed input distribution. More precisely, Nowak et al. (2018) studied whether one can learn to solve this problem using a MGNN trained on a dataset of already solved instances. However, as shown below, both the baselines and their approach fail on regular graphs, a class of graph considered as particularly hard for isomorphism testing.

To remedy this weakness, we consider 2-FGNN$_E$. We then follow the siamese method of (Nowak et al., 2018): given two graphs, our system produces an embedding in $\mathbb{F}^n$ for each graph, where $n$ is the number of nodes, which are then multiplied together to obtain a $n \times n$ similarity matrix on nodes. A permutation is finally computed by solving a Linear Assignment Problem (LAP) with this resulting $n \times n$ as cost matrix. We tested our architecture on two distribution: the Erdős–Rényi model and random regular graphs. The accuracy in matching the graphs is much improved compare to previous works. The experimental setup is described more precisely in Section A.1.

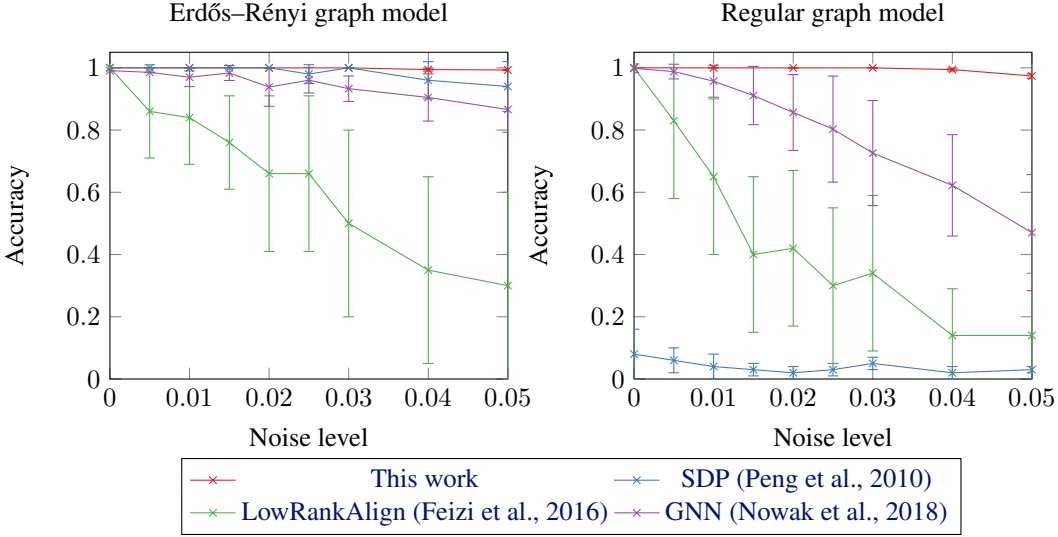

Figure 1: Fraction of matched nodes for pairs of correlated graphs (with edge density 0.2) as a function of the noise, see Section A.1 for details.

## 7 CONCLUSION

We derived the expressive power of various practical GNN architectures: message passing GNN, linear GNN and folklore GNN; both for their invariant and equivariant counterparts. Our results unify and extend the recent works in this direction. In particular, we are able to recover all the universality results proved for GNNs so far. Similarly to existing results in the literature, we do not deal here with the sizes of the embeddings constructed at different layers, i.e. the sizes of the spaces $\mathbb{F}_\ell$, and these sizes are supposed to grow to infinity with the number of nodes $n$ in the graph. Obtaining bounds on the scaling of the sizes of the features to ensure that the results presented here are still valid is an interesting open question. We show that folklore GNNs have the best power of approximation among all GNNs studied here dealing with tensors of order 2. From a practical perspective, we demonstrate their improved performance on the QAP with a significant gap in performances compared to other approaches.

## ACKNOWLEDGMENTS

This work was supported in part by the French government under management of Agence Nationale de la Recherche as part of the "Investissements d'avenir" program, reference ANR19-P3IA-0001 (PRAIRIE 3IA Institute). M.L. thanks Google for Google Cloud Platform research credits and NVIDIA for a NVIDIA GPU Grant.

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

## A    Experimental results

### A.1    Details on the experimental setup

We consider a 2-FGNN$_E$ and train it to solve random planted problem instances of the QAP. Given a pair of graphs $G_1, G_2$ with $n$ nodes each, we consider the siamese 2-FGNN$_E$ encoder producing embeddings $E_1, E_2 \in \mathbb{R}^{n \times k}$. Those embeddings are used to predict a matching as follows: we first compute the outer product $E_1 E_2^T$, then we take a softmax along each row and use standard cross-entropy loss to predict the corresponding permutation index. We used 2-FGNN$_E$ with 2 layers, each MLP having depth 3 and hidden states of size 64. We trained for 25 epochs with batches of size 32, a learning rate of 1e-4 and Adam optimizer. The PyTorch code is available in the supplementary material.

For each experiment, the dataset was made of 20000 graphs for the train set, 1000 for the validation set and 1000 for the test set. For the experiment with Erdős–Rényi random graphs, we consider $G_1$ to be a random Erdős–Rényi graph with edge density $p_e = 0.2$ and $n = 50$ vertices. The graph $G_2$ is a small perturbation of $G_1$ according to the following error model considered in Feizi et al. (2016):

$$G_2 = G_1 \odot (1 - Q) + (1 - G_1) \odot Q', \tag{7}$$

where $Q$ and $Q'$ are Erdős–Rényi random graphs with edge density $p_1$ and $p_2 = p_1 p_e / (1 - p_e)$ respectively, so that $G_2$ has the same expected degree as $G_1$. The noise level is the parameter $p_1$. For regular graphs, we followed the same experimental setup but now $G_1$ is a random regular graph with degree $d = 10$. Regular graphs are interesting example as they tend to be considered harder to align due to their more symmetric structure.

### A.2    Experimental results on graphs of varying size

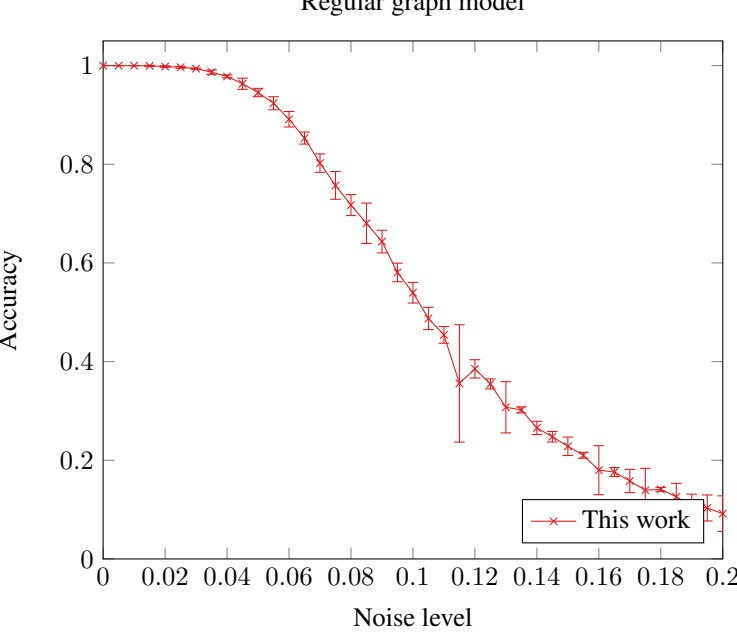

Figure 2: Fraction of matched nodes for pairs of correlated graphs (with edge density 0.2) as a function of the noise, see Section A.1 for details.

We tested our models on dataset of graphs of varying size, as this setting is also encompassed by our theory.

However, contrary to message-passing GNN, GNN based on tensors do not work well with batches of graphs of varying size. Previous implementations, such as the one of Maron et al. (2019a), group the graphs in the dataset by size, enabling the GNN to only deal with batches of graphs on the same size.

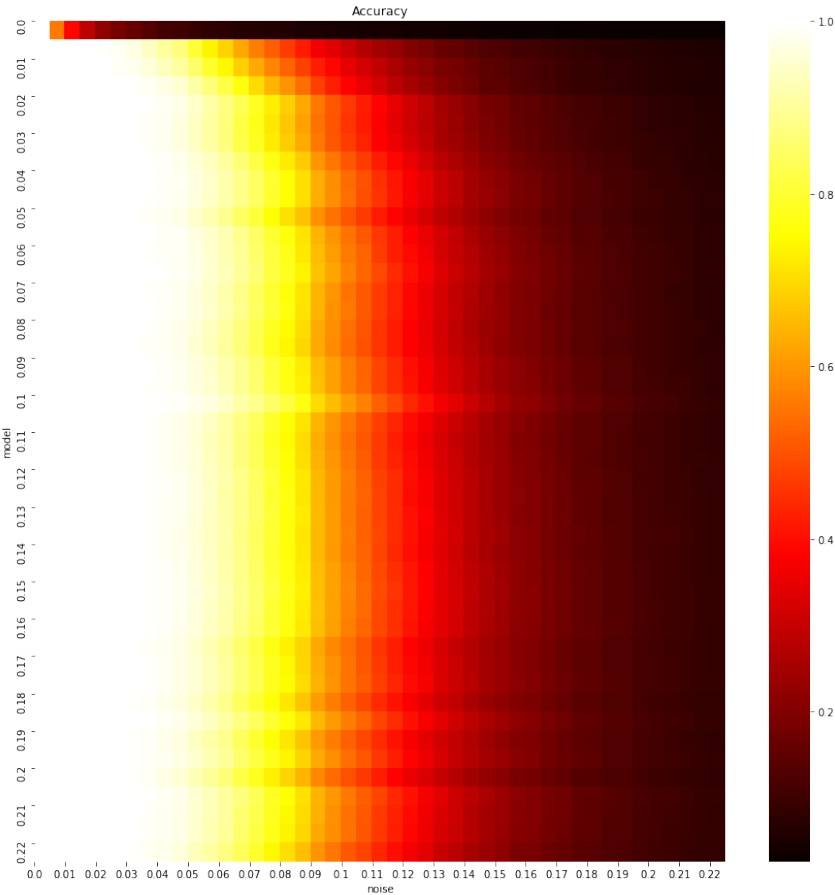

Figure 3: Each line corresponds to a model trained at a given noise level and shows its accuracy across all noise levels.

Instead, we use masking, which is a standard practice in recurrent neural networks. A batch of $b$ tensors of sizes $n_1 \times n_1, n_2 \times n_2, \ldots, n_b \times n_b$ is represented as a tensor $b \times n_{max} \times n_{max}$ where $n_{max} = \max_{i=1,\ldots,b} n_i$. A mask is created at initialization and is used to ensure that the operations on the full tensor translates to valid operations on each of the individual tensor.

We implemented this functionnality as a class `MaskedTensors`. Thanks to the newest improvements of PyTorch (Paszke et al., 2019), `MaskedTensors` act as a subclass of fundamental `Tensor` class. Thus they almost seamlessly integrate into standard PyTorch code. We refer the reader to the code for more details: https://github.com/mlelarge/graph_neural_net

Results of our architecture and implementation with graphs of varying size are shown below on Figure 2. The only difference with the setting described above is that the number of nodes is now random. The number of vertices of a graph is indeed chosen randomly according to binomial distribution of parameters $n = 50$ and $p_n = 0.9$.

## A.3 GENERALIZATION FOR REGULAR GRAPHS

We made the following experiment with the same general setting as in Section A.1 with regular graphs. We trained different models for all noise levels between 0 and 0.22 but in Figure 3, we plot the accuracy of each model across all noise levels. We observe that a majority of the models actually generalize to settings with noise level on which they were not trained. Indeed, the model trained with noise level $\approx 0.1$ is performing best among all models across all noise levels!

# B WEISFEILER-LEHMAN TESTS

Here we describe more precisely this hierarchy of tests, which will be used extensively to characterize the discirminatory power of classes of GNN. See Douglas (2011); Grohe (2017); Fürer (2017) for graph-theoretic introductions to these algorithms.

## B.1 WEISFEILER-LEHMAN TEST ON VERTICES

We now present the initial vertex coloring algorithm.

**Input.** This algorithm takes as input a discrete graph structure $G = (V, E) \in \mathcal{G}_n$ with $V = [n]$, $E \subseteq V^2$ and $h \in \mathbb{F}_0^n$ features on the vertices.

**Initialization.** Each vertex $s \in V$ is given a color $c_{\mathrm{WL}}^0(G, h)_s = h_s$ corresponding to its features vector.

**Refining the coloring.** The colors of the vertices are updated as follows,

$$c_{\mathrm{WL}}^{t+1}(G, h)_s = \mathrm{Lex}\left(c_{\mathrm{WL}}^t(G, h)_s, \left\{\left\{c_{\mathrm{WL}}^t(G, h)(\tilde{s}) : \tilde{s} \sim s\right\}\right\}\right),$$

and the function Lex means that all occuring colors are lexicographically ordered.

For each graph $G \in \mathcal{G}_n$ and each vector of features $h \in \mathbb{F}_0^n$, there exists a time $T(G, h)$ from which the sequence of colorings $(c_{\mathrm{WL}}^t(G, h))_{t \geq 0}$ is stationary. More exactly, the colorings are not refined anymore: for any $t \geq T(G, h)$, $s, s' \in V$,

$$c_{\mathrm{WL}}^t(G, h)_s = c_{\mathrm{WL}}^t(G, h)_{s'} \iff c_{\mathrm{WL}}^{T(G,h)}(G, h)_s = c_{\mathrm{WL}}^{T(G,h)}(G, h)_{s'}.$$

Denote the resulting coloring $c_{\mathrm{WL}}^{T(G,h)}(G, h)$ by simply $c_{\mathrm{WL}}(G, h)$. $c_{\mathrm{WL}}$ is now a mapping from $\mathcal{G}_n \times \mathbb{F}_0^n \to Z^n$ for some space of colors $Z$.

**Invariant tests** The proper Weisfeiler-Lehman test is invariant and is defined by, for $t \geq 0$ and $(G, h) \in \mathcal{G}_n \times \mathbb{F}_0^n$ a graph,

$$\mathrm{WL}_I^t(G) = \left\{\left\{c_{\mathrm{WL}}^t(G)_s : s \in V\right\}\right\}$$
$$\mathrm{WL}_I(G) = \left\{\left\{c_{\mathrm{WL}}(G)_s : s \in V\right\}\right\}$$

**Equivariant tests** For the vertex coloring algorithm, $c_{\mathrm{WL}}$ is already an equiavriant mapping so we define, for $t \geq 0$ and $(G, h) \in \mathcal{G}_n \times \mathbb{F}_0^n$ a graph,

$$\mathrm{WL}_E^t = c_{\mathrm{WL}}^t$$
$$\mathrm{WL}_E = c_{\mathrm{WL}}$$

## B.2 ISOMORPHISM TYPE

The initialization of the higher-order variants of the Weisfeiler-Lehman test is slightly more intricate. For this we need to define the isomorphism type of a $k$-tuple w.r.t. a graph described by a tensor $\mathbb{F}_0^{n^2}$.

A $k$-tuple $(i_1, \ldots, i_k) \in [n]^k$ in a graph $G \in \mathbb{F}_0^{n^2}$ and a $k$-tuple $(j_1, \ldots, j_k) \in [n]^k$ in a graph $H \in \mathbb{F}_0^{n^2}$ are said to have the same isomoprhism type if the mapping $i_w \mapsto j_w$ is a well-defined partial isormophism. Explicitly, this means that,

- $\forall w, w' \in [k], i_w = i_{w'} \iff j_w = j_{w'}$.
- $\forall w, w' \in [k], G_{i_w, i_{w'}} = H_{j_w, j_{w'}}$.

Denote by $\mathrm{iso}(G)_{i_1, \ldots, i_k}$ the isomorphism type of the $k$-tuple $(i_1, \ldots, i_k) \in [n]^k$ in a graph $G \in \mathbb{F}_0^{n^2}$.

### B.3 WEISFEILER-LEHMAN AND FOLKLORE WEISFEILER-LEHMAN TESTS OF ORDER $k \geq 2$

We now present the folklore version of the Weisfeiler-Lehman test of order $k$ ($k$-FWL), for $k \geq 2$, along with $k$-WL for clarity. For both, we follow the presentation of Maron et al. (2019a) (except for the equivariant tests).

**Input.** These algorithms take as input a graph $G \in \mathbb{F}_0^{n^2}$ which can be seen as a coloring on the pair of nodes.

**Initialization.** Each $k$-tuple $s \in V^k$ is given a color $c_{k\text{-WL}}^0(G)_s = c_{k\text{-FWL}}^0(G)_s$ corresponding to its isomorphism type.

**$k$-WL.** The $k$-WL test relies on the following notion of neighborhood, defined by, for any $w \in [k]$, and $s = (i_1, \ldots, i_k) \in V^k$,

$$N_w(s) = \{(i_1, \ldots, i_{w-1}, j, i_{w+1}, \ldots, i_k) : j \in V\}. \tag{WL}$$

Then, the colors of the $k$-tuples $s \in V^k$ are refined as follows,

$$c_{\text{WL}}^{t+1}(G)_s = \text{Lex}\left(c_{k\text{-WL}}^t(G)_s, (C_1^t(s), \ldots, C_k^t(s))\right).$$

where, for $w \in [k]$,

$$C_w^t(s) = \{\{c_{k\text{-WL}}^t(G)_{\tilde{s}} : \tilde{s} \in N_w(s)\}\}.$$

For each graph $G \in \mathbb{F}_0^n$, there exists a time $T(G)$ from which the sequence of colorings $(c_{k\text{-WL}}^t(G))_{t \geq 0}$ is stationary. More exactly, the colorings are not refined anymore: for any $t \geq T(G)$, $s, s' \in V^k$,

$$c_{k\text{-WL}}^t(G)_s = c_{k\text{-WL}}^t(G)_{s'} \iff c_{k\text{-WL}}^{T(G)}(G)_s = c_{k\text{-WL}}^{T(G)}(G)_{s'}.$$

Denote the resulting coloring $c_{k\text{-WL}}^{T(G)}(G)$ by simply $c_{k\text{-WL}}(G)$.

**$k$-FWL.** For $k$-FWL, the corresponding notion of neighborhood is defined by, for any $j \in V$, and $s = (i_1, \ldots, i_k) \in V^k$,

$$N_j^F(s) = \{(j, i_2, \ldots, i_k), (i_1, j, i_3, \ldots, i_k), \ldots, (i_1, i_2, \ldots, i_{k-1}, j)\} \tag{FWL}$$

Then, the colors of the $k$-tuples $s \in V^k$ are refined as follows,

$$c_{k\text{-WL}}^{t+1}(G)_s = \text{Lex}\left(c_{k\text{-FWL}}^t(G)_s, \{\{C_j^t(s) : j \in V\}\}\right),$$

where, for $j \in V$,

$$C_j^t(s) = \left(c_{k\text{-FWL}}^t(G)_{\tilde{s}} : \tilde{s} \in N_j^F(s)\right).$$

Like $k$-WL, for each graph $G \in \mathbb{F}_0^n$, there exists a time $T(G)$ from which the sequence of colorings $(c_{k\text{-WL}}^t(G))_{t \geq 0}$ is stationary. Similarly, denote the resulting coloring $c_{k\text{-WL}}^{T(G)}(G)$ by simply $c_{k\text{-WL}}(G)$.

The colors $c_{k\text{-FWL}}^t$ and $c_{k\text{-FWL}}^t$ at iteration $t$ define a mapping from $\mathbb{F}_0^{n^2}$ to space of colorings of $k$-tuples, $Z^{n^k}$ for some space $Z$.

**Invariant tests** The standard versions of the Weisfeiler-Lehman tests are invariant and can be defined by, for $t \geq 0$ and $G \in \mathbb{F}_0^{n^2}$ a graph,

$$k\text{-WL}_I^t(G) = \{\{c_{k\text{-WL}}^t(G)_s : s \in V^k\}\}$$
$$k\text{-WL}_I(G) = \{\{c_{k\text{-WL}}(G)_s : s \in V^k\}\}$$
$$k\text{-FWL}_I^t(G) = \{\{c_{k\text{-FWL}}^t(G)_s : s \in V^k\}\}$$
$$k\text{-FWL}_I(G) = \{\{c_{k\text{-FWL}}(G)_s : s \in V^k\}\}.$$

**Equivariant tests** We now introduce the equivariant version of these tests. Many extensions are possible, we chose this one for its simplicity. For $t \geq 0$, $G \in \mathbb{F}_0^{n^2}$ a graph, $i \in V$,

$$k\text{-WL}_E^t(G)_i = \{\{c_{k\text{-WL}}^t(G)_s : s \in V^k, s_1 = i\}\}$$
$$k\text{-WL}_E(G)_i = \{\{c_{k\text{-WL}}(G)_s : s \in V^k, s_1 = i\}\}$$
$$k\text{-FWL}_E^t(G)_i = \{\{c_{k\text{-FWL}}^t(G)_s : s \in V^k, s_1 = i\}\}$$
$$k\text{-FWL}_E(G)_i = \{\{c_{k\text{-FWL}}(G)_s : s \in V^k, s_1 = i\}\} .$$

# C   SEPARATING POWER OF GNN

The goal of this section is to prove,

**Proposition 3.** *We have, for $k \geq 2$,*

$$\rho\left(MGNN_I\right) = \rho\left(2\text{-}WL_I\right) \qquad\qquad \rho\left(MGNN_E\right) = \rho\left(2\text{-}WL_E\right) \qquad (4)$$
$$\rho\left(k\text{-}LGNN_I\right) = \rho\left(k\text{-}WL_I\right) \qquad\qquad \rho\left(k\text{-}LGNN_E\right) \subseteq \rho\left(k\text{-}WL_E\right) \qquad (5)$$
$$\rho\left(k\text{-}FGNN_I\right) = \rho\left((k+1)\text{-}WL_I\right) \qquad\qquad \rho\left(k\text{-}FGNN_E\right) = \rho\left((k+1)\text{-}WL_E\right) \qquad (6)$$

## C.1   MUTLI-LINEAR PERCEPTRONS

In the following we will use extensively multi-linear perceptrons (MLP) and their universality properties. Yet, for the sake of simplicity, we do not define precisely what we mean by MLP.

Given two finite-dimensional feature spaces $\mathbb{F}_0$ and $\mathbb{F}_1$, we only assume we are given a class of (continuous) MLP from $\mathbb{F}_0$ to $\mathbb{F}_1$ which is large enough to be dense in $\mathcal{C}(\mathbb{F}_0, \mathbb{F}_1)$. See for instance Hornik et al. (1989); Cybenko (1989) for precise conditions for MLP to be universal.

## C.2   AUGMENTED SEPARATING POWER

To factor the proof, we introduce another notion of separating power.

**Definition 7.** For $n, m \geq 1$ fixed, let $X$ be a set, $\mathbb{F}$ be a some space and $\mathcal{F}$ be a set of functions from $X$ to $Y = \mathbb{F}^{n \times m}$. Then, the augmented separating power of $\mathcal{F}$ is,

$$\rho_{n,m}^{augm}\left(\mathcal{F}\right) = \rho\left(\{(x, i, j) \in X \times \{1, \ldots, n\} \times \{1, \ldots, m\} \mapsto f(x)_{i,j} : f \in \mathcal{F}\}\right) .$$

Explicitly, for $x, y \in X$, $i, j \in \{1, \ldots, n\}$,

$$(x, i, j, y, k, l) \in \rho_{n,m}^{augm}\left(\mathcal{F}\right) \iff \forall f \in \mathcal{F}, \ f(x)_{i,j} = f(y)_{k,l} .$$

Note that when $n = m = 1$, the augmented separating power is exactly the same as the original separating power, so we identify $\rho\left(.\right)$ with $\rho_{1,1}^{augm}\left(.\right)$. We also identify $\rho_{n,1}^{augm}\left(.\right)$ with $\rho_{1,n}^{augm}\left(.\right)$, that we denote by $\rho_n^{augm}\left(.\right)$.

First, it is easy to see that this notion is more precise than the separating power.

**Lemma 8.** *If $\mathcal{F}$ and $\mathcal{G}$ are set of functions from $X$ to $\mathbb{F}^{n \times m}$,*

$$\rho_{n,m}^{augm}\left(\mathcal{F}\right) \subseteq \rho_{n,m}^{augm}\left(\mathcal{G}\right) \implies \rho\left(\mathcal{F}\right) \subseteq \rho\left(\mathcal{G}\right) ,$$

*and, in particular,*

$$\rho_{n,m}^{augm}\left(\mathcal{F}\right) = \rho_{n,m}^{augm}\left(\mathcal{G}\right) \implies \rho\left(\mathcal{F}\right) = \rho\left(\mathcal{G}\right) ,$$

The interest in this notion is justified by the following lemma, which shows that this notion behaves well under composition with "reduction layers".

**Lemma 9.** *For $n, m \geq 1$ fixed, let $X$ be a compact topological space, $Y = \mathbb{F}^{n \times m}$, $\mathbb{F}$ some finite-dimensional space, $\mathcal{F} \subseteq \mathcal{C}(X, Y)$, and $\tau : X \to Z^{n \times m}$ a function for some space $Z$. Define $\widetilde{\mathcal{F}} \subseteq \mathcal{C}(X, \mathbb{F}^n)$ by,*

$$\widetilde{\mathcal{F}} = \left\{ x \in X \longmapsto \left( \sum_{j=1}^m h(f(x)_{1,j}), \sum_{j=1}^m h(f(x)_{2,j}), \ldots, \sum_{i=1}^m h(f(x)_{n,j}) \right) : h : \mathbb{F} \to \mathbb{F} \ MLP, \ f \in \mathcal{F} \right\} .$$

*and $\widetilde{\tau} : X \to \widetilde{Z}^n$ by, for $x \in X$,*

$$\forall i \in \{1, \ldots, n\}, \ \widetilde{\tau}(x)_i = \{\{\tau(x)_{i,j} : 1 \leq j \leq m\}\} \ .$$

*Then,*

$$\rho_{n,m}^{augm}(\mathcal{F}) \subseteq \rho_{n,m}^{augm}(\tau) \implies \rho_{n,m}^{augm}\left(\widetilde{\mathcal{F}}\right) \subseteq \rho_{n,m}^{augm}(\widetilde{\tau})$$

$$\rho_{n,m}^{augm}(\mathcal{F}) \supset \rho_{n,m}^{augm}(\tau) \implies \rho_{n,m}^{augm}\left(\widetilde{\mathcal{F}}\right) \supset \rho_{n,m}^{augm}(\widetilde{\tau}) \ .$$

Note that, in the statement, we implicitly see $\widetilde{\mathcal{F}}$ as a set of functions from $X$ to $\mathbb{F}^{n \times 1}$ to fit the definition of augmented separating power.

*Proof.* We show the two inclusions independently.

($\subseteq$) We first show that,

$$\rho_{n,m}^{augm}(\mathcal{F}) \subseteq \rho_{n,m}^{augm}(\tau) \implies \rho_{n,m}^{augm}\left(\widetilde{\mathcal{F}}\right) \subseteq \rho_{n,m}^{augm}(\widetilde{\tau})$$

Take $(x, i, y, k) \in \rho_{n,m}^{augm}\left(\widetilde{\mathcal{F}}\right)$. This means that, for any $h : \mathbb{F} \to \mathbb{F}$, $f \in \mathcal{F}$,

$$\sum_{j=1}^{m} h(f(x)_{i,j}) = \sum_{j=1}^{m} h(f(y)_{k,j}) \ .$$

By Lem. 31 and the universality of MLP, there exists a permutation $\sigma \in \mathcal{S}_m$ such that $(f(x)_{i,\sigma(1)}, \ldots, f(x)_{i,\sigma(m)}) = (f(y)_{k,1}, \ldots, f(y)_{k,m})$. By definition of the augmented separating power, this means that, for any $j \in \{1, \ldots, m\}$, $(x, i, \sigma(j), y, k, j) \in \rho_{n,m}^{augm}(\mathcal{F})$. Hence, by assumption, for any $j \in \{1, \ldots, m\}$, $(x, i, \sigma(j), y, k, j) \in \rho_{n,m}^{augm}(\tau)$, i.e. $\tau(x)_{i,\sigma(j)} = \tau(y)_{k,j}$. But this exactly means that $\widetilde{\tau}(x)_i = \widetilde{\tau}(y)_k$ so that $(x, i, y, k) \in \rho_{n,m}^{augm}(\widetilde{\tau})$ as required.

($\supset$) We now show the other inclusion,

$$\rho_{n,m}^{augm}(\mathcal{F}) \supset \rho_{n,m}^{augm}(\tau) \implies \rho_{n,m}^{augm}\left(\widetilde{\mathcal{F}}\right) \supset \rho_{n,m}^{augm}(\widetilde{\tau}) \ .$$

Take $(x, i, y, k) \in \rho_{n,m}^{augm}(\widetilde{\tau})$. By definition of $\widetilde{\tau}$, this means that there exists $\sigma \in \mathcal{S}_m$ such that $\tau(x)_{i,\sigma(j)} = \tau(y)_{k,j}$ so that $(x, i, \sigma(j), y, k, j) \in \rho_{n,m}^{augm}(\tau) \subseteq \rho_{n,m}^{augm}(\mathcal{F})$. Hence, for any $f \in \mathcal{F}$, $(f(x)_{i,\sigma(1)}, \ldots, f(x)_{i,\sigma(m)}) = (f(y)_{k,1}, \ldots, f(y)_{k,m})$, and so, for any $h : \mathbb{F} \to \mathbb{F}$,

$$\sum_{j=1}^{m} h(f(x)_{i,j}) = \sum_{j=1}^{m} h(f(y)_{k,j}) \ .$$

Therefore, $(x, i, y, k) \in \rho_{n,m}^{augm}\left(\widetilde{\mathcal{F}}\right)$, which concludes the proof.

$\square$

## C.3 INITIALIZATION LAYER

We use the same initialization layer as Maron et al. (2019a); Chen et al. (2020) and recall it below. The initial graph is a tensor of the form $G \in \mathbb{F}_0^{n^2}$ with $\mathbb{F}_0 = \mathbb{R}^{e+2}$; the last channel of $G_{:,:,e+1}$ encodes the adjacency matrix of the graph and the first $e$ channels $G_{:,:,1:e}$ are zero outside the diagonal and $G_{i,i,1:e} \in \mathbb{R}^e$ is the color of vertex $v_i \in V$. We then define $I^k : \mathbb{F}_0^{n^2} \to \mathbb{F}_1^{n^k}$ with $\mathbb{F}_1 = \mathbb{R}^{k^2 \times (e+2)}$ as follows:

$$I^k(G)_{\mathbf{i},r,s,w} = G_{i_r,i_s,w}, \ w \in [e+1],$$
$$I^k(G)_{\mathbf{i},r,s,e+2} = \mathbf{1}(i_r = i_s),$$

for $\mathbf{i} \in [n]^k$ and $r, s \in [k]$. This linear equivariant layer has the same separating power as the isormorphism type, which is defined in Section B.2.

**Lemma 10** ((Maron et al., 2019a, C.1)). *For $k \geq 2$, $p_0 \geq 1$, $\mathbb{F}_0 = \mathbb{R}^{p_0}$, $\mathbb{F}_1 = \mathbb{R}^{k^2 \times (p_0 + 1)}$, there exists $I^k : \mathbb{F}_0^{n^2} \to \mathbb{F}_1^{n^k}$ such that,*

$$\rho_{n,n^{k-1}}^{augm} \left( I^k \right) = \rho_{n,n^{k-1}}^{augm} (\text{iso}) .$$

## C.4 KNOWN RESULTS ABOUT THE SEPARATING POWER OF SOME GNN CLASSES

First, we need to define some classes of GNN from which both the invariant and equivaraint GNN we considered are built. See Section 3 for details about the different layers.

$$\text{MGNN}_{emb} = \{F_T \circ \dots F_2 \circ F_1 : F_t : \mathbb{F}_t^n \to \mathbb{F}_{t+1}^n \text{ message passing layer}, t = 1, \dots, T, T \geq 1\}$$

$$k\text{-LGNN}_{emb} = \{F_T \circ \dots F_2 \circ F_1 \circ I^k : \mathbb{F}_t^n \to \mathbb{F}_{t+1}^n \text{ linear equivariant layer}, t = 1, \dots, T, T \geq 1\}$$

$$k\text{-FGNN}_{emb} = \{F_T \circ \dots F_2 \circ F_1 \circ I^k : \mathbb{F}_t^n \to \mathbb{F}_{t+1}^n \text{ FGL}, t = 1, \dots, T, T \geq 1\} .$$

Then, the precise results from the literature can be rephrased as,

**Lemma 11** (Xu et al. (2018),Maron et al. (2018),Maron et al. (2019a)). *For $k \geq 2$,*

$$\rho_n^{augm} (\text{MGNN}_{emb}) = \rho_n^{augm} (c_{WL}) \tag{8}$$

$$\rho_{n,n^{k-1}}^{augm} (k\text{-LGNN}_{emb}) \subseteq \rho_{n,n^{k-1}}^{augm} (c_{k\text{-WL}}) \tag{9}$$

$$\rho_{n,n^{k-1}}^{augm} (k\text{-FGNN}_{emb}) \subseteq \rho_{n,n^{k-1}}^{augm} (c_{k\text{-FWL}}) \tag{10}$$

(8) comes from Xu et al. (2018, §A, §B), (9) and (10) from Maron et al. (2019a, §C, §D).

## C.5 BOUNDING THE SEPARATING POWER OF $k$-FGNN

We complete the results of the literature with a bound on the separating power of $k$-FGNN. Note that the particular case of $k = 2$ is already proven in Geerts (2020a).

**Lemma 12.** *For any $k \geq 2$,*

$$\rho_{n,n^{k-1}}^{augm} (k\text{-FGNN}_{emb}) \supset \rho_{n,n^{k-1}}^{augm} (c_{k\text{-FWL}}) ,$$

*so that*

$$\rho_{n,n^{k-1}}^{augm} (k\text{-FGNN}_{emb}) = \rho_{n,n^{k-1}}^{augm} (c_{k\text{-FWL}}) .$$

*Proof.* Define,

$$k\text{-FGNN}_{emb}^T = \{F_T \circ \dots F_2 \circ F_1 \circ I^k : \mathbb{F}_t^n \to \mathbb{F}_{t+1}^n \text{ FGL}, t = 1, \dots, T\} ,$$

the set of functions defined by exactly $T$ FGL layers. We show by induction that, for any $T \geq 0$,

$$\rho_{n,n^{k-1}}^{augm} \left( k\text{-FGNN}_{emb}^T \right) \supset \rho_{n,n^{k-1}}^{augm} \left( c_{k\text{-FWL}}^T \right) .$$

For $T = 0$, this is immediate by the definition of $I^k$ in Section C.3.

Assume now that this inclusion holds at $T - 1 \geq 0$. We show that it also holds at $T$. Take $G, G' \in \mathbb{F}_0^{n^k}$ and $s, s' \in [n]^k$ such that,

$$c_{k\text{-FWL}}^T (G)_s = c_{k\text{-FWL}}^T (G')_{s'} .$$

We need to show that, for any $f \in k\text{-FGNN}_{emb}^T$,

$$f(G)_s = f(G')_{s'} .$$

But, by definition of the update rule $k$-FWL, the equality of the colors of $s$ and $s'$ above implies that,

$$c_{k\text{-FWL}}^{T-1} (G)_s = c_{k\text{-FWL}}^{T-1} (G')_{s'} , \tag{11}$$

and that there exists $\sigma \in \mathcal{S}_n$ such that, for any $j \in [n]$,

$$\left( c_{k\text{-FWL}}^{T-1} (G)_{\tilde{s}} : \tilde{s} \in N_j^F(s) \right) = \left( c_{k\text{-FWL}}^{T-1} (G')_{\tilde{s}} : \tilde{s} \in N_{\sigma(j)}^F(s') \right) .$$

Let $s = (i_1, \ldots, i_k)$ and $s' = (j_1, \ldots, j_k)$. Then this implies that, for any $w \in [k]$, $j \in [n]$,

$$c_{k\text{-FWL}}^{T-1}(G)_{i_1,\ldots,i_{w-1},j,i_{w+1},\ldots,i_k} = c_{k\text{-FWL}}^{T-1}(G')_{j_1,\ldots,j_{w-1},\sigma(j),j_{w+1},\ldots,j_k} . \tag{12}$$

We now use the induction hypothesis, i.e. that,

$$\rho_{n,n^{k-1}}^{augm}\left(k\text{-FGNN}_{emb}^{T-1}\right) \supset \rho_{n,n^{k-1}}^{augm}\left(c_{k\text{-FWL}}^{T-1}\right) .$$

Take any $f_{T-1} \in k\text{-FGNN}_{emb}^{T-1}$. By (11),

$$f_{T-1}(G)_s = f_{T-1}(G')_{s'} .$$

By (12), for any $w \in [k]$, $j \in [n]$,

$$f_{T-1}(G)_{i_1,\ldots,i_{w-1},j,i_{w+1},\ldots,i_k} = f^{T-1}(G')_{j_1,\ldots,j_{w-1},\sigma(j),j_{w+1},\ldots,j_k} .$$

By the definition of FGL Section 3.3, for any $F_T : \mathbb{F}_T^{n^k} \to \mathbb{F}_{T+1}^{n^k}$ FGL, $F_T \circ f_{T-1}(G)_s = F_T \circ f_{T-1}(G')_{s'}$ . Therefore, for any $f \in k\text{-FGNN}_{emb}^T$,

$$f(G)_s = f(G')_{s'} ,$$

which concludes the proof. $\qquad\square$

## C.6 Conclusion

**Proposition 13.** *We have, for $k \geq 2$,*

$$\rho\left(MGNN_I\right) = \rho\left(2\text{-}WL_I\right) \qquad\qquad \rho\left(MGNN_E\right) = \rho\left(2\text{-}WL_E\right) \tag{13}$$
$$\rho\left(2\text{-}LGNN_I\right) = \rho\left(2\text{-}WL_I\right) \tag{14}$$
$$\rho\left(k\text{-}LGNN_I\right) \subseteq \rho\left(k\text{-}WL_I\right) \qquad\qquad \rho\left(k\text{-}LGNN_E\right) \subseteq \rho\left(k\text{-}WL_E\right) \tag{15}$$
$$\rho\left(k\text{-}FGNN_I\right) = \rho\left(k\text{-}FWL_I\right) \qquad\qquad \rho\left(k\text{-}FGNN_E\right) = \rho\left(k\text{-}FWL_E\right) \tag{16}$$

*Proof.* Most of the statements come from the literature or are direct consequences of the lemmas above.

- Proof of (13). The invariant case is proven in Xu et al. (2018, Lem. 2, Thm. 3). The equivariant case comes from Lem. 11, Lem. 8, the fact that the layers $m_E \circ (\text{Id} + \lambda S^1)$ does not change the separating power and recalling that simply $\text{WL}_E = c_{WL}$.

- (14) is the exact result Chen et al. (2020, Thm. 6).

- Proof of (15). The invariant case is exactly Maron et al. (2019a, Thm. 1). The equivariant case comes from Lem. 11, Lem. 9, Lem. 8 and the fact that the layer $m_E$ does not change the separating power.

- Proof of (16). The direct inclusion of the invariant case corresponds to Maron et al. (2019a, Thm. 2). The other cases are a consequence of Lem. 11, Lem. 9, Lem. 8 and the fact that the layers $m_I$ and $m_E$ does not change the separating power.

$\qquad\square$

## D Stone-Weierstrass theorem with symmetries

This section presents our extension of the Stone-Weierstrass theorem dealing with functions with symmetries. The scope of this section is not restricted to graphs or even tensors and we will deal with general spaces and general symmetries. To illustrate it, we will present applications for the PointNet architecture Qi et al. (2017). Our approximation results for GNNs (Theorems Thm. 5 and Thm. 6) are then obtained from these theoretical results applied to tensors and the symmetric group in Section D.9

## D.1 GENERAL NOTATIONS

As explained above, we are dealing in this section with a much larger scope than graphs and permutations. We first need to extend the notations introduced above. The notations introduced below will make this section self-contained.

If $X$ is some topological space, and $F \subseteq X$, denote by $\overline{F}$ its closure.

If $X$ is a topological space and $Y = \mathbb{R}^p$ some finite-dimensional space, denote by $\mathcal{C}(X, Y)$ the set of continuous functions from $X$ to $Y$.

Moreover, if $X$ is compact, we endow $\mathcal{C}(X, Y)$ with the topology of uniform convergence, which is defined by the norm, $f \mapsto \sup_{x \in X} \|f(x)\|$ for some norm $\|.\|$ on $Y$.

If $G$ is a finite group acting on some topological space $X$, we say that $G$ acts continuously on $X$ if, for all $g \in G$, $x \mapsto g \cdot x$ is continuous.

If $G$ is a finite group acting on some compact set $X$ and some topological space $Y$, we define the sets of equivariant and invariant continuous functions by,

$$\mathcal{C}_E(X, Y) = \{f \in \mathcal{C}(X, Y) : \forall x \in X, \ \forall g \in G, \ f(g \cdot x) = g \cdot f(x)\}$$
$$\mathcal{C}_I(X, Y) = \{f \in \mathcal{C}(X, Y) : \forall x \in X, \ \forall g \in G, \ f(g \cdot x) = f(x)\}$$

Note that these definitions extend Definition 1 to a general group.

If $Y = \mathbb{R}^p$, we denote the coordinate-wise multiplication, or Hadamard product of $y, y' \in Y$ simply by $yy' = (y_1 y_1', \ldots, y_p y_p') \in Y$. We say that a subset $A \subseteq \mathbb{R}^p$ is a subalgebra of $\mathbb{R}^p$ if it is both a linear space and stable by multiplication.

This product in turn defines a product on $\mathcal{C}(X, Y)$ with $Y = \mathbb{R}^p$ by, for $f, g \in \mathcal{C}(X, Y)$, $fg : x \mapsto f(x)g(x)$.

In addition, we also extend the scalar-vector product of $Y = \mathbb{R}^p$ to functions: if $g \in \mathcal{C}(X, \mathbb{R})$ and $f \in \mathcal{C}(X, Y)$, their product $gf$ is the function $gf : x \mapsto g(x)f(x)$. Given a set of scalar functions $\mathcal{S} \subseteq \mathcal{C}(X, \mathbb{R})$ and a set of vector-valued functions $\mathcal{F} \subseteq \mathcal{C}(X, Y)$, the set of products of functions of these two sets will be denoted by,

$$\mathcal{S} \cdot \mathcal{F} = \{gf : g \in \mathcal{S}, \ f \in \mathcal{F}\}.$$

Moreover, we denote by $\mathbf{1}$ the continuous function from some $X$ to $\mathbb{R}^p$ defined by $x \mapsto (1, \ldots, 1)$. In particular, if $f$ is a function from $X$ to $\mathbb{R}$, $f\mathbf{1}$ denotes the function $x \mapsto (f(x), \ldots, f(x))$ which goes from $X$ to $Y = \mathbb{R}^p$. Finally, we say that $\mathcal{F} \subseteq \mathcal{C}(X, Y)$ with $Y$ being some $\mathbb{R}^p$ is a subalgebra if it is a linear space which is also stable by multiplication.

## D.2 SEPARATING POWER

We recall the definition of separating power that we introduced above:

**Definition 14.** Let $\mathcal{F}$ be a set of functions $f$ defined on a set $X$, where each $f$ takes its values in some $Y_f$. The equivalence relation $\rho(\mathcal{F})$ defined by $\mathcal{F}$ on $X$ is: for any $x, x' \in X$,

$$(x, x') \in \rho(\mathcal{F}) \iff \forall f \in \mathcal{F}, \ f(x) = f(x').$$

For a function $f$, we write $\rho(f)$ for $\rho(\{f\})$.

Separating power is stable by closure:

**Lemma 15.** *Let $X$ be a compact topological space, $Y$ be some finite-dimensional and $\mathcal{F} \subseteq \mathcal{C}(X, Y)$. Then,*

$$\rho(\mathcal{F}) = \rho(\overline{\mathcal{F}})$$

*Proof.* As $\mathcal{F} \subseteq \overline{\mathcal{F}}$, $\overline{\mathcal{F}}$ is more separating than $\mathcal{F}$, i.e. $\rho(\overline{\mathcal{F}}) \subseteq \rho(\mathcal{F})$.

Conversely, take $(x, y) \notin \rho(\overline{\mathcal{F}})$. By definition, there exists $h \in \overline{\mathcal{F}}$ such that $h(x) \neq h(y)$ so that if $\epsilon = \|h(x) - h(y)\|$, $\epsilon > 0$ (for some norm $\|.\|$ on $Y$). As $h \in \overline{\mathcal{F}}$, there is some $f \in \mathcal{F}$ such that

$\sup_X \|h - f\| \leq \frac{\epsilon}{3}$. Therefore, by the triangular inequality,

$$\epsilon = \|h(x) - h(y)\| \leq \|f(x) - h(x)\| + \|f(x) - f(y)\| + \|f(y) - h(y)\| \leq \frac{2}{3}\epsilon + \|f(x) - f(y)\|.$$

It follows that $\|f(x) - f(y)\| \geq \frac{\epsilon}{3} > 0$ so that $f(x) \neq f(y)$ and $(x, y) \notin \rho(\mathcal{F})$. □

### D.3   APPROXIMATION THEOREMS FOR REAL-VALUED FUNCTIONS

We start by recalling Stone-Weierstrass theorem, see Rudin (1991, Thm. 5.7).

**Theorem 16** (Stone-Weierstrass). *Let $X$ be a compact space, and $\mathcal{F}$ be a subalgebra of $\mathcal{C}(X, \mathbb{R})$ the space of real-valued continuous functions of $X$, which contains the constant function $\mathbf{1}$. If $\mathcal{F}$ separates points, i.e. $\rho(\mathcal{F}) = \{(x, x) : x \in X\}$, then $\mathcal{F}$ is dense in $\mathcal{C}(X, \mathbb{R})$.*

We now prove an extension of this classical result due to Timofte (2005) allowing us to deal with much smaller $\mathcal{F}$ by dropping the requirement that $\mathcal{F}$ separates points.

**Corollary 17.** *Let $X$ be a compact space, and $\mathcal{F}$ be a subalgebra of $\mathcal{C}(X, \mathbb{R})$ the space of real-valued continuous functions of $X$, which contains the constant function $\mathbf{1}$. Then,*

$$\overline{\mathcal{F}} = \{f \in \mathcal{C}(X, \mathbb{R}) : \rho(\mathcal{F}) \subseteq \rho(f)\}.$$

Note that if $\mathcal{F}$ separates points, we get back the classical result as every function satisfies $\{(x, x) : x \in X\} \subseteq \rho(f)$.

**Example 18.** The invariant version of PointNet is able to learn functions of the form $\sum_i f(x_i)$ for $f \in \mathcal{C}(\mathbb{R}^p, \mathbb{R})$. We can apply Corollary 17 to this setting. Consider the case where $X$ is a compact subset of $\mathbb{R}^p$ and $\mathcal{F} = \{x \mapsto g(\sum_{i=1}^n f(x_i)), f \in \mathcal{C}(X, \mathbb{R}^h), g \in \mathcal{C}(\mathbb{R}^h, \mathbb{R})\}$. $\mathcal{F}$ is a subalgebra of $\mathcal{C}(X, \mathbb{R})$ (indeed of $\mathcal{C}_I(X, \mathbb{R})$) which contains the constant function $\mathbf{1}$. Then, it easy to see that $\rho(\mathcal{F}) = \{(x, \sigma \star x), \sigma \in \mathcal{S}_n\}$, where $\sigma \star x$ is defined by $(\sigma \star x)_{\sigma(i)} = x_i$ for all $i$ (see Lem. 31 for a formal statement).

Now note that for a function $f \in \mathcal{C}(X, \mathbb{R})$, the condition $\{(x, \sigma \star x), \sigma \in \mathcal{S}_n\} \subseteq \rho(f)$ is equivalent to $f \in \mathcal{C}_I(X, \mathbb{R})$. So that Corollary 17 implies that $\overline{\mathcal{F}} = \mathcal{C}_I(X, \mathbb{R})$ which means that PointNet is universal for approximating invariant functions. This was already proved in Qi et al. (2017) .

We now provide a proof of Corollary 17 for completeness.

*Proof.* The first inclusion $\overline{\mathcal{F}} \subseteq \{f \in \mathcal{C}(X, \mathbb{R}) : \rho(\mathcal{F}) \subseteq \rho(f)\}$ follows from the same argument as the one given below Theorem 4 so we focus on the other one.

For every $x \in X$, let $x_{\mathcal{F}}$ denote its $\rho(\mathcal{F})$-class. The quotient set and the canonical surjection are: $X_{\mathcal{F}} = X/\rho(\mathcal{F}) = \{x_{\mathcal{F}}, x \in X\}$ and $\pi_{\mathcal{F}} : X \to X_{\mathcal{F}}, \pi_{\mathcal{F}}(x) = x_{\mathcal{F}}$.

A function $g : X \to \mathbb{R}$ factorizes as $g = \hat{g} \circ \pi_{\mathcal{F}}$ for some $\hat{g} : X_{\mathcal{F}} \to \mathbb{R}$ if and only if $\rho(\mathcal{F}) \subseteq \rho(g)$. In this case $\hat{g}$ is unique, since $\pi_{\mathcal{F}}$ is a surjection. In particular. every $f \in \mathcal{F}$ factorizes uniquely as $f = \hat{f} \circ \pi_{\mathcal{F}}, \hat{f} : X_{\mathcal{F}} \to \mathbb{R}$, and $\hat{\mathcal{F}} = \{\hat{f}, f \in \mathcal{F}\}$ clearly separates points on $X_{\mathcal{F}}$. We refer to Munkres (2000, §22) for the properties of the quotient topology. In particular, by the properties of the quotient topology on $X_{\mathcal{F}}$, $\hat{\mathcal{F}}$ is a subalgebra of $\mathcal{C}(X_{\mathcal{F}}, \mathbb{R})$ and $X_{\mathcal{F}}$ is compact. Hence, we can apply Theorem 16 to $\hat{\mathcal{F}}$ and $\hat{\mathcal{F}}$ is dense in $\mathcal{C}(X_{\mathcal{F}}, \mathbb{R})$.

Now take $f \in \mathcal{C}(X, \mathbb{R})$ with $\rho(\mathcal{F}) \subseteq \rho(f)$ and we show that $f \in \overline{\mathcal{F}}$. Again, $\rho(\mathcal{F}) \subseteq \rho(f)$ implies that $f = \hat{f} \circ \pi_{\mathcal{F}}$. Let $\epsilon > 0$. By density of $\hat{\mathcal{F}}$, there is some $\hat{h} \in \hat{\mathcal{F}}$ such that $\sup_{x_{\mathcal{F}}} |\hat{h}(x_{\mathcal{F}})) - \hat{f}(x_{\mathcal{F}})| \leq \epsilon$. But, by construction of $\hat{F}$, there exits $h \in \mathcal{F}$ such that $\hat{h} \circ \pi_{\mathcal{F}} = h$. Thus,

$$\sup_{x \in X} |h(x) - f(x)| = \sup_{x \in X} |\hat{h}(\pi_{\mathcal{F}}(x)) - \hat{f}(\pi_{\mathcal{F}}(x))| = \sup_{x_{\mathcal{F}} \in X_{\mathcal{F}}} |\hat{h}(x_{\mathcal{F}})) - \hat{f}(x_{\mathcal{F}})| \leq \epsilon.$$

As this holds for any $\epsilon > 0$, we have proven that $f \in \overline{\mathcal{F}}$. □

### D.4   THE EQUIVARIANT APPROXIMATION THEOREM

We first need to extend Corollary 17 to vector-valued functions. For this, we need to have a vector-valued version of the Stone-Weierstrass theorem and as shown by the example below additional assumptions have to be made.

**Example 19.** We consider now the equivariant version of PointNet corresponding to the particular case where $X$ is a compact subset of $(\mathbb{R}^p)^n$, $Y = \mathbb{R}^n$ and $\mathcal{F} = \{x \mapsto (f(x_1), \ldots, f(x_n)),\ f \in \mathcal{C}(\mathbb{R}, \mathbb{R})\}$. Then clearly $\mathcal{F}$ is a subalgebra of $\mathcal{C}_E(X, Y)$ containing the constant function $\mathbf{1}$ and $\rho(\mathcal{F}) = \{(x, x),\ x \in X\}$. Hence if Corollary 17 would be true with vector-valued functions instead of real-valued functions, we would have that $\mathcal{F}$ is dense in $\mathcal{C}(X, Y)$. But this can clearly not be true as $\mathcal{F} \subseteq \mathcal{C}_E(X, Y)$ which is clearly not dense in $\mathcal{C}(X, Y)$.

We now present an extension of Corollary 17 also due to Timofte (2005):

**Proposition 20.** *Let $X$ be a compact space, $Y = \mathbb{R}^p$ for some $p \geq 1$. Let $\mathcal{F} \subseteq \mathcal{C}(X, Y)$. If there exists a nonempty subset $S \subseteq \mathcal{C}(X, \mathbb{R})$ such that:*

$$S \cdot \mathcal{F} \subseteq \mathcal{F} \text{ and, } \rho(S) \subseteq \rho(\mathcal{F}). \tag{17}$$

*Then we have*

$$\overline{\mathcal{F}} = \left\{ f \in \mathcal{C}(X, Y),\ \rho(\mathcal{F}) \subseteq \rho(f),\ f(x) \in \overline{\mathcal{F}(x)} \right\}, \tag{18}$$

*where $\mathcal{F}(x) = \{f(x),\ f \in \mathcal{F}\}$. Moreover in (18), we can replace $\rho(\mathcal{F})$ by $\rho(S)$.*

Note that in the particular case $Y = \mathbb{R}$ and if $\mathcal{F}$ is a subalgebra of $\mathcal{C}(X, \mathbb{R})$, then we can take $S = \mathcal{F}$ in (17) and if the constant function $1$ is in $\mathcal{F}$, then $\mathcal{F}(x) = \mathbb{R}$, so that we recover Corollary 17.

Now consider the case where $\mathcal{F}$ is a subalgebra of $\mathcal{C}(X, \mathbb{R}^p)$. We need to find a set $S \in \mathcal{C}(X, \mathbb{R})$ satisfying (17) i.e. with a better separating power than $\mathcal{F}$ but containing real-valued functions such that $sf \in \mathcal{F}$ for all $s \in S$ and $f \in \mathcal{F}$. In the sequel, we will consider the set $\mathcal{F}_{scal} = \{f \in \mathcal{C}(X, \mathbb{R}) : f\mathbf{1} \in \mathcal{F}\}$. We clearly have $\mathcal{F}_{scal} \cdot \mathcal{F} \subseteq \mathcal{F}$ since $\mathcal{F}$ is a subalgebra. Hence, in this setting, Prop. 20 can be rewritten as follows:

**Corollary 21.** *Let $X$ be a compact space, $Y = \mathbb{R}^p$ for some $p$, $G$ be a finite group acting (continuously) on $X$ and $\mathcal{F} \subseteq \mathcal{C}_I(X, Y)$ a (non-empty) set of invariant functions associated to $G$.*

*Consider the following assumptions,*

1. *$\mathcal{F}$ is a sub-algebra of $\mathcal{C}(X, Y)$ and the constant function $\mathbf{1}$ is in $\mathcal{F}$.*

2. *The set of functions $\mathcal{F}_{scal} \subseteq \mathcal{C}(X, \mathbb{R})$ defined by,*

$$\mathcal{F}_{scal} = \{f \in \mathcal{C}(X, \mathbb{R}) : f\mathbf{1} \in \mathcal{F}\}$$

   *satisfy,*

$$\rho(\mathcal{F}_{scal}) \subseteq \rho(\mathcal{F}).$$

3. *For any $x \in X$, there exists $f \in \mathcal{F}$ such that $f(x)$ has pairwise distinct coordinates, i.e., for any indices $i, j \in \{1, \ldots, p\}$ with $i \neq j$, $f(x)_i \neq f(x)_j$.*

*Then the closure of $\mathcal{F}$ (for the topology of uniform convergence) is,*

$$\overline{\mathcal{F}} = \{f \in \mathcal{C}_I(X, Y) : \rho(\mathcal{F}) \subseteq \rho(f)\}.$$

Note that Assumptions 1 and 2 ensures that (17) is valid, while Assumption 3 ensures that $\mathcal{F}(x) = \mathbb{R}^p$. Unfortunately, in the equivariant case, the condition $\rho(\mathcal{F}_{scal}) \subseteq \rho(\mathcal{F})$ is too strong and we now explain how we will relax it.

For the sake of simplicity, we consider here the particular setting adapted to graphs: let $n \geq 1$ be a fixed number (corresponding to the number of nodes), $X$ be a compact set of graphs in $\mathbb{R}^{n^2}$ and $Y = \mathbb{F}^n$ with $\mathbb{F} = \mathbb{R}^p$ for some $p \geq 1$. We define the action of the symmetric group $\mathcal{S}_n$ on $X$ by $(\sigma \star x)_{\sigma(i), \sigma(j)} = x_{i.j}$ and on $Y$ by $(\sigma \star y)_{\sigma(i)} = y_i \in \mathbb{R}^p$. Hence the set of continuous equivariant functions $\mathcal{C}_E(X, Y)$ agrees with Definition 1.

Now consider the case where $\mathcal{F} \subseteq \mathcal{C}_E(X, Y)$ is a subalgebra of equivariant functions. Then, $f \in \mathcal{F}_{scal}$ needs to be invariant in order for $f\mathbf{1}$ to be equivariant and hence in $\mathcal{F}$. As a result, we see that $\mathcal{F}_{scal}$ will not separate points of $X$ in the same orbit, i.e. $x$ and $\sigma \star x$. But these points will typically be separated by $\mathcal{F}$, since for any $f \in \mathcal{F}$, we have $f(\sigma \star x) = \sigma \star f(x)$ which is not equal to $f(x)$ unless $f$ is invariant.

We see that we need somehow to require a weaker separating power for $\mathcal{F}$. More formally, two isomorphic graphs will have permuted outputs through an equivariant function, but should not be considered as separated. Let $Orb(x) = \{\sigma \star x, \ \sigma \in \mathcal{S}_n\}$ and $Orb(y) = \{\sigma \star y, \ \sigma \in \mathcal{S}_n\}$. For any equivariant function $f \in \mathcal{C}_E(X, Y)$, for any $z \in Orb(x)$, we have $f(z) \in Orb(f(x))$. Then let $\pi : Y \to Y/\mathcal{S}_n$ be the canonical projection $\pi(y) = Orb(y)$. We define

$$
\begin{aligned}
(x, x') \in \rho\left(\pi \circ \mathcal{F}\right) \quad &\Leftrightarrow \quad \forall f \in \mathcal{F}, \ Orb(f(x)) = Orb(f(x')) \\
&\Leftrightarrow \quad \forall f \in \mathcal{F}, \exists \sigma \in \mathcal{S}_n, \ f(\sigma \star x) = f(x').
\end{aligned}
$$

In particular, we see that if $x' \in Orb(x)$ then $(x, x') \in \rho\left(\pi \circ \mathcal{F}\right)$ for any $\mathcal{F} \in \mathcal{C}_E(X, Y)$. Moreover, two graphs $x$ and $x'$ are $\rho\left(\pi \circ \mathcal{F}\right)$-distinct if there exists a function $f \in \mathcal{F}$ such that $\forall \sigma, \ f(\sigma \star x) \neq f(x')$, i.e. the function $f$ discriminates $Orb(x)$ from $Orb(x')$ in the sense that for any $z \in Orb(x)$ and $z' \in Orb(x')$, we have $f(z) \neq f(z')$.

To obtain an equivalent of Proposition 20 with $\mathcal{C}_E(X, Y)$ replacing $\mathcal{C}(X, Y)$, we are able to relax assumption (17) to $\rho\left(\mathcal{F}_{scal}\right) \subseteq \rho\left(\pi \circ \mathcal{F}\right)$. Our main general result in this direction is the following theorem (proved in Section D.7) which might be of independent interest:

**Theorem 22.** *Let $X$ be a compact space, $Y = \mathbb{R}^p$ for some p, G be a finite group acting (continuously) on $X$ and $Y$ and $\mathcal{F} \subseteq \mathcal{C}_E(X, Y)$ a (non-empty) set of equivariant functions.*

*Denote by $\pi : Y \longrightarrow Y/G$ the canonical projection on the quotient space $Y/G$. Consider the following assumptions,*

1. *$\mathcal{F}$ is a sub-algebra of $\mathcal{C}(X, Y)$ and the constant function $\mathbf{1}$ is in $\mathcal{F}$.*

2. *The set of functions $\mathcal{F}_{scal} \subseteq \mathcal{C}(X, \mathbb{R})$ defined by,*

$$
\mathcal{F}_{scal} = \{f \in \mathcal{C}(X, \mathbb{R}) : f\mathbf{1} \in \mathcal{F}\}
$$

*satisfy,*

$$
\rho\left(\mathcal{F}_{scal}\right) \subseteq \rho\left(\pi \circ \mathcal{F}\right) .
$$

*Then the closure of $\mathcal{F}$ (for the topology of uniform convergence) is,*

$$
\overline{\mathcal{F}} = \{f \in \mathcal{C}_E(X, Y) : \rho\left(\mathcal{F}\right) \subseteq \rho\left(f\right), \ \forall x \in X, \ f(x) \in \mathcal{F}(x)\} ,
$$

*where $\mathcal{F}(x) = \{f(x), \ f \in \mathcal{F}\}$. Moreover, if $I(x) = \{(i, j) \in [p]^2 : \forall y \in \mathcal{F}(x), \ y_i = y_j\}$, then we have:*

$$
\mathcal{F}(x) = \{y \in \mathbb{R}^p : \forall(i, j) \in I(x), \ y_i = y_j\} .
$$

**Example 23.** We now demonstrate how Theorem 22 can be used to recover the universality results in Segol & Lipman (2020). In this paper, the authors study equivariant neural network architectures working with unordered sets, corresponding in our case to $X = Y = \mathbb{R}^n$ and the group being the symmetric group $\mathcal{S}_n$. They show that the PointNet architecture cannot approximate any (continuous) equivariant function and that adding a single so-called transmission layer is enough to make this architecture universal.

Indeed, PointNet can only learn maps of the form $x \in \mathbb{R}^n \mapsto (f(x_1) \dots f(x_n))$, which are not universal in the class of equivariant functions, as shown by Segol & Lipman (2020, Lem. 3). Now, their transmission layer is a map of the form $x \in \mathbb{R}^n \mapsto (\mathbf{1}^T x)\mathbf{1}$. Therefore, in PointNetST, adding such a layer precisely adds a large class of functions to $\mathcal{F} = \{(f(x_1, \sum_i g(x_i)), \dots, f(x_n, \sum_i g(x_i))), f \in \mathcal{C}(\mathbb{R} \times \mathbb{R}^h, \mathbb{R}), \ g \in \mathcal{C}(\mathbb{R}, \mathbb{R}^h), h \geq 1\}$. $\mathcal{F}$ is still an algebra and as shown in Example 19, we have $\rho\left(\mathcal{F}\right) = \{(x, x), x \in X\}$. Moreover, we have $\mathcal{F}_{scal} = \mathcal{C}_I(X, \mathbb{R})$ by Lem. 33 in particular, we get $\rho\left(\mathcal{F}_{scal}\right) = \{(x, \sigma \star x), \ x \in X\}$, so that we obviously have $\rho\left(\mathcal{F}_{scal}\right) \subseteq \rho\left(\pi \circ \mathcal{F}\right)$. In summary, Theorem 22 implies the universality of PointNetST in $\mathcal{C}_E(X, Y)$.

### D.5 A PRELIMINARY VERSION OF THE EQUIVARIANT APPROXIMATION THEOREM

We start by proving a version of Theorem 22 with a slightly weaker condition:

**Proposition 24.** *Let $X$ be a compact space, $Y = \mathbb{R}^p$ for some $p$, $G$ be a finite group acting (continuously) on $X$ and $Y$ and $\mathcal{F} \subseteq \mathcal{C}_E(X, Y)$ a (non-empty) set of equivariant functions.*

*Consider the following assumptions,*

1. *$\mathcal{F}$ is a subalgebra $\mathcal{C}_E(X, Y)$.*

2. *The set of real-valued functions $\mathcal{F}_{scal} \subseteq \mathcal{C}(X, \mathbb{R})$ defined by,*

$$\mathcal{F}_{scal} = \{f \in \mathcal{C}(X, \mathbb{R}) : f\mathbf{1} \in \mathcal{F}\}$$

   *satisfies,*

$$\rho\left(\mathcal{F}_{scal}\right) \subseteq \left\{(x, x') \in X \times X : \exists g \in G, \ (g \cdot x, x') \in \rho\left(\mathcal{F}\right)\right\}.$$

*Then the closure of $\mathcal{F}$ (for the topology of uniform convergence) is,*

$$\overline{\mathcal{F}} = \left\{f \in \mathcal{C}_E(X, Y) : \rho\left(\mathcal{F}\right) \subseteq \rho\left(f\right), \ \forall x \in X, \ f(x) \in \overline{\mathcal{F}(x)}\right\}, \tag{19}$$

*where $\mathcal{F}(x) = \{f(x), \ f \in \mathcal{F}\}$.*

The proof of this theorem relies on two main ingredients. First, following the elegant idea of Maehara & Hoang (2019), we augment the input space to transform the vector-valued equivariant functions into scalar maps. Second, we apply the fine-grained approximation result Cor. 17.

*Proof.* As uniform convergence implies point-wise convergence, the first inclusion is immediate,

$$\overline{\mathcal{F}} \subseteq \left\{f \in \mathcal{C}_E(X, Y) : \rho\left(\mathcal{F}\right) \subseteq \rho\left(f\right), \ \forall x \in X, \ f(x) \in \overline{\mathcal{F}(x)}\right\}.$$

The rest of the proof is devoted to the other direction.

For convenience, denote by $\Phi$ the family of linear forms associated to the canonical basis of $\mathbb{R}^p$, i.e.,

$$\Phi = \{y \mapsto y_i : 1 \leq i \leq p\} \subseteq \mathcal{C}(Y, \mathbb{R}).$$

Define our augmented input space as $\tilde{X} = X \times \Phi$. As $\Phi$ is finite and $X$ is compact, $\tilde{X}$ is still a compact space. We now transform $\mathcal{F}$, a class of equivariant functions from $X$ to $Y$, into $\tilde{\mathcal{F}}$ a class of maps from $\tilde{X}$ to $\mathbb{R}$. Define

$$\tilde{\mathcal{F}} = \{(x, \varphi) \mapsto \varphi(f(x)) : f \in \mathcal{F}\}.$$

We check that $\tilde{\mathcal{F}}$ is indeed a subset of $\mathcal{C}(\tilde{X}, \mathbb{R})$. Indeed, as $\Phi$ is finite, it is equipped with the discrete topology. Hence, each singleton $\{\varphi\}$ for $\varphi \in \Phi$ is open in $\Phi$ and it suffices to check the continuity in the first variable with $\varphi$ fixed. But, if $f \in \mathcal{F}$, $x \mapsto \varphi(f(x))$ is continuous as a composition of continuous maps.

We can now apply Cor. 17 to $\tilde{\mathcal{F}} \subseteq \mathcal{C}(\tilde{X}, \mathbb{R})$. Therefore, the closure of $\tilde{F}$ in $\mathcal{C}(\tilde{X}, \mathbb{R})$ is,

$$\overline{\tilde{\mathcal{F}}} = \left\{v \in \mathcal{C}(\tilde{X}, \mathbb{R}) : \rho\left(\tilde{\mathcal{F}}\right) \subseteq \rho\left(v\right), \ \forall(x, \varphi) \in \tilde{X}, \ v(x, \varphi) \in \overline{\tilde{\mathcal{F}}(x, \varphi)}\right\}.$$

We now show the equality of (19). Take $h$ in the right-hand side of (19), i.e. $h \in \mathcal{C}_E(X, Y)$ such that $\rho\left(\mathcal{F}\right) \subseteq \rho\left(h\right)$ and $h(x) \in \overline{\mathcal{F}(x)}$ for all $x \in X$. We show that $\tilde{h}$, defined by $\tilde{h} : (x, \varphi) \mapsto \varphi(h(x))$, belongs to $\overline{\tilde{F}}$ using the result above.

- As $h$ is continuous, by the same argument as above, $(x, \varphi) \mapsto \varphi(h(x))$ is continuous on $\tilde{X}$.

- We check that $\rho\left(\tilde{\mathcal{F}}\right) \subseteq \rho\left(\tilde{h}\right)$.

  Take $(x, \varphi), (y, \psi) \in \tilde{X}$ such that, for all $f \in \mathcal{F}$,

  $$\varphi(f(x)) = \psi(f(y)), \tag{20}$$

  and we aim at showing that $\varphi(h(x)) = \psi(h(y))$.

  To gain more information from (20), we apply it to functions of the form $f\mathbf{1}$ with $f \in \mathcal{F}_{scal}$. By definition of $\Phi$, this translates to $f(x) = f(y)$, for any $f \in \mathcal{F}_{scal}$. Therefore $(x, y) \in \rho\left(\mathcal{F}_{scal}\right)$ and so there exists $g \in G$ such that $(g \cdot x, y) \in \rho\left(\mathcal{F}\right)$. Plugging this into (20) and using the equivariance of $f$,

  $$\forall f \in \mathcal{F}, \ \varphi(f(x)) = \psi(g \cdot f(x)).$$

  As $G$ acts continuously on $Y$, both $\varphi$ and $z \mapsto \psi(g \cdot z)$ are continuous and, as a consequence of the equality above, coincide on $\overline{\mathcal{F}(x)}$. But we assumed that $h(x) \in \overline{\mathcal{F}(x)}$ and therefore the equality also holds for $h$, i.e. $\varphi(h(x)) = \psi(g \cdot h(x))$.

  Finally, recalls that, by assumption, $\rho\left(\mathcal{F}\right) \subseteq \rho\left(h\right)$. Therefore $(g \cdot x, y) \in \rho\left(\mathcal{F}\right)$ implies that $h(g \cdot x) = h(y)$ and, combined with the result above, $\varphi(h(x)) = \psi(h(y))$.

- We verify that, for $x \in X$, $\varphi \in \Phi$, $\varphi(h(x))$ belongs to $\overline{\tilde{\mathcal{F}}(x)}$. Indeed, recall that $h(x) \in \overline{\mathcal{F}(x)}$. Therefore, as $\varphi$ is continuous, $\varphi(h(x))$ is in $\overline{\varphi(\mathcal{F}(x))}$ which is included in $\overline{\tilde{\mathcal{F}}(x)}$.

This shows that $\tilde{h} : (x, \varphi) \mapsto \varphi(h(x))$ is in $\overline{\overline{\tilde{\mathcal{F}}}}$. Consequently, for any $\epsilon > 0$, there exists $f \in \mathcal{F}$ such that,

$$\forall x \in X, \ \forall \varphi \in \Phi, \ |\varphi(h(x)) - \varphi(f(x))| \leq \epsilon.$$

If $Y = \mathbb{R}^p$ is endowed with the infinity norms on coordinates, by definition of $\Phi$, this means that,

$$\forall x \in X, \ \|h(x) - f(x)\| \leq \epsilon.$$

$\square$

*Remark* 25. In the particular case of $G \subseteq \mathcal{S}_n$ being a group of permutations acting on $\mathbb{R}^p$ by, for $g \in G, x \in \mathbb{R}^p$,

$$\forall i \in \{1, \ldots, p\}, \ (g \cdot x)_i = x_{g^{-1}(i)},$$

the functions of $\tilde{\mathcal{F}}$ are indeed invariant, as shown by Maehara & Hoang (2019). For this, a left action on $\Phi$ is defined by, for $g \in G, \varphi \in \Phi$,

$$\forall x \in \mathbb{R}^p, \ (g \cdot \varphi)(x) = \varphi(g^{-1} \cdot x).$$

In other words, the action of $g$ on the linear form associated to the $i^{th}$ coordinate yields the linear form associated to the $g(i)^{th}$ coordinate. One can now check that the functions $\tilde{\mathcal{F}}$ are invariant.

## D.6 CHARACTERIZING THE SUBALGEBRAS OF $\mathbb{R}^p$

Before moving to our general result, we need to study the structure of the subalgebras of $\mathbb{R}^p$. For this, we will use the following simple lemma.

In the following lemma, $\mathbb{R}[X_1, \ldots, X_p]$ denotes the set of multivariate polynomials with $p$ indeterminates (and real coefficients).

**Lemma 26.** *Let $C \subseteq \mathbb{R}^p$ be a finite subset of $\mathbb{R}^p$. There exists $P \in \mathbb{R}[X_1, \ldots, X_p]$ such that $P_{|C}$, the restriction of $P$ to $C$, is an injective map.*

*Proof.* Let $x^1, \ldots, x^m \in \mathbb{R}^p$ be distinct vectors such that $\{x^1, \ldots, x^m\} = C$. Similarly to Lagrange polynomials, define,

$$P(X_1, \ldots, X_p) = \sum_{i=1}^{m} i \prod_{j \neq i} \frac{\sum_{l=1}^{p}(X_l - x_l^j)^2}{\|x^i - x^j\|_2^2}, \tag{21}$$

which is a well-defined multivariate polynomial. Note that, seeing $X = (X_1, \ldots, X_p)$ as a vector in $\mathbb{R}^p$, it can also be written as,

$$P(X_1, \ldots, X_p) = \sum_{i=1}^{m} i \prod_{j \neq i} \frac{\|X - x^j\|_2^2}{\|x^i - x^j\|_2^2}. \tag{22}$$

By construction, $P(x^i) = i$ and therefore $P$ is an injective map on $C$ $\qquad\square$

**Lemma 27.** *For a subalgebra $\mathcal{A}$ of $\mathbb{R}^p$, we define:*

$$J = \{j \in \{1, \ldots, p\} : \forall x \in \mathcal{A}, \ x_j = 0\}$$
$$I = \{(i, j) \in (J^C)^2 : \forall x \in \mathcal{A}, \ x_i = x_j\}.$$

*Then, we have :*

$$\mathcal{A} = \{x \in \mathbb{R}^p : \forall (i, j) \in I, \ x_i = x_j, \ \forall j \in J, \ x_j = 0\}.$$

*Proof.* Before proving the general case, we focus on the situation where $\mathcal{A}$ will turn out to be the whole $\mathbb{R}^p$.

Assume that the two following conditions holds,

$$\forall i \neq j, \ \exists x \in \mathcal{A}, \ x_i \neq x_j \tag{23}$$
$$\forall i, \ \exists x \in \mathcal{A}, \ x_i \neq 0 \tag{24}$$

Our goal is to show that, under these additional assumptions, $\mathcal{A} = \mathbb{R}^p$. We divide the proof in three parts, first we show that $\mathbf{1} \in \mathcal{A}$ using (24), giving us that $\mathcal{A}$ is closed under polynomials, then that there is $x \in \mathcal{A}$ with pairwise distinct coordinates thanks to (23) and finally that this implies that $\mathcal{A}$ is the whole space.

Note that if $p = 1$, (24) and the linear space property of $\mathcal{A}$ immediately give the result.

- Here we prove that (24) implies that $\mathbf{1} \in \mathcal{A}$.

  - First, we construct by induction $x \in \mathcal{A}$ such that $x_i \neq 0$ for any index $i$. More precisely, our induction hypothesis at step $j \in \{1, \ldots, p\}$ is,

    $$\exists x \in \mathcal{A}, \ \forall 1 \leq i \leq j, \ x_i \neq 0. \tag{25}$$

    By (24), this holds for $j = 1$.
    Now, assume that it holds at $j - 1$ for some $p \geq j \geq 2$ and take $x \in \mathcal{A}$ such that $x_i \neq 0$ for any $1 \leq i \leq j - 1$ and $y \in \mathcal{A}$ such that $j_j \neq 0$ by (24).
    By definition of $x$, the set,

    $$\{\lambda \in \mathbb{R} : \exists 1 \leq i \leq j - 1, \ \lambda x_i + y_i = 0\},$$

    is finite. Thus, there exists $\lambda \in \mathbb{R}$ such that $\lambda x_i + y_i \neq 0$ for any $1 \leq i \leq j - 1$ and for $i = j$ too as $x_j = 0$ and $y_j \neq 0$. As $\mathcal{A}$ is a subalgebra, $\lambda x + y \in \mathcal{A}$ and this concludes the induction step.
  - Let $x \in \mathcal{A}$ be the vector constructed, i.e. such that $x_i \neq 0$ for every index $i$. We prove that $\mathbf{1} \in \mathcal{A}$ by constructing $\mathbf{1}$ from $x$.
    Indeed, using Lagrange interpolation, take $P \in \mathbb{R}[X]$ such that $P(x_i) = \frac{1}{x_i}$ for every $i$. (Note that this noes not matter if some $x_i$ are equal, as the $\frac{1}{x_i}$ would also be the same.)
    Finally, as $\mathcal{A}$ is a subalgebra, $xP(x)$, which is to be understood coordinate-wise, is in $\mathcal{A}$ and so is $\mathbf{1} = xP(x)$.

- We show that the previous point and (23) imply that there exists a vector in $\mathcal{A}$ with pairwise distinct coordinates, i.e. that there exists $x \in \mathcal{A}$ such that, for any $i \neq j$, $x_i \neq x_j$. Using (23), for any $i < j$, there exists $x^{ij} \in \mathcal{A}$ such that $x_i^{ij} \neq x_j^{ij}$. We wish to combine the family $(x^{ij})_{i<j}$ into a single vector.

  For this we use Lem. 26. Seeing each collection $(x_k^{ij})_{i<j}$ as vector of $\mathbb{R}^{p(p-1)/2}$, we define $C = \{(x_k^{ij})_{i<j} : 1 \leq k \leq p\}$, which is a finite subset (of cardinal $p$) of $\mathbb{R}^{p(p-1)/2}$. By Lem. 26, there exists $P \in \mathbb{R}[X_1, \ldots, X_{p(p-1)/2}]$ such that $P$ is an injective map on $C$.

As $\mathcal{A}$ is a subalgebra and $\mathbf{1} \in \mathcal{A}$, the vector,

$$P\left((x^{ij})_{i<j}\right) = \begin{pmatrix} P\left((x_1^{ij})_{i<j}\right) \\ \vdots \\ P\left((x_p^{ij})_{i<j}\right) \end{pmatrix},$$

is in $\mathcal{A}$ too.

We now check that this vector has pairwise distinct coordinates. Let $l < k$, then $x_l^{lk} \neq x_k^{lk}$ and therefore $(x_l^{ij})_{i<j} \neq (x_k^{ij})_{i<j}$. By construction of $P$, $P\left((x_l^{ij})_{i<j}\right) \neq P\left((x_k^{ij})_{i<j}\right)$, i.e. $P\left((x^{ij})_{i<j}\right)_l \neq P\left((x^{ij})_{i<j}\right)_k$. Thus, $P\left((x^{ij})_{i<j}\right) \in \mathcal{A}$ has pairwise distinct coordinates as required.

- Finally, we show that $\mathcal{A} = \mathbb{R}^p$. This is a direct consequence of the point above and of Lagrange interpolation. Indeed, take any $y \in \mathbb{R}^p$ and denote by $x \in \mathcal{A}$ the vector that we just constructed with pairwise distinct coordinates. Therefore, by Lagrange interpolation, there exists $P \in \mathbb{R}[X]$ such that $P(x_i) = y_i$ for every $i \in \{1, \ldots, p\}$. As $\mathcal{A}$ is a sublagebra and $\mathbf{1} \in \mathcal{A}$, $P(x) \in \mathcal{A}$ and hence $y \in \mathcal{A}$.

Finally, we return to the general case. We introduce the set of indexes of $I$ and $J$ which appear in the result and use them to reduce the situation to the previous case. Define,

$$J = \{j \in \{1, \ldots, p\} : \forall x \in \mathcal{A}, \ x_j = 0\}$$
$$I = \{(i, j) \in (J^C)^2 : \forall x \in \mathcal{A}, \ x_i = x_j\}.$$

and denote by $\mathcal{A}' = \{x \in \mathbb{R}^p : \forall (i, j) \in I, \ x_i = x_j, \ \forall j \in J, \ x_j = 0\}$, which is also a subalgebra. By definition, it holds that $\mathcal{A} \subseteq \mathcal{A}'$.

By construction, $I$ is an equivalence relation on $J^C$ and denote by $J^C/I$ its equivalence classes. Let $p' = |J^C/I|$ and choose $i_1, \ldots, i_{p'}$ representatives of the equivalence classes. Consider the map,

$$\varphi : \begin{cases} \mathbb{R}^p \longrightarrow \mathbb{R}^{p'} \\ x \longmapsto (x_{i_1}, \ldots, x_{i_{p'}}). \end{cases}$$

$\varphi$ is an algebra homomoprhism so that $\varphi(\mathcal{A})$ is a subalgebra of $\mathbb{R}^{p'}$. But, by construction of $I$ and $J$, $\varphi(\mathcal{A})$ satisfies (23) and (24). Whence, by our result in this particular case, $\varphi(\mathcal{A}) = \mathbb{R}^{p'}$.

However, $\mathcal{A} \subseteq \mathcal{A}'$ implies that $\varphi(\mathcal{A}) \subseteq \varphi(\mathcal{A}') \subseteq \mathbb{R}^{p'}$. Therefore, $\mathbb{R}^{p'} = \varphi(\mathcal{A}) = \varphi(\mathcal{A}')$. But, by construction of $\varphi$, $\varphi$ is actually an injective map on $\mathcal{A}'$. Therefore, we deduce from $\varphi(\mathcal{A}) = \varphi(\mathcal{A}')$ that $\mathcal{A} = \mathcal{A}'$, concluding the proof. $\qquad\square$

## D.7 PROOF OF THE MAIN EQUIVARIANT APPROXIMATION THEOREM

We can now fully exploit the structure of subalgebra of $\mathcal{F}$ thanks to the results above, and in particular relax the second assumption of Prop. 24 to give our main theorem 22. We first prove the following lemma.

**Lemma 28.** *Under the assumptions of Thm. 22, for any $H \subseteq G$ subgroup, for any $x, y \in X$,*

$$\forall f \in \mathcal{F}, \ \exists g \in H, \ f(g \cdot x) = f(y) \iff \exists g \in H, \ \forall f \in \mathcal{F}, \ f(g \cdot x) = f(y).$$

*In particular,*
$$(x, y) \in \rho(\pi \circ \mathcal{F}) \iff \exists g \in G, \ (g \cdot x, y) \in \rho(\mathcal{F}).$$

*Proof.* The reverse implication is immediate so we focus on the direct one and prove its contraposition, i.e.,
$$\forall g \in H, \ \exists f \in \mathcal{F}, \ f(g \cdot x) \neq f(y) \implies \exists f \in \mathcal{F}, \ \forall g \in H, \ f(g \cdot x) \neq f(y).$$
To prove this, we take advantage of $\mathcal{F}$ being a subalgebra and $H$ being finite. Let $H = \{g_1, \ldots, g_h\}$ and define,
$$\mathcal{A} = \{(f(g_1 \cdot x), \ldots, f(g_h \cdot x), f(y)) : f \in \mathcal{F}\}.$$

As $\mathcal{F}$ is a subalgebra of $\mathcal{C}(X, \mathbb{R}^p)$, $\mathcal{A}$ is a subalgebra of $\mathbb{R}^{p'}$ with $p' = p(h+1)$. By Lem. 27,

$$\mathcal{A} = \{z \in \mathbb{R}^{p'} : \forall (i,j) \in I, \; z_i = z_j, \; \forall j \in J, \; z_j = 0\}.$$

where $I$ and $J$ can be chosen to be,[1]

$$J = \{j \in \{1, \ldots, p\} : \forall z \in \mathcal{A}, \; z_j = 0\}$$
$$I = \{(i,j) \in \{1, \ldots, p\}^2 : \forall z \in \mathcal{A}, \; z_i = z_j\}.$$

As $I$ is an equivalence relation, an element of $\mathcal{A}$ is uniquely defined by its coordinates on equivalence classes of $I$. Therefore, one can choose $z \in \mathcal{A}$ such that $z_i = z_j \iff (i,j) \in I$. By definition of $\mathcal{A}$, there exists $f^* \in \mathcal{F}$ such that $(f^*(g_1 \cdot x), \ldots, f^*(g_h \cdot x), f^*(y)) = z$. We now check that $f^*$ is indeed appropriate. Take $l \in \{1, \ldots, h\}$, we want to show that $f^*(g_l \cdot x) \neq f^*(y)$. By assumption, there exists $f \in \mathcal{F}$ such that $f(g_l \cdot x) \neq f(y)$, i.e. there exists $i \in \{1, \ldots, p\}$ such that $f(g_l \cdot x)_i \neq f(y)_i$. Therefore, $((l-1)p + i, hp + i)$ cannot be in $I$ so that $z_{(l-1)p+i} \neq z_{hp+1}$, i.e. $f^*(g_l \cdot x)_i \neq f^*(y)_i$. $\qquad \square$

We now prove our main abstract theorem.

**Theorem 22.** *Let $X$ be a compact space, $Y = \mathbb{R}^p$ for some $p$, $G$ be a finite group acting (continuously) on $X$ and $Y$ and $\mathcal{F} \subseteq \mathcal{C}_E(X, Y)$ a (non-empty) set of equivariant functions.*

*Denote by $\pi : Y \longrightarrow Y/G$ the canonical projection on the quotient space $Y/G$. Consider the following assumptions,*

1. *$\mathcal{F}$ is a sub-algebra of $\mathcal{C}(X, Y)$ and the constant function $\mathbf{1}$ is in $\mathcal{F}$.*

2. *The set of functions $\mathcal{F}_{scal} \subseteq \mathcal{C}(X, \mathbb{R})$ defined by,*
$$\mathcal{F}_{scal} = \{f \in \mathcal{C}(X, \mathbb{R}) : f\mathbf{1} \in \mathcal{F}\}$$
   *satisfy,*
$$\rho\left(\mathcal{F}_{scal}\right) \subseteq \rho\left(\pi \circ \mathcal{F}\right).$$

*Then the closure of $\mathcal{F}$ (for the topology of uniform convergence) is,*
$$\overline{\mathcal{F}} = \{f \in \mathcal{C}_E(X, Y) : \rho\left(\mathcal{F}\right) \subseteq \rho\left(f\right), \; \forall x \in X, \; f(x) \in \mathcal{F}(x)\},$$

*where $\mathcal{F}(x) = \{f(x), \; f \in \mathcal{F}\}$. Moreover, if $I(x) = \{(i,j) \in [p]^2 : \forall y \in \mathcal{F}(x), \; y_i = y_j\}$, then we have:*
$$\mathcal{F}(x) = \{y \in \mathbb{R}^p : \forall (i,j) \in I(x), \; y_i = y_j\}.$$

*Proof of Thm. 22.* By Lem. 28, the second assumption of Prop. 24 is also satisfied. To get the conclusion of Thm. 22, note that $\mathcal{F}(x)$ is now a linear subspace of a finite-dimensional vector space and therefore it is closed. Thus, $\overline{\mathcal{F}(x)} = \mathcal{F}(x)$ which is a subalgebra. Applying Lem. 27 to $\mathcal{F}(x)$ and noting that, necessarily $J = \emptyset$ as $\mathbf{1} \in \mathcal{F}(x)$ by assumption, gives the result of Thm. 22. $\qquad \square$

### D.8 PRACTICAL REDUCTIONS

Though the results we proved above were formulated using classic hypotheses, such as requiring $\mathcal{F}$ to be a subalgebra, we can give much more compact versions for our setting. We also reduce the assumption that $\rho\left(\mathcal{F}_{scal}\right) \subseteq \rho\left(\pi \circ \mathcal{F}\right)$ to a more practical one.

We start with the invariant case.

**Corollary 29.** *Let $X$ be a compact space, $Y = \mathbb{F} = \mathbb{R}^p$ be some finite-dimensional vector space, $G$ be a finite group acting (continuously) on $X$ and $\mathcal{F} \subseteq \mathcal{C}_I(X, Y)$ a (non-empty) set of invariant functions.*

*Assume that, for any $h \in \mathcal{C}(\mathbb{F}^2, \mathbb{F})$ and $f, g \in \mathcal{F}$,*
$$x \mapsto h(f(x), g(x)) \in \mathcal{F}.$$

*Then the closure of $\mathcal{F}$ is,*
$$\overline{\mathcal{F}} = \{f \in \mathcal{C}_I(X, Y) : \rho\left(\mathcal{F}\right) \subseteq \rho\left(f\right)\}.$$

---

[1]We slightly change the definition of $I$ compared to the statement of the lemma to add $J^2$, which does not change the result.

*Proof.* We wish to apply Thm. 22 but for this we need $G$ to act on $Y$. Define a (trivial) action of $G$ on $Y$ by,

$$\forall g \in G, \ \forall y \in Y, g \cdot y = y \,.$$

With this action on $Y$, $\mathcal{C}_E(X, Y) = \mathcal{C}_I(X, Y)$. Moreover, $Y/G = Y$, $\pi : Y \to Y/G$ is the identity so that $\rho\left(\pi \circ \mathcal{F}\right) = \rho\left(\mathcal{F}\right)$.

Our assumption clearly ensure that $\mathcal{F}$ is indeed a subalgebra and contains the constant function $\mathbb{1}$.

All that is left to show to apply Thm. 22 is that the set of functions $\mathcal{F}_{scal} \subseteq \mathcal{C}(X, \mathbb{R})$ defined by,

$$\mathcal{F}_{scal} = \{f \in \mathcal{C}(X, \mathbb{R}) : f\mathbf{1} \in \mathcal{F}\}$$

satisfies,

$$\rho\left(\mathcal{F}_{scal}\right) \subseteq \rho\left(\mathcal{F}\right) \,.$$

Take $(x, y) \notin \rho\left(\mathcal{F}\right)$ and we show that $(x, y) \notin \rho\left(\mathcal{F}_{scal}\right)$. Indeed, by definition there exists $f \in \mathcal{F}$, $i \in \{1, \ldots, p\}$ such that $f(x)_i \neq f(y)_i$. Let $l \in \mathcal{C}(\mathbb{F}, \mathbb{R})$ defined by $l(z) = z_i$ and $h \in \mathcal{C}(\mathbb{F}, \mathbb{F})$ defined by, for $z \in \mathbb{F}$, $h(z) = (l(z), \ldots, l(z)) = l(z)\mathbb{1}$. Then, by assumption $h \circ f \in \mathcal{F}$. But $h \circ f$ is $(l \circ f)\mathbf{1}$ with $l \circ f \in \mathcal{C}(X, \mathbb{R})$, so that, by definition, $l \circ f \in \mathcal{F}_{scal}$. Moreover, $l \circ f(x) \neq l \circ f(y)$. Therefore, $(x, y) \notin \rho\left(\mathcal{F}_{scal}\right)$. Therefore, we can apply Thm. 22. We get that,

$$\overline{\mathcal{F}} = \{f \in \mathcal{C}_E(X, Y) : \rho\left(\mathcal{F}\right) \subseteq \rho\left(f\right), \ \forall x \in X, \ f(x) \in \mathcal{F}(x)\} \,,$$

and

$$\mathcal{F}(x) = \{y \in \mathbb{F}^p : \forall (i, j) \in I(x), \ y_i = y_j\} \,,$$

with $I(x)$ given by,

$$I(x) = \{(i, j) \in \{1, \ldots, p\}^2 : \forall y \in \mathcal{F}(x), \ y_i = y_j\} \,.$$

To conclude the proof, we now show that $I(x) = \{(i, i) : i \in \{1, \ldots, p\}\}$, which will imply that $\mathcal{F}(x) = \mathbb{R}^p = Y$.

Indeed, the constant function $z \mapsto (1, 2, \ldots, p)$ is in $\mathcal{F}$ by assumption. Therefore, $I(x)$ is reduced to $\{(i, i) : i \in \{1, \ldots, p\}\}$. $\qquad\square$

In our previous version of our approximation result for node embedding, we did not allow features in the output as it would have made the statement and the proof a bit convoluted. With this new assumption, this is much easier.

**Corollary 30.** *Let $X$ be a compact space, $Y = \mathbb{F}^n$, with $\mathbb{F} = \mathbb{R}^p$ and $G = \mathcal{S}_n$ the permutation group, acting (continuously) on $X$ and acting on $\mathbb{F}^n$ by, for $\sigma \in \mathcal{S}_n$, $x \in \mathbb{F}^n$,*

$$\forall i \in \{1, \ldots, p\}, \ (\sigma \cdot x)_i = x_{\sigma^{-1}(i)} \,,$$

*Let $\mathcal{F} \subseteq \mathcal{C}_E(X, \mathbb{F}^n)$ be a (non-empty) set of equivariant functions.*

*Consider the following assumptions,*

*1. For any $h \in \mathcal{C}(\mathbb{F}^2, \mathbb{F})$, $f, g \in \mathcal{F}$,*

$$x \mapsto (h(f(x)_1, g(x)_1), \ldots, h(f(x)_n, g(x)_n)) \in \mathcal{F} \,.$$

*2. If $f \in \mathcal{F}$,*

$$x \mapsto \left(\sum_{i=1}^n f(x)_i, \sum_{i=1}^n f(x)_i \ldots, \sum_{i=1}^n f(x)_i\right) \in \mathcal{F} \,.$$

*Then the closure of $\mathcal{F}$ (for the topology of uniform convergence) is,*

$$\overline{\mathcal{F}} = \{f \in \mathcal{C}_E(X, \mathbb{F}^n) : \rho\left(\mathcal{F}\right) \subseteq \rho\left(f\right)\} \,.$$

For this we need a handy lemma, whose proof relies on a result about multi-symmetric polynomials from Maron et al. (2019a).

**Lemma 31.** *Let $\mathbb{F} = \mathbb{R}^p$ be some finite-dimensional space. Take $x_1, \ldots, x_n, y_1, \ldots, y_n \in \mathbb{F}$ such that, for any $\sigma \in \mathcal{S}_n$, $(x_1, \ldots, x_n) \neq (y_{\sigma(1)}, \ldots, y_{\sigma(n)})$. Then, there exists $h \in \mathcal{C}(\mathbb{F}, \mathbb{R})$ such that,*

$$\sum_{i=1}^{n} h(x_i) \neq \sum_{i=1}^{n} h(y_i) \,.$$

*Moreover, $h$ can be written as $h(x) = h_1(x_1)h_2(x_2)\ldots h_p(x_p)$ for any $x \in \mathbb{F}$ with $h_1, \ldots, h_p \in \mathcal{C}(\mathbb{R}, \mathbb{R})$.*

*Proof.* By Maron et al. (2019a, Prop. 1), there exists $\alpha_1, \ldots, \alpha_p$ non-negative integers such that $\sum_{i=1}^{n} x_{i1}^{\alpha_1} x_{i2}^{\alpha_2} \ldots x_{ip}^{\alpha_p} \neq \sum_{i=1}^{n} y_{i1}^{\alpha_1} y_{i2}^{\alpha_2} \ldots y_{ip}^{\alpha_p}$. Taking $h : x \mapsto x_{i1}^{\alpha_1} x_{i2}^{\alpha_2} \ldots x_{ip}^{\alpha_p}$ yields the result. $\qquad\square$

*Proof of Cor. 30.* We want to apply Thm. 22. With our first assumption, the first assumption of Thm. 22 is easily verified.

We now focus on the second one. As in the statement of Thm. 22, define $\mathcal{F}_{scal} \subseteq \mathcal{C}(X, \mathbb{R})$ by,

$$\mathcal{F}_{scal} = \{f \in \mathcal{C}(X, \mathbb{R}) : f\mathbf{1} \in \mathcal{F}\} \,.$$

We have to show that $\rho(\mathcal{F}_{scal}) \subseteq \rho(\pi \circ \mathcal{F})$. For this, take $x, y \notin \rho(\pi \circ \mathcal{F})$. There exists $f \in \mathcal{F}$ such that for any $\sigma \in \mathcal{S}_n$, $\sigma \cdot f(x) \neq f(y)$. We have to find $l \in \mathcal{F}_{scal}$ such that $l(x) \neq l(y)$. In other words, from a function in $\mathbb{F}^n$ which discriminates between $x$ and $y$ we have to build a function in $\mathbb{R}$.

First, we exhibit a function which discriminates between $x$ and $y$. Apply Lem. 31 to the vectors $f(x)$ and $f(y)$: there exists $h_0 \in \mathcal{C}(\mathbb{F}, \mathbb{R})$ such that $\sum_{i=1}^{n} h_0(f(x_i)) \neq \sum_{i=1}^{n} h_0(f(y_i))$.

To fit the assumptions, we build $h \in \mathcal{C}(\mathbb{F}, \mathbb{F})$ from $h_0$ by $h : x \in \mathbb{F} \mapsto (h_0(x), \ldots, h_0(x)) \in \mathbb{F}$. Take $g \in \mathcal{C}(\mathbb{F}, \mathbb{F})$ such that $g(z) = (z_1, \ldots, z_1)$ for any $z \in \mathbb{F}$ and $l \in \mathcal{C}(X, \mathbb{R})$ defined by, for $w \in X$, $l(w) = \sum_{i=1}^{n} h_0(f(w_i))$. Then, $l(x) \neq l(y)$. All we have to do is show that $l \in \mathcal{F}_{scal}$, i.e., $l\mathbf{1} \in \mathcal{F}$. This is where the two assumptions we made come into play. Indeed, the first one implies that $h \circ f \in \mathcal{F}$ and the second gives

$$z \mapsto \left( \sum_{i=1}^{n} h(f(z_i)), \ldots, \sum_{i=1}^{n} h(f(z_i)) \right) \in \mathcal{F} \,.$$

Finally, the first assumption ensure that,

$$z \mapsto \left( g\left( \sum_{i=1}^{n} h(f(z_i)) \right), \ldots, g\left( \sum_{i=1}^{n} h(f(z_i)) \right) \right) \in \mathcal{F} \,.$$

But this last function is none other than $l\mathbf{1}$, which shows that $l \in \mathcal{F}_{scal}$ as required.

We have successfully verified the hypothesis of Thm. 22. Therefore, the closure of $\mathcal{F}$ is,

$$\overline{\mathcal{F}} = \{f \in \mathcal{C}_E(X, \mathbb{F}^n) : \rho(\mathcal{F}) \subseteq \rho(f), \ \forall x \in X, \ f(x) \in \mathcal{F}(x)\} \,,$$

with

$$\mathcal{F}(x) = \{y \in \mathbb{F}^n : \forall (i, i', j, j') \in I(x), \ y_{i,i'} = y_{j,j'}\} \,,$$

and

$$I(x) = \{(i, i', j, j') \in (\{1, \ldots, n\} \times \{1, \ldots, p\})^2 : \forall y \in \mathcal{F}(x), \ y_{i,i'} = y_{j,j'}\} \,.$$

To get the desired result, we need to get rid of the condition "$f(x) \in \mathcal{F}(x)$" in the description of $\overline{\mathcal{F}}$. Fix $x \in X$.

First, we show that

$$\mathcal{F}(x) = \{y \in \mathbb{F}^n : \forall (i, j) \in J(x), \ y_i = y_j\} \,,$$

with

$$J(x) = \{(i, j) \in \{1, \ldots, n\}^2 : y_i = y_j\} \,,$$

(note that the equalities here are not in $\mathbb{R}$ anymore but in $\mathbb{F}$). The direct inclusion "$\subseteq$" is immediate by construction of $J(x)$ so we focus on the reverse direction. For this, we show that the 4-tuples $(i, i', j, j')$ of $I(x)$ necessarily satisfy $i' = j'$ and $(i, j) \in J(x)$.

First note that, by the first assumption, the vector $y^0 \in \mathbb{F}^n$ such that $y_i^0 = (1, 2, \ldots, p)$ for $i \in \{1, \ldots, n\}$ is in $\mathcal{F}(x)$. Indeed, take the constant function always equal to $(1, 2, \ldots, p)$ as $h$. Now, consider a 4-tuple $(i, i', j, j')$ of $I(x)$ and we show that, actually, $(i, j) \in J(x)$ and $i' = j'$. As $y^0$ in $\mathcal{F}(x)$, and $y_{i,i'}^0 = i'$, $y_{j,j'}^0 = j'$, $(i, i', j, j') \in I(x)$ implies that $j' = i'$. Consider, $k \in \{1, \ldots, p\}$. We show that, for any $y \in \mathcal{F}(x)$, $y_{i,k} = y_{j,k}$. But such a $y$ can be written as $y = f(x)$ for some $f \in \mathcal{F}$. Consider, the function $h \in \mathcal{C}(\mathbb{F}, \mathbb{F})$ associated to the permutation $(i' \ k)$, defined by,

$$z \mapsto (z_{(i' \ k)(1)}, \ldots, z_{(i' \ k)(n)}) = (z_1, \ldots, z_{i'-1}, z_k, z_{i'+1}, \ldots, z_{k-1}, z_{i'}, z_{k+1}, \ldots, z_p).$$

By our first assumption, $z \mapsto (h(f(z)_1), \ldots, h(f(z)_n)) \in \mathcal{F}$ so that $(h(y_1), \ldots, h(y_n)) \in \mathcal{F}(x)$. In particular, as $(i, i', j, i') \in I(x)$, $h(y_i)_{i'} = h(y_j)_{i'}$, i.e. $y_{i,k} = y_{j,k}$. Therefore, $(i, j) \in J(x)$.

Finally, we can conclude that $\mathcal{F}(x) \supset \{y \in \mathbb{F}^n : \forall (i, j) \in J(x), \ y_i = y_j\}$. Indeed, take $y \in \mathbb{F}^n : \forall (i, j) \in J(x), \ y_i = y_j$. We show that all the constraints of $I(x)$ are satisfied. Indeed, take $(i, i', j, j') \in I(x)$. We have shown that $(i, j) \in J(x)$ and $i' = j'$ so that $y_i = y_j$ and in particular $y_{i,i'} = y_{j,j'}$. Therefore, this finishes the proof of $\mathcal{F}(x) \supset \{y \in \mathbb{F}^n : \forall (i, j) \in J(x), \ y_i = y_j\}$.

Thus,

$$\mathcal{F}(x) = \{y \in \mathbb{F}^n : \forall (i, j) \in J(x), \ y_i = y_j\},$$

with

$$J(x) = \{(i, j) \in \{1, \ldots, n\}^2 : y_i = y_j\}.$$

We have proven so far, that,

$$\overline{\mathcal{F}} = \{f \in \mathcal{C}_E(X, \mathbb{F}^n) : \rho(\mathcal{F}) \subseteq \rho(f), \ \forall x \in X, \ f(x) \in \mathcal{F}(x)\} \subseteq \{f \in \mathcal{C}_E(X, \mathbb{F}^n) : \rho(\mathcal{F}) \subseteq \rho(f)\}.$$

Take $h \in \mathcal{C}_E(X, \mathbb{F}^n)$ such that $\rho(\mathcal{F}) \subseteq \rho(h)$ and fix $x \in X$. Our goal is to show that, for any $(i, j) \in J(x)$, $h(x)_i = h(x)_j$ so that $h(x) \in \mathcal{F}(x)$. But, if $(i, j) \in J(x)$, then for any $f \in \mathcal{F}$, $f(x)_i = f(x)_j$ so that $(i \ j) \cdot f(x) = f(x)$, where $(i \ j)$ denotes the permutation which exchanges $i$ and $j$. Moreover, as $(i \ j) \in \mathcal{S}_n$, by equivariance, this means that $f((i \ j) \cdot x) = f(x)$ for every $f \in \mathcal{F}$ and therefore that $((i \ j) \cdot x, x) \in \rho(\mathcal{F})$. By assumption, we infer that $((i \ j) \cdot x, x) \in \rho(h)$ too, i.e. that $h((i \ j) \cdot x) = h(x)$ and so that $h(x)_i = h(x)_j$ by equivariance, which concludes our proof.

$\square$

## D.9 REDUCTIONS FOR GNNS

We now present a lemma which explains how to instantiate the two corollaries above in the case of GNNs by replacing continuous functions with MLPs.

**Lemma 32.** *Fix $X$ some compact space, $n \geq 1$ and $\mathbb{F}$ a finite-dimensional feature space. Let $\mathcal{F}_0 \subseteq \bigcup_{h=1}^{\infty} \mathcal{C}(X, \mathbb{R}^h)$ be stable by concatenation and consider,*

$$\mathcal{F} = \{x \mapsto (m(f(x)_1), \ldots, m(f(x)_n)) : f \in \mathcal{F}_0 \cap \mathcal{C}(X, \mathbb{R}^h), \ m : \mathbb{R}^h \to \mathbb{F} \ MLP, \ h \geq 1\} \subseteq \mathcal{C}(X, \mathbb{F}).$$

*Then, if $\mathcal{E}(\mathcal{F}) \subseteq \mathcal{C}(X, \mathbb{F})$ is the set of functions obtained by replacing the MLP $m$ in the definition of $\mathcal{F}$ by an arbitrary continuous function, $\mathcal{E}(\mathcal{F})$ satisfies,*

*1. $\overline{\mathcal{F}} = \overline{\mathcal{E}(\mathcal{F})}$*

*2. $\rho(\mathcal{F}) = \rho(\mathcal{E}(\mathcal{F}))$*

*3. For any $h \in \mathcal{C}(\mathbb{F}^2, \mathbb{F})$, $f, g \in \mathcal{E}(\mathcal{F})$,*

$$x \mapsto (h(f(x)_1, g(x)_1), \ldots, h(f(x)_n, g(x)_n)) \in \mathcal{E}(\mathcal{F}).$$

*4. If, for any $f \in \mathcal{F}$,*

$$x \mapsto \left(\sum_{i=1}^n f(x)_i, \sum_{i=1}^n f(x)_i \ldots, \sum_{i=1}^n f(x)_i\right) \in \mathcal{F},$$

*then, for any $f \in \mathcal{E}(\mathcal{F})$,*

$$x \mapsto \left(\sum_{i=1}^n f(x)_i, \sum_{i=1}^n f(x)_i \ldots, \sum_{i=1}^n f(x)_i\right) \in \mathcal{E}(\mathcal{F}).$$

5. *If $\mathcal{F}$ is equivariant w.r.t. the action described in Cor. 30, so is $\mathcal{E}(\mathcal{F})$.*

*Proof.* Define,

$$\mathcal{E}(\mathcal{F}) = \{x \mapsto (m(f(x)_1), \ldots, m(f(x)_n)) : f \in \mathcal{F}_0 \cap \mathcal{C}(X, \mathbb{R}^h), \ m \in \mathcal{C}(\mathbb{R}^h, \mathbb{F}), \ h \geq 1\}.$$

As MLP are continuous and by the universality of MLP on a compact set (see Section C.1),

$$\mathcal{F} \subseteq \mathcal{E}(\mathcal{F}) \subseteq \overline{\mathcal{F}}.$$

This already implies 1. and that $\rho\left(\overline{\mathcal{F}}\right) \subseteq \rho\left(\mathcal{E}(\mathcal{F})\right) \subseteq \rho\left(\mathcal{F}\right)$. Using Lem. 15 yields 2.

We now show 3. Take $h \in \mathcal{C}(\mathbb{F}^2, \mathbb{F})$, $f, g \in \mathcal{F}_0$, $f \in \mathcal{C}(X, \mathbb{R}^{h_f})$, $g \in \mathcal{C}(X, \mathbb{R}^{h_g})$ and $m \in \mathcal{C}(\mathbb{R}^{h_f}, \mathbb{F})$, $l \in \mathcal{C}(\mathbb{R}^{h_g}, \mathbb{F})$. All we have to show is that,

$$x \mapsto (h(m(f(x)_1), l(g(x)_1)), \ldots, h(m(f(x)_n), l(g(x)_n))) \in \mathcal{E}(\mathcal{F}).$$

But as $\mathcal{F}_0$ is stable by concatenation, $x \mapsto (f(x), g(x)) \in \mathbb{R}^{h_f + h_g}$ is still in $\mathcal{F}_0$. Moreover, $y \in \mathbb{R}^{h_f + h_g} \mapsto h(m(y_1, \ldots, y_{h_f}), l(y_{h_f+1}, \ldots, y_{h_f+h_g})$ is also in $\mathcal{C}(\mathbb{R}^{h_f + h_g}, \mathbb{F})$ which shows that the map above is indeed in $\mathcal{E}(\mathcal{F})$. The last two points are immediate consequences of the definition of $\mathcal{E}(\mathcal{F})$. $\qquad\square$

Thm. 5 and Thm. 6 are now obtained by combining Lem. 32 with Cor. 29 and Cor. 30.

## E  PROOFS FOR EXPRESSIVENESS OF GNNS

Note that in the Theorem 6, the additional stability assumption is "almost" necessary to obtain the result. Indeed, if the result holds, i.e.,

$$\overline{\mathcal{F}} = \{f \in \mathcal{C}_E(X, \mathbb{F}^n) : \rho\left(\mathcal{F}_0\right) \subseteq \rho\left(f\right)\},$$

then one can show, that, if $f \in \mathcal{F}$, then

$$\tilde{f} : x \mapsto \left(\sum_{i=1}^n f(x)_i, \sum_{i=1}^n f(x)_i \ldots, \sum_{i=1}^n f(x)_i\right) \in \overline{\mathcal{F}}.$$

Indeed, $f \in \mathcal{F}$ so that it has a weaker discriminating power than $\mathcal{F}$, so that $\rho\left(\mathcal{F}\right) = \rho\left(\mathcal{F}_0\right) \subseteq \rho\left(f\right)$. But, by construction of $\tilde{f}$, $\rho\left(f\right) \subseteq \rho\left(\tilde{f}\right)$ so that $\rho\left(\mathcal{F}_0\right) \subseteq \rho\left(\tilde{f}\right)$. As $\tilde{f}$ is also in $\mathcal{C}_E(X, \mathbb{F}^n)$, $\tilde{f} \in \overline{\mathcal{F}}$.

### E.1  EXPRESSIVITY OF GNN LAYERS

**Lemma 33.** *Fix $\mathbb{F}_0, \mathbb{F}_1$ (non-trivial) finite dimensionnal vector spaces. Consider the action of $G = \mathcal{S}_n$ on $\mathbb{F}_0 \times \mathbb{F}_0^{n \times k}$ defined by,*

$$\forall \sigma \in \mathcal{S}_n, \ \forall x^0 \in \mathbb{F}_0, \ \forall x \in \mathbb{F}^{n \times k}, \ \sigma \cdot (x^0, x) = (x^0, x_{\sigma^{-1}(1)}, \ldots, x_{\sigma^{-1}(n)})$$

*Let $K \subseteq \mathbb{F}_0 \times \mathbb{F}_0^{n \times k}$ be a compact set. Then, the set of functions from $K \subseteq \mathbb{F}_0 \times \mathbb{F}_0^{n \times k}$ to $\mathbb{F}_1$ of the form,*

$$(x^0, x) \longmapsto f_0\left(x^0, \sum_{j=1}^n \prod_{w=1}^k f_w(x_{j,w})\right)$$

*where $f_0 : \mathbb{F}_0 \times \mathbb{R}^h \to \mathbb{F}_1$, $f_j : \mathbb{F}_0 \to \mathbb{R}^h$, $j = 1, \ldots, k$ are multi-linear perceptrons and $h \geq 1$, is dense in $\mathcal{C}_I(K, \mathbb{F}_1)$.*

*Proof.* Denote by $\mathcal{F} \subseteq \mathcal{C}_I(K, \mathbb{F}_1)$ the set of such functions. To prove that $\overline{\mathcal{F}} = \mathcal{C}_I(K, \mathbb{F}_1)$, we first apply Thm. 5.

We get, that,

$$\overline{\mathcal{F}} = \{f \in \mathcal{C}_I(\mathbb{F}_0 \times \mathbb{F}_0^{n \times k}, \mathbb{F}_1) : \rho\left(\mathcal{F}\right) \subseteq \rho\left(f\right)\}.$$

We now characterize $\rho(\mathcal{F})$. Actually, it is equal to $\rho(inv) = \{((x^0, x), (y^0, y)) \in (\mathbb{F}_0 \times \mathbb{F}_0^{n \times k})^2 : \exists \sigma \in \mathcal{S}_n, (x^0, x) = \sigma \cdot (y^0, y)\}$. As the functions of $\mathcal{F}$ are invariant, $\rho(inv) \subseteq \rho(\mathcal{F})$. We now show the reverse. Take $(x^0, x), (y^0, y) \in \mathbb{F}_0 \times \mathbb{F}_0^{n \times k}$ such that there does not exist $\sigma \in \mathcal{S}_n$ such that $(x^0, x) = \sigma \cdot (y^0, y)$. If $x^0 \neq y^0$, there exists $f_0 : \mathbb{F}_0 \to \mathbb{F}_1$ MLP such that $f(x^0) \neq f(y^0)$ so that $((x^0, x), (y^0, y)) \notin \rho(\mathcal{F})$.

Otherwise, there does not exists $\sigma \in \mathcal{S}_n$ such that $(x_{\sigma^{-1}(1)}, \ldots, x_{\sigma^{-1}(n)}) = (y_1, \ldots, y_n)$ by definition of the action of $G = \mathcal{S}_n$. Now, apply Lem. 31 with $\mathbb{F} \leftarrow \mathbb{F}_0^k$, the universality of MLP and use the decomposition given to get $f_j : \mathbb{F}_0 \to \mathbb{R}^h$, $j = 1, \ldots, k$ such that $\sum_{j=1}^n \prod_{w=1}^k f_w(x_{j,w}) \neq \sum_{j=1}^n \prod_{w=1}^k f_w(y_{j,w})$. Choosing an appropriate MLP $f_0 : \mathbb{R}^h \to \mathbb{F}_1$ yields that $((x^0, x), (y^0, y)) \notin \rho(\mathcal{F})$.

Hence, we have shown that,

$$\overline{\mathcal{F}} = \{f \in \mathcal{C}_I(\mathbb{F}_0 \times \mathbb{F}_0^{n \times k}, \mathbb{F}_1) : \rho(inv) \subseteq \rho(f)\} = \mathcal{C}_I(\mathbb{F}_0 \times \mathbb{F}_0^{n \times k}, \mathbb{F}_1).$$

$\square$

### E.2 APPROXIMATION THEOREMS FOR GNNS

We now have all the tools to finally prove our main result.

**Theorem 34.** *Let $K_{discr} \subseteq \mathcal{G}_n \times \mathbb{F}_0^n$, $K \subseteq \mathbb{F}_0^{n^2}$ be compact sets. For the invariant case, we have:*

$$\overline{MGNN_I} = \{f \in \mathcal{C}_I(K_{discr}, \mathbb{F}) : \rho(2\text{-}WL_I) \subseteq \rho(f)\}$$

$$\overline{2\text{-}LGNN_I} = \{f \in \mathcal{C}_I(K, \mathbb{F}) : \rho(2\text{-}WL_I) \subseteq \rho(f)\}$$

$$\overline{k\text{-}LGNN_I} = \{f \in \mathcal{C}_I(K, \mathbb{F}) : \rho(k\text{-}LGNN_I) \subseteq \rho(f)\} \supset \{f \in \mathcal{C}_I(K, \mathbb{F}) : \rho(k\text{-}WL_I) \subseteq \rho(f)\}$$

$$\overline{k\text{-}FGNN_I} = \{f \in \mathcal{C}_I(K, \mathbb{F}) : \rho(k\text{-}FWL_I) \subseteq \rho(f)\}$$

*For the equivariant case, we have:*

$$\overline{MGNN_E} = \{f \in \mathcal{C}_E(K_{discr}, \mathbb{F}^n) : \rho(2\text{-}WL_E) \subseteq \rho(f)\}$$

$$\overline{k\text{-}LGNN_E} = \{f \in \mathcal{C}_E(K, \mathbb{F}^n) : \rho(k\text{-}LGNN_E) \subseteq \rho(f)\} \supset \{f \in \mathcal{C}_E(K, \mathbb{F}^n) : \rho(k\text{-}WL_E) \subseteq \rho(f)\}$$

$$\overline{k\text{-}FGNN_E} = \{f \in \mathcal{C}_E(K, \mathbb{F}^n) : \rho(k\text{-}FWL_E) \subseteq \rho(f)\}$$

We decompose the proof with an additional lemma.

**Lemma 35.** *Let $K_{discr} \subseteq \mathcal{G}_n \times \mathbb{F}_0^n$, $K \subseteq \mathbb{F}_0^{n^2}$ be compact sets. For the invariant case, and any $k \geq 2$, we have,*

$$\overline{MGNN_I} = \{f \in \mathcal{C}_I(K_{discr}, \mathbb{F}) : \rho(MGNN_I) \subseteq \rho(f)\}$$

$$\overline{k\text{-}LGNN_I} = \{f \in \mathcal{C}_I(K, \mathbb{F}) : \rho(k\text{-}LGNN_I) \subseteq \rho(f)\}$$

$$\overline{k\text{-}FGNN_I} = \{f \in \mathcal{C}_I(K, \mathbb{F}) : \rho(k\text{-}FGNN_I) \subseteq \rho(f)\}$$

*For the equivariant case, and any $k \geq 2$, we have:*

$$\overline{MGNN_E} = \{f \in \mathcal{C}_E(K_{discr}, \mathbb{F}) : \rho(MGNN_E) \subseteq \rho(f)\}$$

$$\overline{k\text{-}LGNN_E} = \{f \in \mathcal{C}_E(K, \mathbb{F}) : \rho(k\text{-}LGNN_E) \subseteq \rho(f)\}$$

$$\overline{k\text{-}FGNN_E} = \{f \in \mathcal{C}_E(K, \mathbb{F}) : \rho(k\text{-}FGNN_E) \subseteq \rho(f)\}$$

*Proof of Thm. 34.* The theorem is now a direct consequence of Prop. 13 and Lem. 35. $\square$

We now move to the proof of Lem. 35.

*Proof of Lem. 35.* First, focus on the invariant case. Let $\mathcal{F}$ denote $MGNN_I$, $k\text{-}LGNN_I$ or $k\text{-}FGNN_I$ and $X$ be either $K_{discr}$ or $K$ so that $X$ is compact and $\mathcal{F} \subseteq \mathcal{C}_I(X, \mathbb{F})$. Applying Thm. 5 directly gives,

$$\overline{\mathcal{F}} = \{f \in \mathcal{C}_I(X, \mathbb{F}) : \rho(\mathcal{F}) \subseteq \rho(f)\},$$

which is the desired result.

We now move to the equivariant case. First, let us replace $\mathrm{MGNN}_E$ by another class, which is slightly simpler to analyze. Define,

$$\mathrm{MGNN}_{E'} = \{m_E \circ ((1-\lambda)\,\mathrm{Id} + \lambda S^1) \circ F_T \circ \dots F_2 \circ F_1 : F_t :$$
$$\mathbb{F}_t^n \to \mathbb{F}_{t+1}^n \text{ message passing layer}, t = 1, \dots, T, \ T \geq 1, \lambda \in \{0, 1\}\}.$$

It holds that $\rho\left(\mathrm{MGNN}_{E'}\right) = \rho\left(\mathrm{MGNN}_E\right)$ and,

$$\mathrm{MGNN}_{E'} \subseteq \overline{\mathrm{MGNN}_E} \subseteq \{f \in \mathcal{C}_E(K_{discr}, \mathbb{F}^n) : \ \rho\left(\mathrm{MGNN}_E\right) \subseteq \rho\left(f\right)\}$$
$$= \{f \in \mathcal{C}_E(K_{discr}, \mathbb{F}^n) : \ \rho\left(\mathrm{MGNN}_{E'}\right) \subseteq \rho\left(f\right)\}$$

Therefore, if we show that,

$$\overline{\mathrm{MGNN}_{E'}} = \{f \in \mathcal{C}_E(K_{discr}, \mathbb{F}^n) : \ \rho\left(\mathrm{MGNN}_{E'}\right) \subseteq \rho\left(f\right)\},$$

we will have the desired result.

Let $\mathcal{F}$ denote $\mathrm{MGNN}_{E'}$, $k$-$\mathrm{LGNN}_E$ or $k$-$\mathrm{FGNN}_E$ and $X$ be either $K_{discr}$ or $K$ so that $X$ is compact and $\mathcal{F} \subseteq \mathcal{C}_E(X, \mathbb{F}^n)$. We now wish to apply Thm. 6, but we need to verify the stability assumption first. Thus, we show that, for any $f \in \mathcal{F}$,

$$x \mapsto \left(\sum_{i=1}^n f(x)_i, \sum_{i=1}^n f(x)_i \dots, \sum_{i=1}^n f(x)_i\right) \in \mathcal{F},$$

- If $\mathcal{F} = \mathrm{MGNN}_{E'}$. Take $f \in \mathrm{MGNN}_{E'}$. $f$ is of the form,

$$m_E \circ ((1-\lambda)\,\mathrm{Id} + \lambda S^1) \circ F_T \circ \dots F_2 \circ F_1,$$

where $F_t : \mathbb{F}_t^n \to \mathbb{F}_{t+1}^n$ are message passing layers, $\mathbb{F}_{T+1} = \mathbb{F}$ and $\lambda \in \{0, 1\}$. We need to show that there is a $\mathrm{MGNN}_{E'}$ which implements,

$$x \mapsto \left(\sum_{i=1}^n f(x)_i, \sum_{i=1}^n f(x)_i \dots, \sum_{i=1}^n f(x)_i\right). \tag{26}$$

If $\lambda = 1$, $f$ is exactly the function of (26). Otherwise, if $\lambda = 0$, we build another $\mathrm{MGNN}_{E'}$ implementing this function. Denote by $F_{T+1} : \mathbb{F}^n \to \mathbb{F}^n$ the (simple) message passing layer defined by $F_{T+1}(h)_i = m_E(h_i)$ for $i \in [n]$ and any $h \in \mathbb{F}^n$. Then, the $\mathrm{MGNN}_{E'}$ $(1-\lambda')\,\mathrm{Id} + \lambda' S^1) \circ F_{T+1} \circ F_T \circ \dots F_2 \circ F_1$ with $\lambda' = 1$ exactly implements (26).

- If $\mathcal{F} = k$-$\mathrm{LGNN}_E$. A function $f$ of this class is of the form,

$$m_E \circ S_1^k \circ F_T \circ \dots F_2 \circ F_1 \circ I^k,$$

where $F_t : \mathbb{F}_t^{n^k} \to \mathbb{F}_{t+1}^{n^k}$ are linear graph layers and $\mathbb{F}_{T+1} = \mathbb{F}$. Our goal is to show that there is a GNN of $k$-$\mathrm{LGNN}_E$ which implements,

$$x \mapsto \left(\sum_{i=1}^n f(x)_i, \sum_{i=1}^n f(x)_i \dots, \sum_{i=1}^n f(x)_i\right), \tag{27}$$

By definition of linear graph layers, the map $F_{T+1} : \mathbb{F}^{n^k} \to \mathbb{F}^{n^k}$ defined by, for $G \in \mathbb{F}^{n^k}$,

$$\forall (i_1, \dots, i_k) \in [n]^k, \ F_{T+1}(G)_{i_1, \dots, i_k} = m_E(S_1^k(G)_i),$$

is a linear graph layer as defined in Section 3. Now, consider, the linear graph layer, $F_{T+2} : \mathbb{F}^{n^k} \to \mathbb{F}^{n^k}$ defined by, for $G \in \mathbb{F}^{n^k}$,

$$\forall (i_1, \dots, i_k) \in [n]^k, \ F_{T+1}(G)_{i_1, \dots, i_k} = \sum_{i=1}^n G_{i, i_2, \dots, i_k}.$$

Then, the $k$-$\mathrm{LGNN}$ $F_{T+2} \circ F_{T+1} \circ F_T \circ \dots F_2 \circ F_1 \circ I^k$ exactly implements (27).

- If $\mathcal{F} = k\text{-FGNN}_E$. A function $f$ of this class is of the form,

$$m_E \circ S_1^k \circ F_T \circ \ldots F_2 \circ F_1 \circ I^k,$$

where $F_t : \mathbb{F}_t^{n^k} \to \mathbb{F}_{t+1}^{n^k}$ are FGL (see Section 3) and $\mathbb{F}_{T+1} = \mathbb{F}$. We build a GNN of $k\text{-MGNN}_E$ which implements,

$$x \mapsto \left( \sum_{i=1}^n f(x)_i, \sum_{i=1}^n f(x)_i \ldots, \sum_{i=1}^n f(x)_i \right), \tag{28}$$

For $w \in [k]$, define the FGL $H_w : \mathbb{F}^{n^k} \to \mathbb{F}^{n^k}$ by,

$$\forall G \in \mathbb{F}^{n^k}, \ \forall (i_1, \ldots, i_k) \in [n]^k, \ H_w(G)_{i_1, \ldots, i_k} = \sum_{j=1}^n G_{i_1, \ldots, i_{w-1}, j, i_{w+1}, \ldots, i_k}.$$

Then $H_2 \circ \cdots \circ H_k : \mathbb{F}^{n^k} \to \mathbb{F}^{n^k}$ computes the sum of the elements of the input tensor over the last $k-1$ dimensions like to $S_1^k : \mathbb{F}^{n^k} \to \mathbb{F}^n$ and $H_1 \circ H_2 \circ \cdots \circ H_k : \mathbb{F}^{n^k} \to \mathbb{F}^{n^k}$ computes the full sum of the elements of the input tensor like to $S^k : \mathbb{F}^{n^k} \to \mathbb{F}$. Finally, consider the FGL $F_{T+1} : \mathbb{F}^{n^k} \to \mathbb{F}^{n^k}$ associated to $m_E$, i.e. such that, for any $G \in \mathbb{F}^{n^k}$,

$$\forall (i_1, \ldots, i_k) \in [n]^k, \ F_{T+1}(G)_{i_1, \ldots, i_k} = m_E(G_{i_1, \ldots, i_k}).$$

Now, the $k\text{-FGNN}$

$$H_1 \circ H_2 \circ \cdots \circ H_k \circ F_{T+1} \circ H_2 \circ \cdots \circ H_k \circ F_{T+1} \circ F_T \circ \ldots F_2 \circ F_1 \circ I^k$$

exactly implements (28).

$\square$

## F EXTENSION TO GRAPHS OF VARYING SIZES

### F.1 EXTENSION TO DISCONNECTED INPUT SPACES

Here we show that our results can be extended to graphs of varying sizes similarly to Keriven & Peyré (2019). There are two ways to do it. The first would be to directly adapt all the proofs but it would make them more cumbersome. Instead, we extend them with a simple argument presented below. As a side benefit, this general lemma makes it possible to extend almost all the approximation results for graph neural networks from the literature.

The abstract setting is the following. Given a compact input space $X$, assume that there is some finite set $A$, and $X_\alpha$, $\alpha \in A$ a family of pairwise disjoints compact sets such that $X = \bigsqcup_{\alpha \in A} X_\alpha$. Crucially, we will assume that the $X_\alpha$ are in distinct connected components. Intuitively, they do not "touch" each other. Similarly, assume that the output space $Y$ can be written as $Y = \bigsqcup_{\alpha \in A} Y_\alpha$, with $Y_\alpha$, $\alpha \in A$ a family of (pairwise disjoints) real vector spaces.

Before moving to the results, we need a last definition. Informally, we need that the functions $f$ that we will consider do not change the number of nodes of their inputs. Formally, we say that $f : X \longrightarrow Y$ is *adapted* if $f(X_\alpha) \subseteq Y_\alpha$ for each $\alpha \in A$, and denote by $\mathcal{C}_{ad}(X, Y)$ the of continuous and adapted functions from $X$ to $Y$.

We can now state our lemma to adapt our results to graphs with varying node sizes.

**Lemma 36.** *Let $X$ be a compact space, $Y$ be a topological space and $A$ a finite set. Assume that there exists $X_\alpha$, $\alpha \in A$ a family of pairwise disjoints compact sets such that $X = \bigsqcup_{\alpha \in A} X_\alpha$ and the $X_\alpha$'s are in distinct connected components. Also assume that there is $Y_\alpha$, $\alpha \in A$ a family of (pairwise disjoints) real vector spaces such that $Y = \bigsqcup_{\alpha \in A} Y_\alpha$. Consider $\mathcal{F} \subseteq \mathcal{C}_{ad}(X, Y)$ a (non-empty) set of adapted functions and, for each $\alpha \in A$, define $\mathcal{F}_{|X_\alpha} = \{ f_{|X_\alpha} : f \in \mathcal{F} \} \subseteq \mathcal{C}(X_\alpha, Y_\alpha)$ the restriction of the functions of $\mathcal{F}$ to $X_\alpha$.*

*Assume that the following holds,*

1. *Each $\mathcal{F}_{|X_\alpha}$ is a sub-algebra of $\mathcal{C}(X_\alpha, Y_\alpha)$ which contains the constant function $\mathbf{1}_{Y_\alpha}$.*

2. *There is a set of functions $\mathcal{F}_{scal} \subseteq \mathcal{C}(X, \mathbb{R})$ such that $\mathcal{F}_{scal}.\mathcal{F} \subseteq \mathcal{F}$ and it discriminates between the $X_\alpha$'s, i.e. $\rho(\mathcal{F}_{scal}) \subseteq \bigsqcup_{\alpha \in A} X_\alpha^2$.*

*Then the closure of $\mathcal{F}$ is,*

$$\overline{\mathcal{F}} = \left\{ f \in \mathcal{C}_{ad}(X, Y) : \forall \alpha \in A, \ f_{|X_\alpha} \in \overline{\mathcal{F}_{|X_\alpha}} \right\},$$

*for the distance on $\mathcal{C}_{ad}(X, Y)$ defined by, for $f, g \in \mathcal{C}_{ad}(X, Y)$,*

$$d(f, g) = \max_{\alpha \in A} \sup_{X_\alpha} \|f - g\|_{Y_\alpha}.$$

*Proof.* Define the set of functions $S \subseteq \mathcal{C}(X, \mathbb{R})$ by,

$$S = \{ f \in \mathcal{C}(X, \mathbb{R}) : f.\mathcal{F} \subseteq \mathcal{F} \}.$$

By definition, $\mathcal{F}_{scal} \subseteq S$. Moreover, as each $\mathcal{F}_{|X_\alpha}$ is a subalgebra, $S$ is also a subalgebra of $\mathcal{C}(X, \mathbb{R})$.

As $X$ is compact, we can apply Cor. 17 to $S$ to get that, in $\mathcal{C}(X, \mathbb{R})$,

$$\overline{S} = \{ f \in \mathcal{C}(X, \mathbb{R}) : \rho(S) \subseteq \rho(f) \}.$$

In the first part of the proof, we check that, for each $\alpha$, the function $f_\alpha : X \longrightarrow [0, 1]$ defined by

$$f_{\alpha|X_\alpha} = 1$$
$$\forall \beta \neq \alpha, \ f_{\alpha|X_\beta} = 0.$$

is continuous and satisfy $\rho(S) \subseteq \rho(f_\alpha)$. This will means that such functions belong to $\overline{\mathcal{F}_{scal}}$.

As the $X_\beta$'s are in different connected components, it is enough to check its continuity on each $X_\beta$. Indeed, each $f_{\alpha|X_\beta}$ is constant so continuous. The second fact comes from the second assumption. Indeed, take $(x, y) \in X$ such that $(x, y) \in \rho(S)$. As $\mathcal{F}_{scal} \subseteq S$, in particular $(x, y) \in \rho(\mathcal{F}_{scal})$. But, by assumption, $\rho(\mathcal{F}_{scal}) \subseteq \bigsqcup_{\beta \in A} X_\beta^2$. Thus, necessarily $x$ and $y$ belong to the same $X_\beta$ so that $f_\alpha(x) = f_\alpha(y)$.

Therefore, we conclude that $f_\alpha \in \overline{S}$.

We now prove the announced equality. By the definition of the distance, the first inclusion, $\overline{\mathcal{F}} \subseteq \left\{ f \in \mathcal{C}_{ad}(X, Y) : f_{|X_\alpha} \in \overline{\mathcal{F}_\alpha} \right\}$ is immediate. We focus on the other way. Take $h \in \mathcal{C}_{ad}(X, Y)$ such that $h_{|X_\alpha} \in \overline{\mathcal{F}_\alpha}$ for every $\alpha \in A$ and $\epsilon > 0$. By definition of $h$, there exists, for each $\alpha \in A$, $g_\alpha \in \mathcal{F}$ such that,

$$\sup_{X_\alpha} \|h - g_\alpha\|_{Y_\alpha} \leq \epsilon,$$

As the $X_\alpha$'s are compact and the $g_\alpha$'s are continuous, $\max_{\alpha \in A} \sup_{X_\alpha} \|g_\alpha\| < +\infty$ and denote by $M > 0$ a bound on this quantity. We have shown above that each $f_\alpha$ is in $\overline{S}$ so there exists, for each $\alpha \in A$, $l_\alpha \in S$ such that,

$$\sup_{X} |f_\alpha - l_\alpha| \leq \frac{\epsilon}{M}.$$

By definition of $S$ and, as $\mathcal{F}$ is a subalgebra, $\sum_{\alpha \in A} l_\alpha g_\alpha \in \mathcal{F}$ and, for each $\beta \in A$,

$$\sup_{X_\beta} \left\| h - \sum_{\alpha \in A} l_\alpha g_\alpha \in \mathcal{F} \right\| \leq \sup_{X_\beta} \|h - g_\beta\| + \|g_\beta - l_\beta g_\beta\| + \left\| \sum_{\alpha \neq \beta} l_\alpha g_\alpha \right\|$$

$$\leq \epsilon + M \times \frac{\epsilon}{M} + (|A| - 1)M \times \frac{\epsilon}{M} = (|A| + 1)\epsilon,$$

which concludes the proof. $\qquad\square$

### F.2 APPROXIMATION THEOREM WITH VARYING GRAPH SIZE

We now state our theorem in the case of varying graph size like Keriven & Peyré (2019). With Lem. 36, it is indeed straightforward to extend any approximation result initially proven for a class of graphs of fixed size. However, as the complete proof would require new notations again, we only give a sketch of proof.

Fix $N \geq 1$ and consider the space of graphs (described by tensors) of size less than $N$, $\mathbb{F}_0^{\leq N^2} = \bigcup_{n=1}^{N} \mathbb{F}_0^{n^2}$. Equip this space with the final topology or, equivalently, the graph edit distance. Then, the last thing to check to apply Lem. 36 is that the classes of GNN that we consider indeed discriminate between graphs of different sizes, which is immediate.

Likewise, for message passing GNN, consider $\mathcal{G}_{\leq N} \times \mathbb{F}_0^{\leq N} = \bigcup_{n=1}^{N} \mathcal{G}_n \times \mathbb{F}_0^n$, with a similar topology or the graph edit distance.

**Corollary 37.** *Let $K_{discr} \subseteq \mathcal{G}_{\leq N} \times \mathbb{F}_0^{\leq N}$, $K \subseteq \mathbb{F}_0^{\leq N^2}$ be compact sets. For the invariant case, we have:*

$$\overline{MGNN_I} = \{f \in \mathcal{C}_I(K_{discr}, \mathbb{F}) : \rho(2\text{-}WL_I) \subseteq \rho(f)\}$$

$$\overline{2\text{-}LGNN_I} = \{f \in \mathcal{C}_I(K, \mathbb{F}) : \rho(2\text{-}WL_I) \subseteq \rho(f)\}$$

$$\overline{k\text{-}LGNN_I} = \{f \in \mathcal{C}_I(K, \mathbb{F}) : \rho(k\text{-}LGNN_I) \subseteq \rho(f)\} \supset \{f \in \mathcal{C}_I(K, \mathbb{F}) : \rho(k\text{-}WL_I) \subseteq \rho(f)\}$$

$$\overline{k\text{-}FGNN_I} = \{f \in \mathcal{C}_I(K, \mathbb{F}) : \rho(k\text{-}FWL_I) \subseteq \rho(f)\}$$

*For the equivariant case, we have:*

$$\overline{MGNN_E} = \{f \in \mathcal{C}_E(K_{discr}, \mathbb{F}^{\leq N}) : \rho(2\text{-}WL_E) \subseteq \rho(f)\}$$

$$\overline{k\text{-}LGNN_E} = \{f \in \mathcal{C}_E(K, \mathbb{F}^{\leq N}) : \rho(k\text{-}LGNN_E) \subseteq \rho(f)\} \supset \{f \in \mathcal{C}_E(K, \mathbb{F}^{\leq N}) : \rho(k\text{-}WL_E) \subseteq \rho(f)\}$$

$$\overline{k\text{-}FGNN_E} = \{f \in \mathcal{C}_E(K, \mathbb{F}^{\leq N}) : \rho(k\text{-}FWL_E) \subseteq \rho(f)\}$$

*Proof.* This corollary is a direct consequence of Thm. 34 and Lem. 36, using Lem. 32 to satisfy the sub-algebra assumption of Lem. 36. $\square$

