# OpenReview forum: "Expressive Power of Invariant and Equivariant Graph Neural Networks"
_ICLR.cc/2021/Conference — ICLR 2021 Spotlight_

### Official Review · AnonReviewer3 · 2020-10-22
**Interesting topic and multiple strong contributions**

**Rating:** 9
**Confidence:** 4

**Review:**

post-discussion:
I read the author's response and other reviews. I still think this is a very strong submission and would like to see it accepted.
I encourage the authors to add a section discussing their generalization of SW theorem + Theorems 31,32 as the authors suggested. another small point: I couldn't easily find the definition of "stable by concatenation".


Summary:

The paper studies the expressive power of several classes of recently suggested models: message passing neural networks, Linear GNNs, and Folklore GNNs. Understanding the expressivity of GNNs is a timely and important topic since unlike other neural architectures as fully connected networks, the most widely-used models today, message-passing networks, are not universal. The paper first formally defines the three model classes mentioned above. The paper then reviews previous results on the separation power of these models (can a model tell the difference between two non-isomorphic graphs) and summarizes them, as well as new results they obtain, in prop. 3. Theorem 4 then connects these separation results to the approximation power of the model classes by a novel adaptation of a recently introduced generalization of the Stone-Weierstrass theorem.  The authors conclude the paper with an experimental section showing that folklore GNNs perform significantly better than other models in solving the quadratic assignment problem, which supports the theoretical claims


Strong points:

New strong results: the paper closes the following knowledge gaps: (1) Folklore GNNs: k Folklore GNNs are as strong as k+1 WL, (2) Equivariant separation: connecting the separation power of equivariant (rather than invariant) models to their equivariant WL counterpart (3) Link separation results to approximation results: the authors suggest a generalization of the stone Weierstrass theorem and use it to show new approximation results.
The paper succinctly summarizes the current results on the separation power of various GNN classes.
The paper Introduces new tools from approximation theory. These tools seem important and will be useful for future papers targeting the expressive power of invariant and equivariant deep models
Interesting experimental results. Showing for the first time that more expressive models  (2-FWL) substantially outperform previous (2-WL) models on an important task.

Weak points:

None. I have several comments on the exposition. See below.
In addition, I would like to ask the authors to explicitly and succinctly state their generalization of the Stone-Weierstrass theorem for general symmetries and deep models so it would be easy for future work to use it. Please do so for both invariant and equivariant case, and consider stating it in the paper itself.



Recommendation:

This is a high-quality paper with several strong contributions as listed above. Most importantly, the paper proves several important results on the expressive power of GNNs and introduces useful mathematical tools that I am sure will be used by the community. Hence, I strongly encourage the acceptance of this paper (and would also recommend it as a spotlight/oral presentation).

Minor comments:

* Can the authors comment on “Building powerful and equivariant graph neural networks with structural message-passing” (Vignac et al. 2020). Specifically, Theorem 2 shows that their models can separate any pair of non-isomorphic graphs while using only second-order tensors
* “So in terms of separating power, when restricted to tensors of order k, k-FGNN is the most powerful architecture. ” => Among the architectures considered in this paper.
* “For regular-grid graphs, they match classical convolutional networks which by design can only approximate translation-invariant functions and hence have limited expressive power. In this paper, we focus instead on more expressive architectures.” => I believe that mentioning the relation between GNNs and 2-WL here is a better example of the limited expressive power of GNNS.
* “Note that 2-WL solves k-cliques for k <= 6 (Fürer (2017)) so that these networks are probably not comparable to 2-WL.” => As far as I know 2WL cannot detect triangles. Please explain.
* k-FWL: maybe write 2 FGNN layers explicitly and discuss matrix multiplication connection?
* Appendix D is written really nicely with good examples
* Algebras are closed under multiplication but sets of neural networks are not (although products can be easily approximated in most cases). I believe this gap is filled in Appendix D but would like to ask the authors to explain it in the main text.
* The paper has several typos.

---

> ### Author Response · Authors · 2020-11-16
> **Author response to Reviewer 3**
>
> We warmly thank the reviewer for his/her comments and constructive suggestions.
> - “In addition, I would like to ask the authors to explicitly and succinctly state their generalization of the Stone-Weierstrass theorem for general symmetries and deep models so it would be easy for future work to use it. Please do so for both invariant and equivariant case, and consider stating it in the paper itself.”
> We propose to add Theorems 31 & 32 in the paper itself. These are the results we directly applied to MGNNs, FGNNs and LGNNs to prove our main results.
> However, for the equivariant case, we use explicitly that the group of symmetries is $\mathcal S_n$ to prove Theorem 32 using Theorem 20. So, for another group of symmetries, on has to come back to Thm. 20 and adapt our proof of Thm. 32. For now, the generalization of Thm. 32 to arbitrary groups of symmetries is left as future work. If you think this is relevant, we can try to add Thm. 20 too to the main paper (subject to space constraints).
> -“Can the authors comment on “Building powerful and equivariant graph neural networks with structural message-passing” (Vignac et al. 2020). Specifically, Theorem 2 shows that their models can separate any pair of non-isomorphic graphs while using only second-order tensors”
> This work seems indeed very promising. However, concerning their Thm. 2, the authors themselves explain some shortcomings of their result  in their appendix B.4.
> We quote their appendix B.4 : “The main assumption of our proof is that the aggregation and update functions can exactly compute any function of their input — this is impossible in practice. An extension of our argument to a universal approximation statement would entail substituting the aggregation and update functions by appropriate universal approximators. In particular, the aggregation function manipulates a set of n × c matrices, which can be represented as a n × n × c tensor with some lines zeroed out. [...] Therefore, proving that a given parametrization of an SMP can be used to approximately reconstruct the adjacency matrix hinges on the identification of a simple universal approximator of equivariant functions on n × n × c tensors.”
> Essentially, their proof relies on an universal equivariant approximator for tensors of order 2. However, no such tractable approximators have been found yet. The only way to do this for now is to use tensors of arbitrary high order. Therefore, to be effectively implemented as described in their Thm 2, tensors of arbitrary high order are needed.
> On the contrary, in this work, we study tractable and practical architectures, which are not universal approximators and characterize their approximation power.
> Nevertheless, we agree that his recent architecture is promising and look forward to a theoretical analysis of a tractable version of SMP. As explained in our response to Reviewer 4, once the separating power of such version of SMP is known, our theorems will apply to characterize its approximation power.
> -“For regular-grid graphs, they match classical convolutional networks which by design can only approximate translation-invariant functions and hence have limited expressive power. In this paper, we focus instead on more expressive architectures.” => I believe that mentioning the relation between GNNs and 2-WL here is a better example of the limited expressive power of GNNS.
> Here we wanted to discuss the connection with GCN like (Defferrard et al., 2016). In this work, the architecture is stacking message-passing layers with pooling layers so that the separating power of such GCN is not known because the pooling layer does not fall into our framework.
> -“Note that 2-WL solves k-cliques for k <= 6 (Fürer (2017)) so that these networks are probably not comparable to 2-WL.” => As far as I know 2WL cannot detect triangles. Please explain.”
> Thank you for highlighting our mistake here. Fürer (2017) indeed refers to 2-FWL, which is equivalent to 3-WL, and not 2-WL. We will correct it.
> -"k-FWL: maybe write 2 FGNN layers explicitly and discuss matrix multiplication connection?"
> We agree that the definition of k-FGNN is much easier for the particular case k=2 and if place permit, we will add a discussion about the relation between 2-FGNN layers and matrix multiplication (feature wise).
> “Algebras are closed under multiplication but sets of neural networks are not (although products can be easily approximated in most cases). I believe this gap is filled in Appendix D but would like to ask the authors to explain it in the main text.”
> Indeed, this is addressed in appendix D.9 precisely. The idea is that, in the theoretical analysis, it is almost as if the MLPs in the final layers of GNNs can be replaced by continuous functions, as MLPs are universal approximators. This new “ideal” class of models is now stable by multiplication. We will add this clarification to the main paper.
>
> We thank the reviewer for the other suggestions that we will take into account as best we can.

---

### Official Review · AnonReviewer2 · 2020-10-28

**Rating:** 6
**Confidence:** 4

**Review:**

1. Summary:

This paper compares the expressive power of three types of invariant and equivariant GNNs against the Weisfeiler-Lehman (WL) tests, proves function approximation results for these GNNs, and demonstrates that 2-FGNN_E works well for the quadratic assignment problem.

2. Clearly state the recommendation (accept or reject) with one or two key reasons for this choice:

I am slightly leaning towards recommending a rejection to this paper, with the main reason being that I am not sure if the contributions are sufficiently significant, as I will elaborate below.

3. Arguments for the recommendation and questions for the authors:

a) It is nice to establish and summarize a thorough comparison of the expressive powers of different GNN families as well as k-WL, and also to incorporate not only invariant but also equivariant GNNs into consideration. However, among the results in Section 4.2, the only novel one is the equivalence between k-FGNN and (k+1)-WL, which, one may argue, is also not surprising considering the equivalence between k-FWL and (k+1)-WL. The value of 2-FGNN has also been demonstrated by the work of [1].

b) The overall framework for the comparison of expressive power as well as function approximation resembles that of [2], which characterizes the expressive power of GNNs and WL tests by the sigma algebra they induce on the space of graphs, and also establishes an equivalence between the ability to distinguish non-isomorphic graphs and the ability to approximate invariant functions. Hence, the authors may want to elaborate more on the connection between this work and [2].

4. Additional comments:

The authors claim that 3-LGNN or 2-FGNN can count the number of 6-cycles, which is an improvement upon the result in [3], which implies that 6-LGNN can count 6-cycles. Technically, however, it is only known (by [4], to my knowledge) that 3-WL can count the number of times 6-cycles appear as subgraphs of a graph, but not induced subgraphs, whereas [3] is concerned with the counting of both subgraphs and induced subgraphs. Therefore, I would recommend that the authors make this statement more precise.

References:

[1] Haggai Maron, Heli Ben-Hamu, Hadar Serviansky, and Yaron Lipman. Provably powerful graph networks.

[2] Zhengdao Chen, Soledad Villar, Lei Chen, and Joan Bruna. On the equivalence between graph isomorphism testing and function approximation with GNNs.

[3] Zhengdao Chen, Lei Chen, Soledad Villar, and Joan Bruna. Can graph neural networks count substructures?

[4] Martin Fu ̈rer. On the Combinatorial Power of the Weisfeiler-Lehman Algorithm.


=== Post Rebuttal ===

I appreciate the careful response provided by the authors, which reminds me of the significance of extending existing theoretical results from invariant to equivariant models. Therefore I have raised my score by 1 point.

---

> ### Author Response · Authors · 2020-11-16
> **Author response to Reviewer 2**
>
> We sincerely thank the reviewer for its constructive criticism. We will respond to each point in turn.
> - “a) It is nice to establish and summarize a thorough comparison of the expressive powers of different GNN families as well as k-WL, and also to incorporate not only invariant but also equivariant GNNs into consideration. However, among the results in Section 4.2, the only novel one is the equivalence between k-FGNN and (k+1)-WL, which, one may argue, is also not surprising considering the equivalence between k-FWL and (k+1)-WL. The value of 2-FGNN has also been demonstrated by the work of [1].”
> We respectfully disagree on this point: to the best of our knowledge, the separating power of GNN was only studied in the invariant case.. In Proposition 3, only the left column of results (4) (5) (6) were known, we stated them for completeness. We do not know any previous result about separating power for equivariant GNNs.
> Moreover, studying the separating powers of GNNs is not the main objective of our work, which is to characterize the approximation power of realistic GNN classes.
> We agree that the empirical value of 2-FGNN was already shown in [1] but only for the invariant case. We explicitly refer to [1] for experimental results in the invariant case (graph classification task, graph regression…), see the “Empirical results for the Quadratic Assignment Problem” paragraph in the Introduction. In our paper, we only look at experimental results validating 2-FGNN in the equivariant case.
> - “b) The overall framework for the comparison of expressive power as well as function approximation resembles that of [2], which characterizes the expressive power of GNNs and WL tests by the sigma algebra they induce on the space of graphs, and also establishes an equivalence between the ability to distinguish non-isomorphic graphs and the ability to approximate invariant functions. Hence, the authors may want to elaborate more on the connection between this work and [2].”
> We agree that our results resemble that of [2], indeed as stated in our paper, our results allow us to recover the theoretical results in [2]. However, we believe results in [2] are much more limited than ours. First, [2] only deals with the invariant case whereas we deal with the equivariant case too which is crucial for our applications with Graph alignment. Second, theoretical results in [2] only deal with universality results, which are not relevant for practical GNN classes, whereas we derive approximation results even for architectures that are not universal (i.e. dealing with fixed order tensors).
> Finally, the framework of representation power based on sigma-algebra they introduce only makes sense for finite input spaces, while the notion of discriminatory power is much more flexible. However, we agree that,  when the input space $X$ is finite, these two notions can be linked. The separating power of a class of function $\mathcal F$ indeed characterizes the sigma algebra generated by $\mathcal F$, denoted by $\sigma(\mathcal F)$ : one can shows that, $\sigma(\mathcal F) = \sigma(X/\rho(\mathcal F))$, where $\sigma(X/\rho(\mathcal F))$ is to be understood as in [2], i.e. as the sigma algebra generated by the equivalence classes of $\rho(\mathcal F)$. If the reviewers think this remark is relevant we can add this remark to the appendix.
>   (Note that, here, “sigma algebra” has nothing to do with the term “algebra” used in our work : here, a sigma algebra refers to the measure theory concept. )
> - “The authors claim that 3-LGNN or 2-FGNN can count the number of 6-cycles, which is an improvement upon the result in [3], which implies that 6-LGNN can count 6-cycles. Technically, however, it is only known (by [4], to my knowledge) that 3-WL can count the number of times 6-cycles appear as subgraphs of a graph, but not induced subgraphs, whereas [3] is concerned with the counting of both subgraphs and induced subgraphs. ”
> Thank you for pointing out this imprecision, we will clarify this.
>
> References:
> [1] Haggai Maron, Heli Ben-Hamu, Hadar Serviansky, and Yaron Lipman. Provably powerful graph networks.
> [2] Zhengdao Chen, Soledad Villar, Lei Chen, and Joan Bruna. On the equivalence between graph isomorphism testing and function approximation with GNNs.
> [3] Zhengdao Chen, Lei Chen, Soledad Villar, and Joan Bruna. Can graph neural networks count substructures?
> [4] Martin Fu ̈rer. On the Combinatorial Power of the Weisfeiler-Lehman Algorithm.

---

### Official Review · AnonReviewer1 · 2020-10-29
**Interesting paper with fixable deficiencies (Likely to advocate for it)**

**Rating:** 8
**Confidence:** 4

**Review:**

#### Goal

The paper describes the approximation power of certain types of graph neural networks.  It considers Message Passing GNNs (MGNNs), and two GNN-type methods proposed by Maron et al., k-FGNN and k-LGNN.

The main challenge with some of the current expressiveness analysis of neural networks is their reliance on discrete attributes. This line of work shows how these can be extended to a general setting.

#### Quality

- This is a great paper, well written, nice appendix, on an interesting topic. The only drawback is that the approach was narrowly applied to k-FGNN and k-LGNN, comparing them to k-WL approaches.

- My main concern is being too narrowly focused on k-FGNN and k-LGNN without a proper empirical comparison with other more expressive competing methods. For instance, it does not even discuss alternative approaches using group averaging. I think the paper would significantly broaden its audience if there is a comparison with novel k-ary group averaging approaches (e.g., Chen et al., 2020, Murphy et al. 2019) in the empirical section.

#### Clarity

The paper is clear and well-written. The notation follows the standard notation used in prior work.
In one aspect the notation could improve: \subset should be replaced by \subseteq to avoid confusion, and be more in line with the use of \subsetneq. I was not sure what \subseteq meant until I saw \subsetneq. It is unfortunately but different branches of mathematics use \subseteq differently and the symbol should be avoided altogether.
The appendix is well organized, with an index that helps the reader find proofs and definitions.

#### Originality
- The work follows from prior work of Maehara & Hoang (2019), Chen et al., 2019, Geerts (2020a), etc.
- There is some novelty in the proof approach
- New results related to k-FGNN and k-LGNN.

#### Significance

The work is of interest to a narrow sub-community working on k-FGNN and k-LGNN. The MGNN results were known but recovered here in a different way (i.e., there is value). The k-FGNN and k-LGNN results are new but also very narrow due to the sub-community size.

The empirical results essentially do not compare with any other alternative approach. Until the community is convinced that k-FGNN and k-LGNN can empirically compete with other approaches (e.g., Morris et al. 2019, Permutation group averages like  Chen et al. 2020, Murphy et al. 2019, among others), these methods will not see broad applicability.

Chen, Zhengdao, Lei Chen, Soledad Villar, and Joan Bruna. "Can graph neural networks count substructures?." NeurIPS (2020).

Murphy, Ryan L., Balasubramaniam Srinivasan, Vinayak Rao, and Bruno Ribeiro. "Relational pooling for graph representations." ICLR (2019).

#### Pros
- Solid theoretical work showing the expressiveness of k-FGNN and k-LGNN
- Interesting ports of ideas from other papers to improve proof techniques of current GNN papers

#### Cons

- The authors seem unfamiliar with group average approaches that increase the expressiveness of MGNNs. These can also be universal approximations. Permutation group averages like  Chen et al. 2020, Murphy et al. 2019, among others, have increased expressiveness of MGNN using the same MGNN architecture.  Since there has been a lot of recent activity related to these methods, this is a blind spot of this work. For instance, “Note also, that if the nodes are given distinct features, MGNNs become much more expressive Loukas (2019) but this is meaningless in some problems such as our graph alignment problem.” does not point to the fact that a simple group average approach would fix the issue. This way, the work feels unnecessarily narrow.

#### Typos:
- "test to express the discriminatory power of equivariant architectures For this" => "test to express the discriminatory power of equivariant architectures. For this"
- "Maron et al. (2018) Geerts (2020a)" => "Maron et al. (2018); Geerts (2020a)"

----------------------------------------

After rebuttal:

Thanks for the answers. I have raised my score even though I still think the paper could have done a better job at comparing against other methods.

- Regarding group-averaging methods for the equivariant case: It is a trivial extension, specially the approach of giving GNNs unique IDs and then averaging their representation, which (Loukas, 2020) shows it is universal (but (Loukas, 2020) did not consider averaging). Regarding training, it is always performed stochastically via data augmentation and Monte Carlo estimated in test. For the generalization error of the stochastic optimization, it is still unknown (some new results show promise (Chen et al. 2020) and (Lyle et al. 2020)) in the same way that the generalization performance of other universal methods is still unknown.

- Chen, S., Dobriban, E., & Lee, J. H. (2020). Invariance reduces Variance: Understanding Data Augmentation in Deep Learning and Beyond.
- Lyle, Clare, Mark van der Wilk, Marta Kwiatkowska, Yarin Gal, and Benjamin Bloem-Reddy. "On the Benefits of Invariance in Neural Networks." arXiv preprint arXiv:2005.00178 (2020).
- Loukas, Andreas. "How hard is to distinguish graphs with graph neural networks?." Advances in Neural Information Processing Systems 33 (2020).

---

> ### Author Response · Authors · 2020-11-16
> **Author response to Reviewer 1**
>
> We sincerely thank the reviewer for the constructive comments and criticism. We will respond to each point in turn.
>
> - “In one aspect the notation could improve: \subset should be replaced by \subseteq to avoid confusion, and be more in line with the use of \subsetneq. I was not sure what \subseteq meant until I saw \subsetneq.“
> Thank you for your suggestion, we will update our notations.
> - “It does not even discuss alternative approaches using group averaging. I think the paper would significantly broaden its audience if there is a comparison with novel k-ary group averaging approaches (e.g., Chen et al., 2020, Murphy et al. 2019) in the empirical section.”
> For our theoretical results, we decided to focus on what we consider to be the most studied architectures theoretically: MGNN, LGNN and FGNN. We agree that the relational pooling introduced in Murphy et al. and studied in Chen et al. is a very interesting idea but to the best of our knowledge, this technique is only valid for invariant GNNs. We do not see any simple extension of this idea to the equivariant case which is the main focus of our empirical study. We will add this remark in our paper.
> -"The work is of interest to a narrow sub-community working on k-FGNN and k-LGNN. The MGNN results were known but recovered here in a different way (i.e., there is value). The k-FGNN and k-LGNN results are new but also very narrow due to the sub-community size.
> The empirical results essentially do not compare with any other alternative approach. "
> For our empirical results, we did compare to MGNN, and two ‘non learning’ algorithms based on SDP and spectral methods.
> - "Until the community is convinced that k-FGNN and k-LGNN can empirically compete with other approaches (e.g., Morris et al. 2019, Permutation group averages like  Chen et al. 2020, Murphy et al. 2019, among others), these methods will not see broad applicability."
> For machine learning task requiring an invariant representation of the graph (like graph classification, graph regression …), we agree with the reviewer that FGNNs will probably be useful only in very particular cases as this is an already ‘mature’ field where MGNNs have been customized for specific tasks/datasets and are both computationally efficient and state of the art. However, we believe the situation is rather different for machine learning tasks requiring an equivariant representation of the graph, i.e. a node embedding as in our QAP application. For such tasks, we believe 2-FGNN can empirically compete and even outperform standard MGNN. We are currently working on more applications and got encouraging preliminary results in this direction.
> - “The authors seem unfamiliar with group average approaches that increase the expressiveness of MGNNs. These can also be universal approximations. Permutation group averages like  Chen et al. 2020, Murphy et al. 2019, among others, have increased expressiveness of MGNN using the same MGNN architecture. “
> Regarding Relational Pooling (RP) and our theoretical analysis :
> As far as we know, the ideal RP, which is a universal approximator, cannot be used for large graph due to its complexity of $O(|V|!)$.
> Regarding the other classes of RP GNN, our theorems in the invariant case can be almost immediately applied to them to characterize their approximation power, but the result involve their separating powers, which have not been studied much yet. Consider, for instance, $k$-ary RP by Murphy et al. 2019, with an additional final MLP. This is an invariant class of GNNs which actually satisfies the assumptions of Theorem 31. Therefore, the set of function these GNNs can approximate is, (informally),
> $$\{f \in \mathcal{C}_I(X, Y): \rho(k\text{-RP-GNN}) \subset \rho(f)\}$$
> However, $\rho(k\text{-RP-GNN})$ is not yet known. Thus, the study of the separating power of these classes of GNN is still an interesting question for the community, but, once it is known, our results will automatically give their approximation power.
> Moreover, in this work we were interested in theoretically studying both invariant and equivariant GNN, especially since QAP needs an equivariant architecture. As far as we know, RP-based architecture have not yet been extended to the equivariant case.
> Furthermore, we do not see how a group average approach combined with Loukas (2019) would fix the issue.
>
> Chen, Zhengdao, Lei Chen, Soledad Villar, and Joan Bruna. "Can graph neural networks count substructures?." NeurIPS (2020).
> Murphy, Ryan L., Balasubramaniam Srinivasan, Vinayak Rao, and Bruno Ribeiro. "Relational pooling for graph representations." ICLR (2019).

---

### Official Review · AnonReviewer4 · 2020-11-02
**Solid theoretical contribution to universality proofs of equivariant networks**

**Rating:** 8
**Confidence:** 3

**Review:**

Summary:
The authors prove several statements about the expressiveness of different classes of graph neural nets (GNNs): conventional message passing networks, linear GNNs (LGNN) and “folklore GNNs” (FGNN). The novel theoretical contributions include analysis of expressiveness of FGNNs that use tensors of arbitrary order in terms of comparison to the Weisfeiler-Lehman tests; a characterization of the functions that these classes of networks can approximate; universality of FGNN as the tensor order goes to infinity. The results are based on a general Stone-Weierstrass-like theorem for equivariant functions. Prior universality results can be recovered as special cases. The authors have a simple experiment that show in a limited setting that a practical implementation agrees with the theory.

Strengths:
-	The paper and appendix are very well written and relatively well understandable for me, unfamiliar with universality proofs. Particular examples of clear writing include: the statement of Theorem 4 is clearly explained below the theorem; providing Example 16 directly after Corollary 15 aids exposition; using example 17 to motivate prop 18.
-	The authors use a very general statement (Thm 20) to derive their results, making them generally applicable.
-	The authors derive a substantial number of expressiveness results from the general theory.

Weaknesses & suggestions for improvement:
-	The main paper only sets up the problem and states the main results, while all theoretical contributions are done in the appendix. The main paper would be more self-contained if some more intuition for the proofs was given in the main paper.
-	The experiments seem to not compare to LGNN. Adding this comparison would help making an empirical argument for why FGNN is best.

Recommendation:
Although I am not very familiar with the field of universality proofs, the paper appears to me to be a very solid contribution to the field and I recommend publication.

Minor points / suggestions:
-	In several instances, the authors write “a compact” without a noun. Is this conventional language?
-	Below Eq 2, there is an F without subscript. Is this the same for all layers?
-	In App C.3, in the second line of the equation, are there suffices missing on the left-hand side?
-	Sec 4.3, first sentence, “the set invariant”, missing “of”
-	Sec 4.3, typo “To clarify the meaning of theses statements”
-	App D.3: typo “as every function satisfy” should be “satisfies”
-	Example 16, typo “is able to learn function” should be plural
-	App D.3, proof of Corollary 15: define when a class separates a set around “clearly separates X_\mathcal{F}”. Presumably means same as separates points?
-	Eq 17, Define S \cdot \mathcal{F}, presumably as the scalar-vector product of the outputs of the functions?
-	Corollary 19, assumption 3, what does “pairwise distinct coordinates” mean?
-	App D.3, typo “For an equivariant function, for any” missing $f$
-	Example 21, \mathcal{C}_{eq} should be \mathcal{C}_E? Happens later more.
-	Lemma 24, define R{X_1, .., X_p] as polynomials

### Post rebuttal
I thank the authors for their response. My previous rating still applies.

---

> ### Author Response · Authors · 2020-11-16
> **Author response to Reviewer 4**
>
> We thank the reviewer for the constructive comments and are grateful for the numerous and helpful suggestions. We will respond to each point in turn.
> - “The main paper only sets up the problem and states the main results, while all theoretical contributions are done in the appendix. The main paper would be more self-contained if some more intuition for the proofs was given in the main paper.”
> If accepted, we will use the additional page for the camera ready version of the paper to include part of the theoretical contributions with examples that are now in the appendix.
> Following the demands of Reviewer 3, we propose to add Theorems 31 & 32 in the paper itself with some of the explanations in D. These are the results we directly applied to MGNNs, FGNNs and LGNNs to prove our main results.
> - “The experiments seem to not compare to LGNN. Adding this comparison would help making an empirical argument for why FGNN is best.”
> MGNNs seem much more popular than LGNNs in practice.  They have the same expressive power as MGNN when dealing with tensors of order 2, in particular, they are useless for our graph alignment problem with regular graphs. Note also that we compare our implementation of FGNN directly with existing results in the literature with MGNN, SDP and spectral algorithms. This makes sure that alternative implementations have been carefully designed.
>     Moreover, (Maron 2019) empirically compared the equivariant versions of LGNN and FGNN and found FGNN performed better.
>     Therefore, we believe that for our QAP problem, LGNN will have similar results as MGNN ( we are ready to implement LGNN if required).
> - “In several instances, the authors write “a compact” without a noun. Is this conventional language?”
>     After looking it up, it seems to be an abuse of language. We will add “subset” or “set” for clarity.
> - “Below Eq 2, there is an F without subscript. Is this the same for all layers?”
> No it depends on the layer, we will fix this.
> - “In App C.3, in the second line of the equation, are there suffices missing on the left-hand side?”
> Indeed, subscripts are missing. The left-hand side of the second equation at the bottom of page 18 should be $I^k(G)_{\mathbf{i}, r, s, e+2}$. This actually makes us realize that $\mathbb{F}_1$ defined above should be $\mathbb{R}^{k^2 \times (e+2)}$. We will improve the clarity of this paragraph by harmonizing the notations with Lemma 8, i.e. by using $p_0$ instead of $e+1$.
> - “App D.3, proof of Corollary 15: define when a class separates a set around “clearly separates X_\mathcal{F}”. Presumably means same as separates points?”
> Indeed, it means that for any $x, y \in X_{\mathcal{F}}$ there exists $f \in \hat{\mathcal{F}}$ such that $f(x) \neq f(y)$. We will clarify this at the beginning of appendix D.
> - “Eq 17, Define S \cdot \mathcal{F}, presumably as the scalar-vector product of the outputs of the functions? “
> Indeed, we will clarify in D.1
> - “Corollary 19, assumption 3, what does “pairwise distinct coordinates” mean?”
> $f(x)$ is a vector $\mathbb R^p$, so, by “pairwise distinct coordinates” we mean that, for any indices $i$, $j$ with $i \neq j$, the coordinates $i$ and $j$ of $f$ differ, i.e. $f(x)_i \neq f(x)_j$.
>
> We warmly thank the reviewer for listing  the other typos and suggestions.

---

### Decision · Program_Chairs · 2021-01-07
**Final Decision**

**Decision:**

Accept (Spotlight)

**Comment:**

This paper is concerned with the ongoing research program of mapping the approximation power of different GNN architectures. It provides significant advances in the study of equivariant GNNs and nice extensions in the invariant case by closing existing gaps between distinct GNN families.
All reviewers agreed that this is a strong submission with substantial new theoretical results. The AC recommends a strong acceptance.